# No Equations Needed: Learning System Dynamics Without Relying on Closed-Form ODEs

**Krzysztof Kacprzyk**
University of Cambridge
kk751@cam.ac.uk

**Mihaela van der Schaar**
University of Cambridge
mv472@cam.ac.uk

## Abstract

Data-driven modeling of dynamical systems is a crucial area of machine learning. In many scenarios, a thorough understanding of the model's behavior becomes essential for practical applications. For instance, understanding the behavior of a pharmacokinetic model, constructed as part of drug development, may allow us to both verify its biological plausibility (e.g., the drug concentration curve is non-negative and decays to zero in the long term) and to design dosing guidelines (e.g., by looking at the peak concentration and its timing). Discovery of closed-form ordinary differential equations (ODEs) can be employed to obtain such insights by finding a compact mathematical equation and then analyzing it (a two-step approach). However, its widespread use is currently hindered because the analysis process may be time-consuming, requiring substantial mathematical expertise, or even impossible if the equation is too complex. Moreover, if the found equation's behavior does not satisfy the requirements, editing it or influencing the discovery algorithms to rectify it is challenging as the link between the symbolic form of an ODE and its behavior can be elusive. This paper proposes a conceptual shift to modeling low-dimensional dynamical systems by departing from the traditional two-step modeling process. Instead of first discovering a closed-form equation and then analyzing it, our approach, direct semantic modeling, predicts the semantic representation of the dynamical system (i.e., description of its behavior) directly from data, bypassing the need for complex post-hoc analysis. This direct approach also allows the incorporation of intuitive inductive biases into the optimization algorithm and editing the model's behavior directly, ensuring that the model meets the desired specifications. Our approach not only simplifies the modeling pipeline but also enhances the transparency and flexibility of the resulting models compared to traditional closed-form ODEs.

## 1 Introduction

**Background: data-driven modeling of dynamical systems through ODE discovery.** Modeling dynamical systems is a pivotal aspect of machine learning (ML), with significant applications across various domains such as physics (Raissi et al., 2019), biology (Neftci & Averbeck, 2019), engineering (Brunton & Kutz, 2022), and medicine (Lee et al., 2020). In real-world applications, understanding the model's behavior is crucial for verification and other domain-specific tasks. For instance, in drug development, it is important to ensure the pharmacokinetic model (Mould & Upton, 2012) is biologically plausible (e.g., the drug concentration is non-negative and decays to zero), and the dosing guidelines may be set up based on the peak concentration and its timing (Han et al., 2018). One effective approach to gain such insights is the discovery of closed-form ordinary differential equations (ODEs) (Bongard & Lipson, 2007; Schmidt & Lipson, 2009; Brunton et al., 2016a), where a concise mathematical representation is first found by an algorithm and then analyzed by a human.

**Motivation: the primary goal of discovering a closed-form ODE is its semantic representation.**
We assume that the primary objective of discovering a closed-form ODE, as opposed to using a black-box model, is to have a model representation that can be analyzed by humans to understand the model's behavior (Qian et al., 2022). Under this assumption, the specific form of the equation, its *syntactic representation*, is just a medium that allows one to obtain the description of the model's behavior, its *semantic representation*, through post-hoc mathematical analysis. We call the process of

discovering an equation and then analyzing it a *two-step modeling* approach. An illustrative example showing the difference between a syntactic and semantic representation of the same ODE (logistic growth model (Verhulst, 1845)) can be seen in Figure 1.

**Limitations of the traditional two-step modeling.** The traditional two-step modeling pipeline, where an ODE is first discovered and then analyzed to understand its behavior, presents several limitations. The analysis process can be time-consuming, and requiring substantial mathematical expertise. It may even be impossible if the discovered equation is too complex. Furthermore, as the link between syntactic and semantic representation may not be straightforward, modifying the discovered equation to adjust the model's behavior may pose significant challenges. This complicates the refinement process and limits the ability to ensure that the model meets specific requirements.

**Proposed approach: direct semantic modeling.** To overcome these limitations, we propose a novel approach, called *direct semantic modeling*, that shifts away from the traditional two-step pipeline. Instead of first discovering a closed-form ODE and then analyzing it, our approach generates the semantic representation of the dynamical system directly from data, eliminating the need for post-hoc mathematical analysis. By working directly with the semantic representation, our method allows for intuitive adjustments and the incorporation of constraints that reflect the system's behavior. This direct approach also facilitates more flexible modeling and improved performance, as it does not rely on a compact closed-form equation.

**Contributions and outline.** In Section 3, we define the *syntactic* and *semantic representation* of ODEs, discuss the limitations of the traditional *two-step modeling pipeline* and introduce *direct semantic modeling* as an alternative. We formalize semantic representation (Section 4) and then use it to introduce *Semantic ODE* in Section 5, a concrete instantiation of our approach for modeling 1D systems. Finally, we illustrate its practical usability and flexibility in (Section 6).

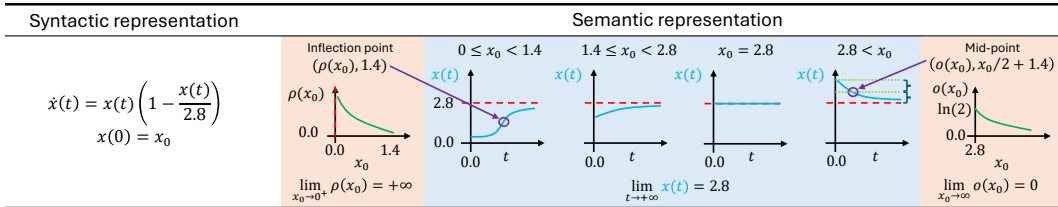

Figure 1: Syntactic representation of a logistic growth model refers to its symbolic form, whereas semantic representation describes its behavior for different initial conditions.

## 2 FORECASTING MODELS AND DISCOVERY OF CLOSED-FORM ODEs

In this section, we formulate the task of discovering closed-form ODEs from data and show how it can be reinterpreted as a more general problem of fitting a forecasting model.

Let $\boldsymbol{f} : \mathbb{R}^{M+1} \to \mathbb{R}^M$, and let $\mathcal{T} = (t_0, +\infty)$. A system of $M$ ODEs is described as

$$\dot{\boldsymbol{x}}(t) = \boldsymbol{f}(\boldsymbol{x}(t), t) \ \forall t \in \mathcal{T}, \tag{1}$$

where $\boldsymbol{x} : \mathcal{T} \to \mathbb{R}^M$ is called a *trajectory* and $\dot{x}_m = \frac{\mathrm{d}x_m}{\mathrm{d}t}$ is the derivative of $x_m$ with respect to $t$. We also assume each $x_m \in C^2(\mathcal{T})$, i.e., it is twice continuously differentiable on $\mathcal{T}$.[1] We denote the dataset of observed trajectories as $\mathcal{D} = \{(t_n^{(d)}, \boldsymbol{y}_n^{(d)})_{n=1}^{N_d}\}_{d=1}^{D}$, where each $\boldsymbol{y}_n^{(d)}$ represents the noisy measurement of some ground truth trajectory $\boldsymbol{x}^{(d)}$ governed by $\boldsymbol{f}$ at time point $t_n^{(d)}$.

A closed-form equation (Qian et al., 2022) is a mathematical expression consisting of a finite number of variables, constants, binary arithmetic operations $(+, -, \times, \div)$, and some well-known functions such as exponential or trigonometric functions. A system of ODEs is called closed-form when each function $f_m$ is closed-form. The task is to find a closed-form $\boldsymbol{f}$ given $\mathcal{D}$.

Traditionally (Bongard & Lipson, 2007; Schmidt & Lipson, 2009) discovery of governing equations has been performed using genetic programming (Koza, 1992). In a seminal paper, Brunton et al.

---

[1]We assume $x_m \in C^2(\mathcal{T})$ instead of $C^1(\mathcal{T})$, so that we can discuss curvature and inflection points.

(2016a) proposed to represent an ODE as a linear combination of terms from a prespecified library. This was followed by numerous extensions, including implicit equations (Kaheman et al., 2020), equations with control (Brunton et al., 2016b), and partial differential equations (Rudy et al., 2017). Approaches based on weak formulation of ODEs that allow to circumvent derivative estimation have also been proposed (Messenger & Bortz, 2021a; Qian et al., 2022). The extended related works section can be found in Appendix F.

Each system of ODEs $\boldsymbol{f}^2$ defines a forecasting model[3] $\boldsymbol{F}$ through the initial value problem (IVP), i.e., for each initial condition $\boldsymbol{x}(t_0) = \boldsymbol{x}_0 \in \mathbb{R}^M$, $\boldsymbol{F}$ maps $\boldsymbol{x}_0$ to a trajectory governed by $\boldsymbol{f}$ satisfying this initial condition. Therefore, ODE discovery can be treated as a special case of fitting a forecasting model $\boldsymbol{F} : \mathbb{R}^M \to C^2(\mathcal{T})$.

## 3 FROM DISCOVERY AND ANALYSIS TO DIRECT SEMANTIC MODELING

In this section, we define the syntactic and semantic representations, describe the traditional two-step modeling and its limitations, and introduce our approach, direct semantic modeling.

### 3.1 SYNTAX VS. SEMANTICS.

ODEs are usually represented symbolically as closed-form equations. For instance, $\dot{x}(t) = (1 - x(t))x(t)$. We refer to this kind of representation as *syntactic*.

> **Syntactic representation** of a closed-form ODE refers to its symbolic form, i.e., the arrangement of variables, arithmetic operations, numerical constants, and some well-known functions.

The output of the current ODE discovery algorithms is in the form of syntactic representation. We assume that the primary objective of discovering a closed-form ODE, as opposed to using a black-box model, is to have a model representation that can be analyzed by humans to understand its behavior. Such understanding is necessary to ensure that the model behaves as expected; for instance, it operates within the range of values and exhibits trends consistent with domain knowledge. We call the description of the dynamical system's behavior its *semantic representation*.

> **Semantic representation** describes the behavior of a dynamical system. Semantic representation of a single trajectory may include its shape, properties, and asymptotic behavior, whereas semantic representation of a forecasting model, including a system of ODEs, describes how they change under different conditions, e.g., for different initial conditions.

Comparison between the syntactic and semantic representation of the same ODE is shown in Figure 1.

### 3.2 TWO-STEP MODELING AND ITS LIMITATIONS

Semantic representation of a dynamical system is usually obtained by first discovering an equation (e.g., using an ODE discovery algorithm) and then analyzing it. This *two-step modeling* approach has several limitations (depicted in Figure 2).

- **Analysis** of a closed-form ODE may be time-consuming, and requiring mathematical expertise. It may be impossible if the discovered equation is too complex. As a result, it may introduce a trade-off between better fitting the data and being simple enough to be analyzed by humans.
- **Obtained insights may be nonactionable**. As the link between syntactic and semantic representations is often far from trivial, it is difficult to edit the syntactic representation of the model to cause a specific change in its semantic representation and to provide feedback to the optimization algorithm to solicit a model with different behavior.
- **Incorporation of prior knowledge**. Often the prior knowledge about the dynamical system concerns its semantic representation rather than its syntax. For instance, we may know what shape the trajectory should have (e.g., decreasing and approaching a horizontal asymptote) rather than what kind of terms or arithmetic operations are present in the best-fitting equation.

### 3.3 DIRECT SEMANTIC MODELING

To address the limitations of two-step modeling, we propose a conceptual shift in modeling low-dimensional dynamical systems. Instead of discovering an equation from data and then analyzing it

---

[2]With some regularity conditions to ensure uniqueness of solutions.

[3]In our work we refer to a forecasting model as any model that outputs a trajectory.

to obtain its semantic representation, our approach, *direct semantic modeling*, generates the semantic representation directly from data, eliminating the need for post-hoc mathematical analysis.

**Forecasting model determined by semantic representation** A major difference between our approach and traditional two-step modeling is how the model ultimately predicts the values of the trajectory. Given a system of closed-form ODEs $f$, a forecasting model $F$ is directly given by the equation. We just need to solve the initial value problem (IVP) for the given initial condition. There are plenty of algorithms to do so numerically, the forward Euler method being the simplest (Butcher, 2016). In contrast, the result of direct semantic modeling is a *semantic predictor* $F_{\text{sem}}$ (that corresponds to the semantic representation of the model) that predicts the semantic representation of the trajectory. Then it passes it to a *trajectory predictor* $F_{\text{traj}}$ whose role is to find a trajectory in a given hypothesis space that has a matching semantic representation. The matching does not need to be unique but $F_{\text{traj}}$ needs to be deterministic. Defining $F$ as $F_{\text{traj}} \circ F_{\text{sem}}$ has multiple advantages. No post-hoc mathematical analysis is required as the semantic representation of $F$ is directly accessed through $F_{\text{sem}}$. The model can be easily edited to enforce a specific change in the semantic representation because we can directly edit $F_{\text{sem}}$. Incorporating prior knowledge and feedback into the optimization algorithm is also streamlined and more intuitive. Finally, as the resulting model does not need to be further analyzed, it does not need to have a compact symbolic representation, increasing its flexibility. Figure 2 compares two-step modeling and direct semantic modeling.

**Semantic ODE as a concrete instantiation** We have outlined the core principles of direct semantic modeling above. In the following sections, we propose a concrete machine learning model that realizes these principles. It is a forecasting model that takes the initial condition $x_0 \in \mathbb{R}$ and predicts a 1-dimensional trajectory, $x : \mathcal{T} \to \mathbb{R}$. We call it *Semantic ODE* because it maps an initial condition to a trajectory (like ODEs implicitly do). Although Semantic ODE can only model 1-dimensional trajectories, we believe direct semantic modeling can be successfully applied to multi-dimensional systems. We describe our proposed roadmap for future research to achieve that goal in Appendix G.2. Before we describe the building blocks of Semantic ODE in Section 5, we need a formal definition of semantic representation.

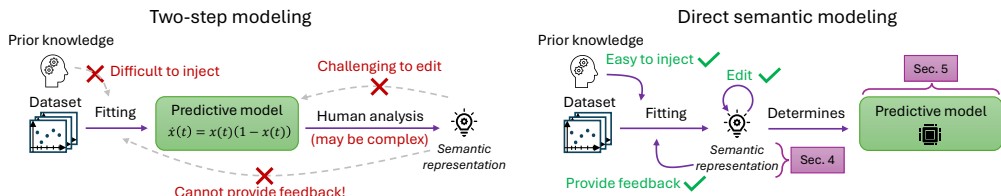

Figure 2: Comparison between two-step modeling and direct semantic modeling. Left: The discovery of closed-form ODEs often allows for human analysis, but editing the equation or providing feedback to the optimization algorithm is challenging. Right: We propose to predict the semantic representation directly from data, which allows for editing the model and steering the optimization algorithm.

## 4 FORMALIZING SEMANTIC REPRESENTATION

To propose a concrete instantiation of direct semantic modeling in Section 5 called *Semantic ODE*, we need to formalize the definition of semantic representation in Section 3 to make it operational. We consider a setting where $F : \mathbb{R} \to C^2(\mathcal{T})$ is a 1D forecasting model (any ODE can be treated as such a model). We first define a semantic representation of a trajectory $x \in C^2(\mathcal{T})$ and then use it to define a semantic representation of $F$ itself.

**Semantic representation as composition and properties** Our definition of semantic representation is motivated by the framework proposed by Kacprzyk et al. (2024b). Following that work, each trajectory $x$ can be assigned a *composition* (denoted $c_x$) that describes the general shape of the trajectory and the set of properties (denoted $p_x$) which is a set of numbers that describes this shape quantitatively. The composition of the trajectory depends on the chosen set of *motifs*. Each motif describes the shape of the trajectory on a particular interval. For instance, "increasing and strictly convex". Given a set of motifs, we can then subdivide $\mathcal{T}$ into shorter intervals such that $x$ is described by a single motif on each of them. This results in a motif sequence and the shortest such sequence is called a composition. The points between two motifs and on the boundaries are called *transition points*. An example of a trajectory, its composition, and its transition points is shown in Figure 3a.

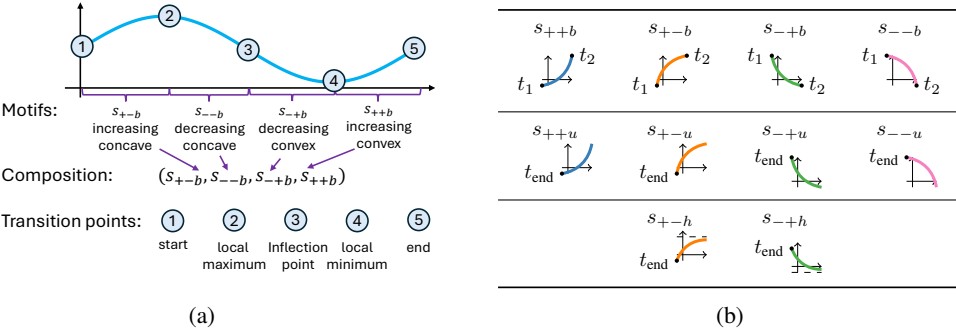

Figure 3: (**a**) Composition and transition points of $x(t) = \sin(t)$ on $[0, 2\pi]$. (**b**) Motif set used in the proposed formalization of semantic representation.

**Extending dynamical motifs**  The set of motifs we choose is inspired by the original set of *dynamical motifs* (Kacprzyk et al., 2024b) but we adjusted and extended it to cover unbounded time domains and different asymptotic behaviors. We define a set of ten motifs, four *bounded* motifs and six *unbounded* motifs. Each motif is of the form $s_{\pm\pm*}$, i.e., is described by two symbols (each $+$ or $-$) and one letter ($b/u/h$). The symbols refer to its first and second derivatives. The letter $b$ signifies the motif is for **b**ounded time domains (e.g., for interval $(t_1, t_2)$). Both $h$ and $u$ refer to **u**nbounded time domains. These motifs are always the last motif of the composition, describing the shape on $(t_{\mathrm{end}}, +\infty)$ where $t_{\mathrm{end}}$ is the $t$-coordinate of the last transition point. $h$ specifically describes motifs with horizontal asymptotes. For instance, $s_{-+h}$ is an unbounded motif that describes a function that is decreasing ($-$), strictly convex ($+$) and with horizontal asymptote ($h$). All motifs are visualized in Figure 3b. Note that we excluded the three original motifs describing straight lines to simplify the modeling process. If necessary, they can be approximated by other motifs with infinitesimal curvature. We denote the set of all compositions constructed from these motifs as $\mathcal{C}$.

**Properties**  Apart from the composition, the semantic representation of a trajectory also involves a set of properties. Ideally, the properties should be sufficient to visualize what each of the motifs looks like and to constrain the space of trajectories with the corresponding semantic representation. Following the original work, we include the coordinates of the transition points as they characterize bounded motifs well. In contrast to their bounded counterparts, the unbounded motifs are not described by their right transition point but by a set of *motif properties*. These, in turn, depend on how we describe the unbounded motif. For instance, we could parameterize $s_{++u}$ as $x(t) = x(t_{\mathrm{end}})2^{(t-t_{\mathrm{end}})/B}$, where $(t_{\mathrm{end}}, x(t_{\mathrm{end}}))$ is the position of the last transition point. In that setting, $B$ is the *property* of $s_{++u}$ that describes the doubling time of $x$ ($x(t + B) = 2x(t)$). In reality, choosing a good parametrization with meaningful properties is challenging, and we discuss it in more detail in Appendix D.2. The set of properties also includes the first derivative at the first transition point ($t_0$) and the first and the second derivative at the last transition point ($t_{\mathrm{end}}$). They are needed for the trajectory predictor described in Section 5.2. Each composition $c \in \mathcal{C}$ may require a different set of properties that we denote $\mathcal{P}_c$. For instance, a trajectory $x$ with $c_x = (s_{++b}, s_{+-h})$ will have $p_x = (t_0, t_1, x(t_0), x(t_1), \dot{x}(t_0), \dot{x}(t_1), \ddot{x}(t_1), h, t_{1/2})$, where each $(t_i, x(t_i))$ is a transition point, and $(h, t_{1/2})$ are the properties of the unbounded motif (see Figure 4). We denote all possible sets of properties as $\mathcal{P}$, where $\mathcal{P} = \bigcup_{c \in \mathcal{C}} \mathcal{P}_c$.

We are finally ready to provide a formal definition of the semantic representation of a trajectory $x \in C^2(\mathcal{T})$ and a forecasting model $F : \mathbb{R} \to C^2(\mathcal{T})$. Given this formal definition of semantic representation, we introduce our model, Semantic ODE, in the next section.

**Definition 1.** The *semantic representation of a trajectory* $x \in C^2(\mathcal{T})$ is a pair $(c_x, p_x)$, where $c_x \in \mathcal{C}$ is the composition of $x$ and $p_x \in \mathcal{P}_{c_x}$ is the set of properties as specified by $c_x$.

**Definition 2.** The *semantic representation* of $F : \mathbb{R} \to C^2(\mathcal{T})$ is a pair $(C_F, P_F) : \mathbb{R} \to \mathcal{C} \times \mathcal{P}$ defined as follows. $C_F : \mathbb{R} \to \mathcal{C}$ is called a *composition map* and it maps any initial condition $x_0 \in \mathbb{R}$ to a composition of the trajectory determined by its initial condition. Formally, $C_F(x_0) = c_{F(x_0)}$. $P_F : \mathbb{R} \to \mathcal{P}$ is called a *property map*, and it maps any initial condition $x_0 \in \mathbb{R}$ to the properties of the predicted trajectory $C_F(x_0)$. Formally, $P_F(x_0) = p_{F(x_0)}$.

## 5 SEMANTIC ODE

ODE discovery methods aim to discover the governing ODE $f$ and thus a forecasting model $F$ (defined by IVP), which is later analyzed to infer its semantic representation $(C_F, P_F)$ in a two-step modeling process. In this section, we introduce a novel forecasting model, called *Semantic ODE*, that allows for direct semantic modeling. As described in Section 3.3 our model $F$ consists of two submodels, $F_{\text{sem}}$ and $F_{\text{traj}}$ (where $F = F_{\text{traj}} \circ F_{\text{sem}}$). We can now define formally $F_{\text{sem}}$ as a function that maps an initial condition $x_0 \in \mathbb{R}$ to the semantic representation of a trajectory, i.e., $F_{\text{sem}} : \mathbb{R} \to \mathcal{C} \times \mathcal{P}$. $F_{\text{traj}}$ then takes this semantic representation and matches it a trajectory with such representation, i.e., $F_{\text{traj}} : \mathcal{C} \times \mathcal{P} \to C^2(\mathcal{T})$ such that if $x = F_{\text{traj}}(c, p)$ then $(c_x, p_x) = (c, p)$. This is visualized in Figure 4. Crucially, by definition, the semantic predictor is the semantic representation of $F$. Indeed, let $x = F(x_0) = F_{\text{traj}}(F_{\text{sem}}(x_0))$. Then by the definition of $F_{\text{traj}}$ above, $(C_F, P_F)(x_0) = (c_{F(x_0)}, p_{F(x_0)}) = (c_x, p_x) = F_{\text{sem}}(x_0)$. Unlike the two-step modeling approach, there is no need for post-hoc mathematical analysis. The semantic representation of the model can be directly inspected through the semantic predictor. In Section 5.1, we propose an implementation for the semantic predictor, and in Section 5.2, we describe the trajectory predictor.

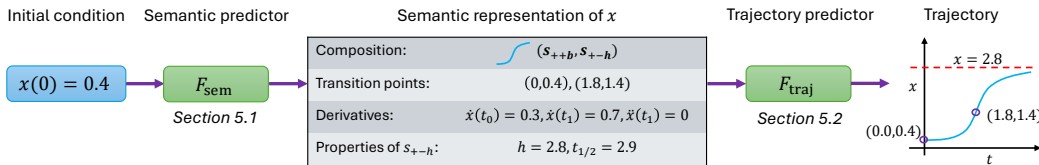

Figure 4: Semantic ODE ($F$) maps the initial condition to the semantic representation of the trajectory (using $F_{\text{sem}}$ and then uses it to predict the actual trajectory (through $F_{\text{traj}}$). Formally, $F = F_{\text{traj}} \circ F_{\text{sem}}$.

### 5.1 SEMANTIC PREDICTOR

Semantic predictor $F_{\text{sem}}$ consists of two models. One that predicts the composition denoted $F_{\text{com}}$ (corresponding to the composition map $C_F$), and one that predicts the properties, denoted $F_{\text{prop}}$ (corresponding to the property map $P_F$).

$F_{\text{com}}$ is a classification algorithm from $\mathbb{R}$ to $\mathcal{C}' \subset \mathcal{C}$, where $\mathcal{C}'$ is our chosen *composition library*. We model it as a partition of $\mathbb{R}$ into intervals (here called branches), each mapped to a different composition. The maximum number of branches $I \in \mathbb{N}$ is selected by the user. For instance, the logistic growth model example in Figure 1 would have 3 branches: $F_{\text{com}}(x_0) = (s_{++b}, s_{+-h})$ for $0 \le x_0 < 1.4$, $F_{\text{com}}(x_0) = (s_{+-h})$ for $1.4 \le x_0 \le 2.8$, and $F_{\text{com}}(x_0) = (s_{-+h})$ for $2.8 < x_0$.

As mentioned earlier, although $x_0 = 2.8$ should be a straight line, we approximate it with $s_{+-h}$.

$F_{\text{prop}}$ is modeled as a set of univariate functions. Each of them describes one of the coordinates of the transition point, the value of the first or second derivative, or the properties of the unbounded motif. These functions are different for different compositions. Continuing our logistic growth example, $F_{\text{prop}}$ is a piecewise function consisting of three composition-specific property sub-maps $F_{\text{prop}}^{(s_{++b}, s_{+-h})}$, $F_{\text{prop}}^{(s_{+-h})}$, $F_{\text{prop}}^{(s_{-+h})}$ that correspond to the three branches described above. Let us focus on $F_{\text{prop}}^{(s_{++b}, s_{+-h})}$ that describes the properties of $(s_{++b}, s_{+-h})$. The list of properties includes the coordinates of both transition points, first and second derivatives, and two properties of the unbounded motif ($s_{+-h}$). This is visualized in Figure 5.

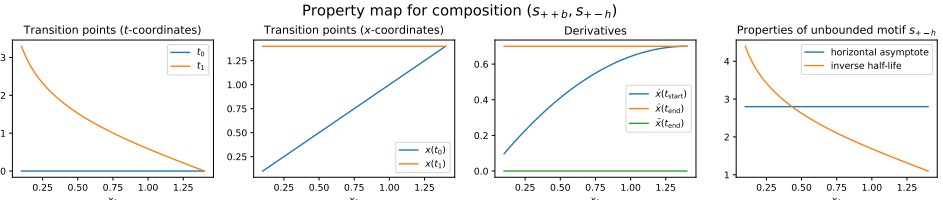

Figure 5: Property sub-map of the logistic growth model (for composition $(s_{++b}, s_{+-h})$) describes how various properties depend on the initial condition. See Figure 9 for a full semantic representation.

Training of $F_{\text{sem}}$ is performed in two steps. First, we train the composition map $F_{\text{com}}$. Then, the dataset is divided into separate subsets, each for a different composition—according to the composition map—and a separate property sub-map is trained on each of the subsets. A block diagram is presented in Figure 6, and the pseudocode of the training procedure can be found in Appendix C.

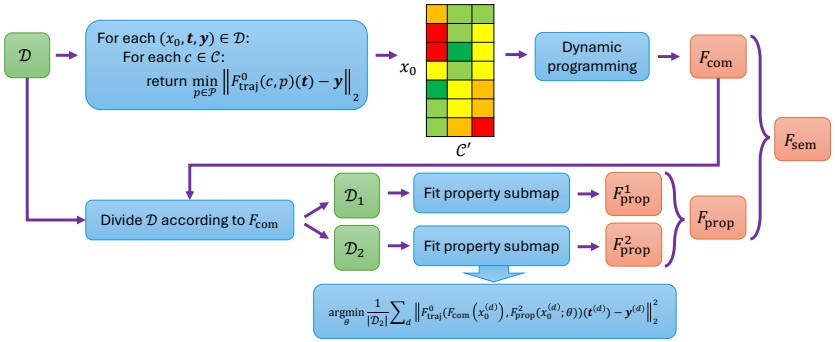

Figure 6: Block diagram showing main elements of training the semantic predictor.

## 5.2 TRAJECTORY PREDICTOR

The goal of the trajectory predictor ($F_{\text{traj}}$) is to map the semantic representation $(c, p)$ to a trajectory $x \in \mathcal{X}$ such that $(c_x, p_x) = (c, p)$. In that case, we say that $x$ *conforms* to $(c, p)$ and write it as $x \equiv (c, p)$. This mapping requires specifying the hypothesis space $\mathcal{X}$ for the predicted trajectories. As mentioned previously, we would also like each trajectory to be twice continuously differentiable, i.e., $\mathcal{X} \subset C^2$. We choose $\mathcal{X}$ to be a set of piecewise functions defined as

$$x(t) = \begin{cases} x_|(t) & t \leq t_{\text{end}} \\ {}_|x(t) & t > t_{\text{end}} \end{cases} \tag{2}$$

where $t_{\text{end}}$ is the last transition point (before the unbounded motif), $x_| \in \mathcal{X}_|$ is called the bounded part of $x$, and ${}_|x \in {}_|\mathcal{X}$ is called the unbounded part of $x$. Finding $x_|$ and ${}_|x$ is done separately and we discuss it respectively in Sections 5.2.1 and 5.2.2.

### 5.2.1 BOUNDED PART OF THE TRAJECTORY

In this section, we define $\mathcal{X}_|$ and describe how we can find the bounded part of the trajectory ($x_|$) given a semantic representation $(c, p)$ without the unbounded motif and its properties, denoted $(c_|, p_|)$.

**Cubic splines** We decide to define $\mathcal{X}_|$ as a set of cubic splines. They are piecewise functions where each piece is defined as a cubic polynomial. The places where two cubics are joined are called *knots*. Cubic splines require that the first and second derivatives at the knots be the same for neighboring cubics so that the cubic spline is guaranteed to be twice continuously differentiable. Cubic splines are promising because they are flexible, and for a fixed set of knots, the equations for their values and derivatives are linear in their parameters. We come up with two different ways of finding a cubic spline that conforms to a particular semantic representation that leads us to develop two trajectory predictors: $F_{\text{traj}}^0 : \mathcal{C} \times \mathcal{P} \to C^0$ and $F_{\text{traj}}^2 : \mathcal{C} \times \mathcal{P} \to C^2$. We use $F_{\text{traj}}^0$ during training of $F_{\text{sem}}$ as it is fast and differentiable, but the found trajectory may not be in $C^2$ (only continuous). At inference, we use $F_{\text{traj}}^2$ that is slower and not differentiable but ensures that the trajectory is in $C^2$. We describe both approaches briefly below. More details are available in Appendix D.1

$C^0$ **trajectory predictor** $F_{\text{traj}}^0$ describes each motif as a single cubic. Each cubic is found by solving a set of four linear equations. Two are for the positions of the transition points, and two are for the derivatives, one at each transition point. As each transition point (apart from $t_0$) is either a local extremum or an inflection point, we know that either the first or the second derivative vanishes. The first derivative at $t_0$ is specified directly in $p_|$. During training, we also make sure that this derivative is always in the range described in Table 9. We prove why this is sufficient to guarantee that the predicted trajectory conforms to $(c_|, p_|)$ in Appendix D.1.2.

$C^2$ **trajectory predictor** $F_{\text{traj}}^2$ describes each motif as two cubics with an additional knot between every two transition points. Given $K$ cubics (and $K+1$ knots), the traditional way to ensure that $x_|$ is in $C^2$ is to set up $3(K-1)$ constraints that match the values and both derivatives at the knots. Instead, we propose to fit the second derivative of the cubic spline ($\ddot{x}_|$), a piecewise linear function, and then integrate it twice to get the desired cubic spline. As it is continuous, we can describe it solely by its values at the knots ($v_k$ at knot $t_k$). Together with the additional two parameters for the integration constants, we not only reduce the number of parameters to $K+3$ (while still guaranteeing the trajectory is in $C^2$) but, more importantly, we can control exactly the value of the second derivative. Importantly, as shown in Appendix D.1, we can impose $v_k > 0$ or $v_k < 0$ accordingly. We set some $v_k$ to make sure that the first and second derivatives at the transition points are correct. Then, we optimize the additional knots and some of $v_k$ to minimize the error between the predicted and target $x$-coordinates of the transition points (with an additional loss term for the signs of the first derivatives at the knots). We minimize this objective using L-BFGS-B (Liu & Nocedal, 1989) and Powell's method (Powell, 1964) until we find a solution where the error on the transition points is smaller than a user-defined threshold. If it does not succeed, then we default to $F_{\text{traj}}^0$.

### 5.2.2 Unbounded part of the trajectory

In this section, we define $_|\mathcal{X}$ and describe how we can find the unbounded part of the trajectory given an unbounded motif and its properties, as well as the coordinates of the last transition point and both derivatives at this point. We need them to ensure that $x$ is twice differentiable at $t_{\text{end}}$. First, we need to choose which properties we are interested in. They should describe the shape (and long-term behavior) of the unbounded motif in sufficient detail such that we do not need to see the underlying equation to visualize it. For instance, for motifs with horizontal asymptotes ($s_{+-h}$ and $s_{-+h}$), we choose these to be $h$ and $t_{1/2}$ where $x = h$ is the horizontal asymptote and $t_{1/2}$ is the time where the trajectory is in the middle between the last transition point and the $h$, i.e., $x(t_{1/2}) = (x(t_{\text{end}}) + h)/2$. In exponential decay, $t_{1/2} - t_{\text{end}}$ would correspond to "half-life". Then, we need to come up with a parametrization that would both guarantee the shape of the trajectory and allow us to impose any possible properties (e.g., $h$ in $s_{-+h}$ needs to satisfy $h < x(t_{\text{end}})$). We parametrize $s_{-+h}$ as $_|x(t) = 2(x(t_{\text{end}}) - h)/(1 + e^{g(t)}) + h$ where $g$ is an appropriately defined cubic spline. We show how we can find such $g$ and why it has the desired properties, as well as properties and parameterizations for other motifs, in Appendix D.2. Importantly, it does not matter how complicated these parameterizations are, as they are not used by humans to understand the model. All information is directly available in $F_{\text{sem}}$.

## 6 Semantic ODE in action

In this section, we want to illustrate the usability of Semantic ODEs and highlight their advantages: semantic inductive biases (Section 6.1), comprehensibility (Section 6.2), editing (Section 6.3) and flexibility and robustness to noise (Section 6.4). To demonstrate the first three advantages, we present a case study of finding a pharmacokinetic model from a dataset of observed drug concentration curves. Such models are essential for drug development and later clinical practice. Details about the dataset can be found in Appendix E.1. We will contrast our approach with one of the most popular ODE discovery algorithms, SINDy (Brunton et al., 2016b). However, many of the observations will apply to other algorithms as well.

### 6.1 Semantic inductive biases

In Semantic ODEs, a user can specify inductive biases about the semantic representation of the model. This is in contrast to the *syntactic inductive biases*, available in ODE discovery algorithms. Semantic inductive biases can be more meaningful and intuitive for users than syntactic ones as the relationship between the syntactic and semantic representation may be non-trivial. The role of syntactic biases is, predominantly, to ensure that the equations can be analyzed by humans. They are not designed to accommodate prior knowledge about the system's behavior. Examples of inductive biases in SINDy and Semantic ODE are shown in Table 1.

The drug concentration curve describes the drug plasma concentration after administration as a function of time. Without any additional doses, we would expect the concentration to decay to 0

Table 1: Examples of inductive biases in SINDy and Semantic ODE.

| Syntactic inductive biases in SINDy | Semantic inductive biases in Semantic ODE |
|---|---|
| **Autonomous system**: whether the governing ODE system is time-invariant. **Library of functions:** whether to include, e.g., polynomials, trigonometric terms, exponential/logarithmic terms, cross-terms. **Sparsity:** the number of terms, strength of penalty terms such as L1 or L2 norms. | **Library compositions:** The maximum number of motifs, the starting motif, the type of asymptotic behavior. **The complexity of the composition map:** how many different compositions it should predict. **Complexity of the property maps:** how many trend changes the property maps could have. |

as $t \to \infty$. This is a semantic inductive bias that can be easily put into Semantic ODE. We can enforce the last motif to always be $s_{-+h}$ (decreasing, convex function with a horizontal asymptote) by removing all biologically impossible compositions from the library $\mathcal{C}'$. Designing syntactic inductive biases based on the prior knowledge is more challenging. SINDy assumes $\dot{x} = \sum_{i=1}^{n} \alpha_i g_i$, i.e., $f$ is a linear combination of pre-specified functions. As choosing which terms should and should not appear in the equation is far from obvious, we choose a general library containing polynomial terms, $\exp$, and trigonometric terms (see Appendix E.2 for details). We also consider different sparsity levels, from 1 to 15 non-zero terms.

## 6.2 COMPREHENSIBILITY

We have fitted Semantic ODE with the inductive bias described earlier and versions of SINDy with different sparsity constraints. The results can be seen in Table 2.

Table 2: Results of fitting Semantic ODE and SINDy to the pharmacokinetic dataset.

| Model | Syntactic biases | Semantic biases | Syntactic representation | Semantic representation | In-domain $(t \leq 1)$ | Out-domain $(t > 1)$ |
|---|---|---|---|---|---|---|
| SINDy | $\dot{x} = \sum_{i=1}^{n} \alpha_i g_i, n \leq 1$ | NA | $\dot{x}(t) = -3.06x(t)t$ | NA | $0.222_{(0.041)}$ | $0.024_{(0.005)}$ |
| SINDy | $\dot{x} = \sum_{i=1}^{n} \alpha_i g_i, n \leq 2$ | NA | $\dot{x}(t) = 5.56-43.10x(t)t$ | NA | $0.112_{(0.027)}$ | $0.054_{(0.010)}$ |
| SINDy | $\dot{x} = \sum_{i=1}^{n} \alpha_i g_i, n \leq 5$ | NA | Equation (7) | NA | $0.101_{(0.023)}$ | $16.850_{(0.021)}$ |
| SINDy | $\dot{x} = \sum_{i=1}^{n} \alpha_i g_i, n \leq 10$ | NA | Equation (8) | NA | $0.029_{(0.005)}$ | $18.686_{(0.003)}$ |
| SINDy | $\dot{x} = \sum_{i=1}^{n} \alpha_i g_i, n \leq 15$ | NA | Equation (9) | NA | $0.020_{(0.004)}$ | $77.577_{(1.249)}$ |
| Semantic ODE | NA | $\|c_x\| \leq 4$, $c_x$ ends with $s_{-+h}$ | NA | Figure 7 | $\mathbf{0.016}_{(.004)}$ | $0.033_{(.006)}$ |
| Semantic ODE | NA | $c_x : (s_{+-b}, s_{--b}, s_{-+h}), h = 0$ | NA | Figure 8 | $0.018_{(.003)}$ | $\mathbf{0.015}_{(.002)}$ |

We can see that Semantic ODE better fits the dataset than even the longest equations found by SINDy. Compact equations, e.g., $\dot{x}(t) = 5.56 - 43.10x(t)t$ have poor performance. To improve it, we need to allow for much more complicated equations, such as Equation (9) in Appendix B. Such equations are very hard to analyze and, as a result, may be no more interpretable than a black box model. They are also more prone to over-fitting, as demonstrated by large out-domain error. More importantly, Semantic ODE can be directly understood by looking at its semantic representation in Figure 7.

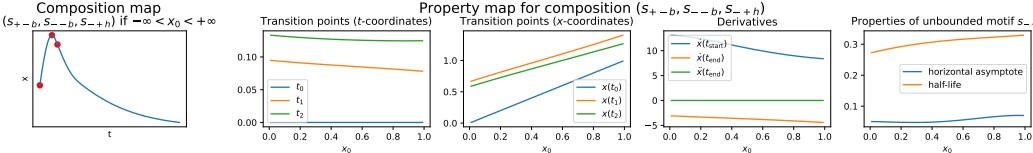

Figure 7: Semantic representation of Semantic ODE fitted to the pharmacological dataset.

The left side of Figure 7 tells us that for all initial conditions, the shape of the predicted trajectory is going to be the same, namely $(s_{+-b}, s_{--b}, s_{-+h})$. As the composition map has only one branch, we have only one property map, and it is visualized on the right of the composition map. It consists of four subplots. Going from left to right, we have the $t$-coordinates of the transition points, $x$-coordinates of the transition points, values of the derivatives at the boundary transition points, and the properties of the unbounded motif. In our case, the unbounded motif is a convex, decreasing function with a horizontal asymptote, and it is described by the value of the horizontal asymptote and its "half-life", which is the $t$-coordinate of the point where the value of $x$ is in the middle between the last transition point and the asymptote. By looking at this representation, we can readily see how the trajectory changes with respect to the initial condition. We see that the $t$-coordinates of the transition points remain fairly constant, whereas $x$-coordinates increase linearly. In particular, we can see how the maximum of the trajectory $(x(t_1))$ increases linearly from $0.6$ up to $1.3$. Arriving at similar observations about the discovered ODEs requires significant time and expertise. In particular,

we do not know what the composition map of the discovered ODEs looks like. Thus, we cannot be sure whether the model behaves correctly.

## 6.3 EDITING

Often, the fitted model does not satisfy all our requirements. A very important requirement in pharmacology would be to make sure that the model is biologically possible. In particular, the predicted concentration should decay to 0 as $t \to \infty$. The horizontal asymptote of our model is close to 0 but not exactly. That is why the extrapolation error for $t > 1$ is substantially higher than for $t \leq 1$. Fortunately, we can edit the property map directly. We can impose the value of the horizontal asymptote to be equal to 0 and retrain the model. The new property map can be seen in Figure 8. Importantly, as shown in the last row of Table 2, the extrapolation error dropped to levels comparable with the in-domain error. In two-step modeling, it may be challenging to verify and impose such requirements for the predicted equations, especially the longer ones. As a result, we may end up with a model that does not obey crucial domain-specific rules.

## 6.4 FLEXIBILITY AND ROBUSTNESS TO NOISE

As Semantic ODE does not assume that the trajectory is governed by a closed-form ODE, it may fit systems that are not described by those. In particular, we show in Table 3 how, beyond standard ODEs (logistic growth model), it can fit systems governed by a general differential equation $\dot{x}(t) = f(x(t), t)$, where $f$ does not have a compact closed-form representation, a multidimensional ODE where only one dimension is observed (pharmacokinetic model), delay differential equation (Mackey & Glass, 1977), and an integro-differential equation (integro-DE). We compare our approach to SINDy (Brunton et al., 2016b) and WSINDy (Reinbold et al., 2020; Messenger & Bortz, 2021a) as implemented in `PySINDy` library (de Silva et al., 2020; Kaptanoglu et al., 2022). We consider variants constrained to 5 terms (in a linear combination) to ensure compactness (SINDy-5, WSINDy-5) and where the number of terms is fine-tuned and may no longer be compact (SINDy, WSINDy). We also include a standard symbolic regression method, PySR (Cranmer, 2020), adapted for ODE discovery and constrained to 20 *symbols*. We also compare with three black box approaches: Neural ODE (Chen et al., 2018), Neural Laplace (Holt et al., 2022), and DeepONet (Lu et al., 2020). Semantic ODE is more or equally robust to noise and performs better than the methods constrained to compact equations in most cases. Moreover, its performance could possibly be further improved by incorporating semantic inductive biases and model editing as discussed earlier. Additional experiments can be found in Appendix B. Details on experiments are available in Appendix E.

Table 3: Comparison of Average RMSE obtained by different models. Average performance over 5 random seeds and data splits is shown with standard deviations in the brackets.

| Method | Logistic Growth | | General ODE | | Pharmacokinetic model | | Mackey-Glass (DDE) | | Integro-DE | |
|---|---|---|---|---|---|---|---|---|---|---|
| | low noise | high noise | low noise | high noise | low noise | high noise | low noise | high noise | low noise | high noise |
| SINDy-5 | $0.012_{(.002)}$ | $0.222_{(.004)}$ | $0.053_{(.012)}$ | $0.103_{(.010)}$ | $0.093_{(.004)}$ | $0.230_{(.014)}$ | $0.238_{(.023)}$ | $0.248_{(.025)}$ | $0.431_{(.051)}$ | $0.268_{(.019)}$ |
| WSINDy-5 | $\mathbf{0.010}_{(.000)}$ | $0.222_{(.009)}$ | $0.066_{(.009)}$ | $0.102_{(.008)}$ | $0.211_{(.009)}$ | $0.415_{(.299)}$ | $0.272_{(.032)}$ | $0.300_{(.061)}$ | $0.160_{(.066)}$ | $0.452_{(.365)}$ |
| PySR-20 | $0.012_{(.002)}$ | $0.224_{(.007)}$ | $0.078_{(.029)}$ | $0.119_{(.029)}$ | $0.053_{(.015)}$ | $0.242_{(.039)}$ | $0.261_{(.021)}$ | $0.288_{(.031)}$ | $0.027_{(.011)}$ | $0.393_{(.144)}$ |
| SINDy | $0.012_{(.001)}$ | $0.218_{(.011)}$ | $0.068_{(.013)}$ | $0.115_{(.012)}$ | $0.020_{(.001)}$ | $0.209_{(.010)}$ | $0.252_{(.026)}$ | $0.257_{(.028)}$ | $0.318_{(.172)}$ | $0.248_{(.016)}$ |
| WSINDy | $\mathbf{0.010}_{(.001)}$ | $0.217_{(.016)}$ | $0.062_{(.009)}$ | $0.112_{(.009)}$ | $0.038_{(.006)}$ | $0.219_{(.016)}$ | $0.200_{(.035)}$ | $0.207_{(.031)}$ | $0.152_{(.086)}$ | $0.300_{(.082)}$ |
| Neural ODE | $0.023_{(.004)}$ | $\mathbf{0.197}_{(.005)}$ | $0.029_{(.005)}$ | $0.075_{(.006)}$ | $0.036_{(.008)}$ | $0.203_{(.007)}$ | $0.177_{(.010)}$ | $0.194_{(.010)}$ | $0.073_{(.007)}$ | $0.215_{(.009)}$ |
| Neural Laplace | $0.126_{(.036)}$ | $0.230_{(.017)}$ | $0.108_{(.030)}$ | $0.138_{(.023)}$ | $0.100_{(.022)}$ | $0.229_{(.013)}$ | $0.057_{(.006)}$ | $0.094_{(.009)}$ | $0.075_{(.044)}$ | $0.249_{(.014)}$ |
| DeepONet | $0.184_{(.040)}$ | $0.306_{(.023)}$ | $0.160_{(.033)}$ | $0.195_{(.027)}$ | $0.058_{(.010)}$ | $0.212_{(.005)}$ | $0.107_{(.014)}$ | $0.132_{(.012)}$ | $0.100_{(.015)}$ | $0.230_{(.014)}$ |
| Semantic ODE | $0.015_{(.005)}$ | $0.198_{(.007)}$ | $\mathbf{0.016}_{(.003)}$ | $\mathbf{0.068}_{(.002)}$ | $\mathbf{0.015}_{(.001)}$ | $\mathbf{0.197}_{(.006)}$ | $\mathbf{0.037}_{(.003)}$ | $\mathbf{0.077}_{(.004)}$ | $\mathbf{0.025}_{(.003)}$ | $\mathbf{0.204}_{(.007)}$ |

## 7 DISCUSSION

**Limitations and open challenges**  As Semantic ODE is the first model that allows for direct semantic modeling, we focused solely on 1-dimensional systems and we describe a possible roadmap to higher-dimensional settings in Appendix G.2. The current definition of semantic representation assumes that the trajectory has a finite composition, i.e., it cannot have an oscillatory behavior (like $\sin$). Of course, we could fit a periodic function on any bounded interval, but it would fail to predict the oscillatory behavior outside of it. We discuss more limitations in Appendix G.1.

**Direct semantic modeling as a new way for modeling dynamical systems**  In this work we outlined the main principles of direct semantic modeling, discussed its advantages, and illustrated how it can be achieved in practice through Semantic ODE. We believe this approach can transform the way dynamical systems are modeled by shifting the focus from the equations to the system's behavior, making the models not only more understandable but also more flexible than other techniques.

**Ethics statement**   In this paper, we introduce a novel approach for enhancing the comprehensibility of a dynamical system's behavior through direct semantic modeling, with a practical implementation called Semantic ODE. Improved transparency of machine learning models is crucial for tasks such as model debugging, ensuring compliance with domain-specific constraints, and addressing potential harmful biases. However, such techniques, if misused, can lead to a false sense of security in model decisions or be leveraged merely for superficial regulatory compliance. As our approach is applicable to high-stakes domains like medicine and pharmacology, it is vital to conduct a thorough evaluation before deploying the model in such contexts. This evaluation must ensure that the model's behavior aligns with ethical considerations and does not support decisions that could negatively impact individuals' health and well-being.

**Reproducibility statement**   All mathematical definitions are provided in Sections 4 and 5 and Appendix A.2. The proofs are provided in Appendix D. The implementation, including a block diagram and pseudocode, is discussed in Section 5 and Appendices C and D. The experimental details are discussed in Section 6 and Appendix E. All experimental code is available at `https://github.com/krzysztof-kacprzyk/SemanticODE`.

**Acknowledgments**   This work was supported by Roche. We would like to thank Max Ruiz Luyten, Harry Amad, Julianna Piskorz, Andrew Rashbass, and anonymous reviewers for their useful comments and feedback on earlier versions of this work.

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

TABLE OF SUPPLEMENTARY MATERIALS

# A    NOTATION AND DEFINITIONS

## A.1    NOTATION

Symbols used throughout this work can be found in Tables 4 and 5.

## A.2    DEFINITIONS

In this section, we provide formal definition of some of the terms introduced in the main text.

From the work by Kacprzyk et al. (2024b).

**Definition 3.** Let $\mathcal{I}$ be a set of intervals on $\mathbb{R}$ and let $\mathcal{F}$ be the set of interval functions, i.e., real functions defined on intervals. A motif $s$ is a binary relation between the set of interval functions $\mathcal{F}$ and the set of intervals $\mathcal{I}$ (i.e., $s \subset \mathcal{F} \times \mathcal{I}$). We denote $(\phi, i) \in s$ as $\phi|i \sim s$ and read it as "$\phi$ on $i$ has a motif $s$". Each motif needs to be

- *well-defined*, i.e., for any $\phi \in \mathcal{F}$, and any $i \in \mathcal{I}$,

$$\phi|i \sim s \implies i \subset \mathrm{dom}(\phi) \tag{3}$$

- *translation-invariant*, i.e., for any $i \in \mathcal{I}$, and any $\phi \in \mathcal{F}$,

$$\phi|i \sim s \iff \phi \circ (t - q)|(i + q) \sim s \; \forall q \in \mathbb{R} \tag{4}$$

In the next definition, we formally define the motifs we use to define semantic representation.

**Definition 4.** Let $x \in C^2$ and let $r_1, r_2 \in \mathbb{R}$ such that $r_1 < r_2$.

- $x|[r_1, r_2] \sim s_{++b}$ if $\forall t \in (r_1, r_2) \; \dot{x}(t) > 0, \; \ddot{x}(t) > 0$

- $x|[r_1, r_2] \sim s_{+-b}$ if $\forall t \in (r_1, r_2) \; \dot{x}(t) > 0, \; \ddot{x}(t) < 0$

- $x|[r_1, r_2] \sim s_{-+b}$ if $\forall t \in (r_1, r_2) \; \dot{x}(t) < 0, \; \ddot{x}(t) > 0$

- $x|[r_1, r_2] \sim s_{--b}$ if $\forall t \in (r_1, r_2) \; \dot{x}(t) < 0, \; \ddot{x}(t) < 0$

- $x|[r_1, +\infty) \sim s_{++u}$ if $\forall t \in (r_1, +\infty) \; \dot{x}(t) > 0, \; \ddot{x}(t) > 0$

- $x|[r_1 + \infty) \sim s_{+-u}$ if $\forall t \in (r_1 + \infty) \; \dot{x}(t) > 0, \; \ddot{x}(t) < 0$ and $\lim_{t \to +\infty} x(t) = +\infty$

- $x|[r_1, +\infty) \sim s_{-+u}$ if $\forall t \in (r_1, +\infty) \; \dot{x}(t) < 0, \; \ddot{x}(t) > 0$ and $\lim_{t \to +\infty} x(t) = -\infty$

- $x|[r_1, +\infty) \sim s_{--u}$ if $\forall t \in (r_1, +\infty) \; \dot{x}(t) < 0, \; \ddot{x}(t) < 0$

- $x|[r_1 + \infty) \sim s_{+-h}$ if $\forall t \in (r_1, +\infty) \; \dot{x}(t) > 0, \; \ddot{x}(t) < 0$ and $\lim_{t \to +\infty} x(t) \in \mathbb{R}$

- $x|[r_1 + \infty) \sim s_{-+h}$ if $\forall t \in (r_1, +\infty) \; \dot{x}(t) < 0, \; \ddot{x}(t) > 0$ and $\lim_{t \to +\infty} x(t) \in \mathbb{R}$

**Definition 5.** Let $u_|, v_|$ be bounded trajectory parts on $(t_0, t_{\mathrm{end}})$ and assume $v_| \equiv (c_|, p_|)$, where $(c_|, p_|)$ is the bounded part of the semantic representation $(c, p)$. We say that $u_|$ *seemingly matches* $(c_|, p_|)$, and write it as $u_| \sim (c_|, p_|)$, if for any transition point $t$ in $(c_|, p_|)$

Table 4: Notation used throughout the paper (Part 1).

| Symbol | Meaning |
|---|---|
| $M$ | The number of dimensions in a dynamical system, $M \in \mathbb{N}$ |
| $m$ | Used for indexing dimensions, $m \in \{1, \ldots, M\}$ |
| $t_0$ | Start of the trajectory, $t_0 \in \mathbb{R}$ |
| $\mathcal{T}$ | Time domain, subset of $\mathbb{R}$, $(t_0, +\infty)$ |
| $\boldsymbol{x}$ | An $M$-dimensional trajectory, $\boldsymbol{x} : \mathcal{T} \to \mathbb{R}^M$ |
| $x$ | A 1-dimensional trajectory, $x : \mathcal{T} \to \mathbb{R}$ |
| $\dot{\boldsymbol{x}}$ or $\dot{x}$ | Derivative of $\boldsymbol{x}$ or $x$ with respect to time $t$ |
| $D$ | The number of samples, $D \in \mathbb{N}$ |
| $d$ | Used for indexing samples, $d \in \{1, \ldots, D\}$ |
| $N_d$ | The number of measurements of sample $d$, $N_d \in \mathbb{N}$ |
| $t_n^{(d)}$ | The time of the $n^{\text{th}}$ measurement of sample $d$, $t_n^{(d)} \in \mathcal{T}$ |
| $\boldsymbol{y}_n^{(d)}$ | The $n^{\text{th}}$ measurement of sample $d$ (taken at time $t_n^{(d)}$), $\boldsymbol{y}_n^{(d)} \in \mathbb{R}^M$ |
| $\boldsymbol{x}_0$ | The initial condition, the value of $\boldsymbol{x}$ at $t_0$, $\boldsymbol{x}_0 \in \mathbb{R}^M$ |
| $\boldsymbol{f}$ | A system of $M$ ODEs, $\boldsymbol{f} : \mathbb{R}^{M+1} \to \mathbb{R}^M$ |
| $f$ | A single ODE, $f : \mathbb{R}^2 \to \mathbb{R}$ |
| $C^0(\mathcal{T})$ | The set of continuous functions on $\mathcal{T}$ |
| $C^2(\mathcal{T})$ | The set of twice continuously differentiable functions on $\mathcal{T}$ |
| $\boldsymbol{F}$ | A forecasting model predicting an $M$-dimensional trajectory, $F_i : \mathbb{R} \to C^2$ |
| $F$ | A forecasting model predicting a 1-dimensional trajectory, $F : \mathbb{R} \to C^2$ |
| $F_{\text{sem}}$ | A semantic predictor, predicts a semantic representation of trajectory from the initial condition |
| $F_{\text{traj}}$ | A trajectory predictor, predict a trajectory from its semantic representation |
| $c_x$ | A composition of trajectory $x$ |
| $p_x$ | A set of properties of trajectory $x$ |
| $t_{\text{end}}$ | The last transition point, $t_{\text{end}} \in \mathcal{T}$ |
| $\mathcal{C}$ | The set of all possible compositions |
| $\mathcal{P}_c$ | The set of all possible properties for composition $c$ |
| $\mathcal{P}$ | The set of all possible properties for all compositions |
| $s_{\pm\pm*}$ | Motifs, formally defined in Definition 4 |
| $(c_x, p_x)$ | Semantic representation of $x$ |
| $(C_F, P_F)$ | Semantic representation of $F$ |
| $\circ$ | Function composition |
| $h$ | Value of the horizontal asymptote in motifs $s_{-+h}$ and $_{+-h}$ |
| $t_{1/2}$ | "half-life" property of motifs $s_{-+h}$ and $_{+-h}$ |
| $F_{\text{prop}}$ | A property map, $F_{\text{prop}} : \mathbb{R} \to \mathcal{P}$ |
| $F_{\text{com}}$ | A composition map, $F_{\text{com}} : \mathbb{R} \to \mathcal{C}$ |
| $F_{\text{prop}}^{(c)}$ | A property sub-map, $F_{\text{prop}} : \mathbb{R} \to \mathcal{P}_c$ |
| $x \equiv (c, p)$ | $x \in C^2$ conforms to $(c, p) \in \mathcal{C} \times \mathcal{P}$, $(c, p) = (c_x, p_x)$ |
| $x \sim (c, p)$ | $x \in C^2$ seemingly matches $(c, p) \in \mathcal{C} \times \mathcal{P}$, Definition 5 |

- $u_|(t) = v_|(t)$,

- $\dot{u}_|(t) = \dot{v}_|(t)$ if $t \in \{t_0, t_{\text{end}}\}$ or $t$ is a local extremum,

- $\ddot{u}_|(t) = \ddot{v}_|(t)$ if $t = t_{\text{end}}$ or $t$ is an inflection point.

We also say $u_|$ *seemingly matches* $v_|$ and write it $u_| \sim v_|$.

## B  ADDITIONAL RESULTS

### B.1  EQUATIONS DISCOVERED BY SINDY

Below are the equations discovered by different variants of SINDy in Section 6.2.

$$\dot{x}(t) = -3.06x(t)t \tag{5}$$

Table 5: Notation used throughout the paper (Part 2).

| Symbol | Meaning |
| --- | --- |
| $\mathcal{X}$ | set of trajectories |
| $x_{\vert}$ | Bounded part of $x$, $x_{\vert} : [t_0, t_{\text{end}}] \to \mathbb{R}$ |
| $_{\vert}x$ | Unbounded part of $x$, $_{\vert}x : [t_{\text{end}}, +\infty)$ |
| $(c_{\vert}, p_{\vert})$ | Bounded part of $(c, p)$, i.e., $(c, p)$ without the unbounded motif and its properties |
| $(_{\vert}c, _{\vert}p)$ | Unbounded part of $(c, p)$, i.e., the unbounded motif, its properties, the last transition point, and derivatives at it |
| $F_{\text{traj}}^0$ | $C^0$ trajectory predictor, $F_{\text{traj}}^0 : \mathcal{C} \times \mathcal{P} \to C^0$ |
| $F_{\text{traj}}^2$ | $C^2$ trajectory predictor, $F_{\text{traj}}^2 : \mathcal{C} \times \mathcal{P} \to C^2$ |
| $\mathcal{C}'$ | Composition library, $\mathcal{C}' \subset \mathcal{C}$ |
| $I$ | The maximum number of intervals/branches in the composition map $F_{\text{com}}$, $I \in \mathbb{N}$ |

$$\dot{x}(t) = 5.56 - 43.10 x(t) t \tag{6}$$

$$\dot{x}(t) = -27.56t + 14.83 \cos(2x(t)) - 12.36 \sin(2t) - 13.79 \cos(3t) + 8.78 \exp(x(t)) \tag{7}$$

$$\begin{aligned} \dot{x}(t) = {} & 39828.60 - 50679.07t - 1.74x(t)t + 54549.25 \sin(t) - 54591.64 \cos(t) + 1.58 \sin(2x(t)) \\ & + 16141.41 \cos(2t) + 1.63 \cos(3x(t)) - 1362.80 \sin(3t) - 1366.28 \cos(3t) \end{aligned} \tag{8}$$

$$\begin{aligned} \dot{x}(t) = {} & 8347995.48 + 3372040.53t + 3.75x(t)t - 3350455.09t^2 - 36.16 \cos(x(t)) \\ & - 4904904.06 \sin(t) - 8903838.66 \cos(t) + 12.19 \cos(2x(t)) \\ & + 889658.07 \sin(2t) + 560285.98 \cos(2t) - 0.29 \sin(3x(t)) \\ & - 2.81 \cos(3x(t)) - 82220.29 \sin(3t) - 4394.42 \cos(3t) - 5.23 \exp(x(t)) \end{aligned} \tag{9}$$

### B.2 SEMANTIC REPRESENTATION AFTER EDITING

Figure 8 shows semantic representation of Semantic ODE after editing performed in Section 6.3.

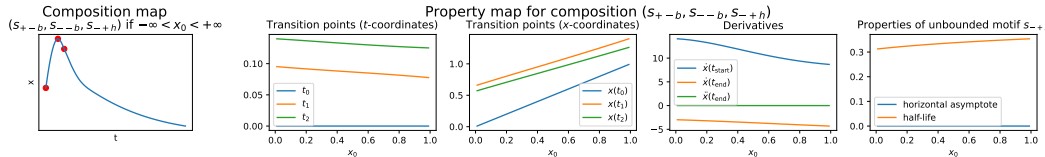

Figure 8: Property map of the Semantic ODE after imposing the horizontal asymptote to be 0.

### B.3 SEMANTIC REPRESENTATION OF THE LOGISTIC GROWTH MODEL

Figure 9 demonstrates a semantic representation of the logistic growth model $\dot{x}(t) = x(t)(1 - \frac{x(t)}{2.8})$.

### B.4 INSIDE OF FITTING A COMPOSITION MAP

Figure 10 shows the logarithm of the loss for each sample and for each considered composition while fitting a composition map.

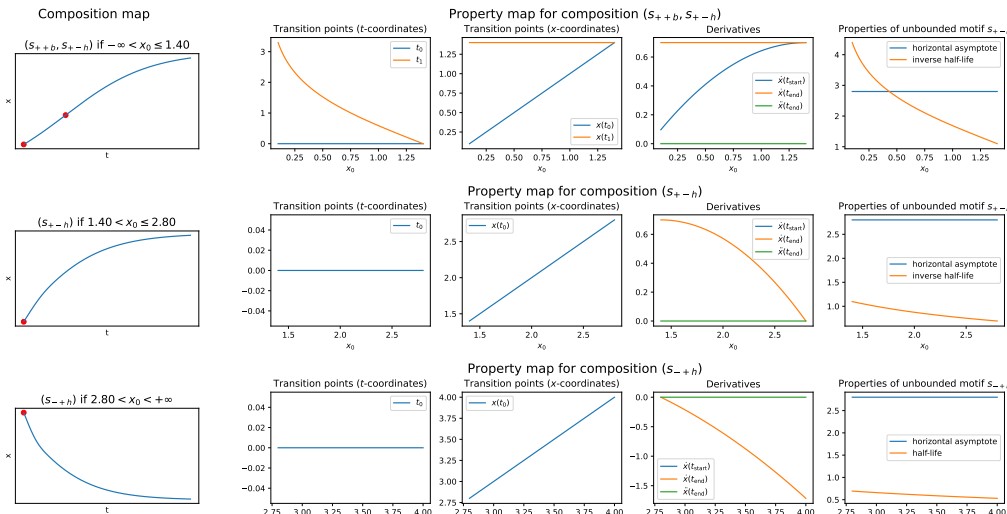

Figure 9: Semantic representation of the logistic growth model.

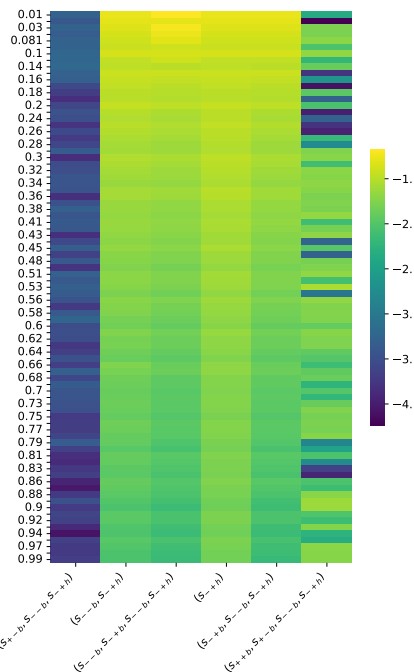

Figure 10: Logarithm of loss for each sample for each considered composition while fitting a composition map to the pharmacokinetic dataset in Section 6.2.

## B.5 EXTENSION TO MULTIPLE DIMENSION: PROOF OF CONCEPT

In this section, we show a proof of concept of how semantic modeling can be extended to multiple dimensions (as was described in Appendix G.2).

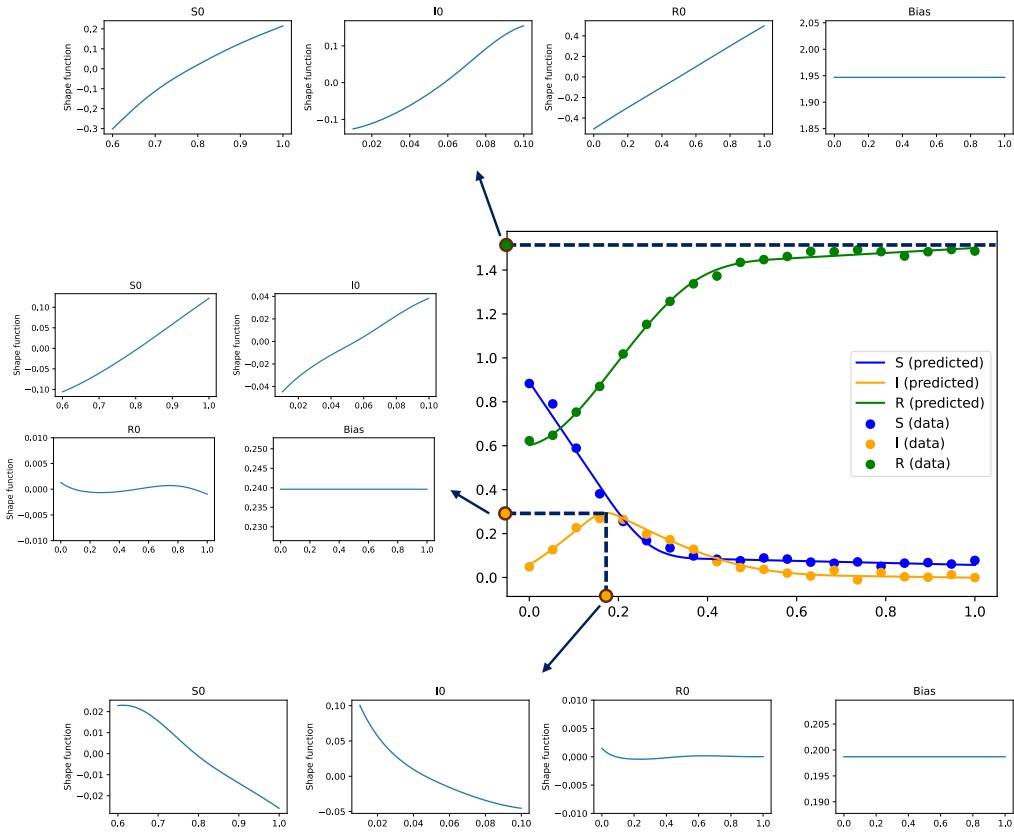

Figure 11: Semantic ODE model extended to multiple trajectories and fitted to data governed by the SIR epidemiological model. We show three generalized additive models describing the horizontal asymptote of $R$ (top), the maximum of $I$ (middle), and the time when $I$ is at its maximum (bottom).

We implemented the property maps as described in Appendix G.2 (each property described as a generalized additive model) and fitted data following an SIR epidemiological model for different initial conditions. To simplify the problem we specify the composition map (we only train property maps). We assume $S$ follows $(s_{--c}, s_{-+h})$, $I$ follows $(s_{++c}, s_{+-c}, s_{--c}, s_{-+h})$, and $R$ follows $(s_{++c}, s_{+-h})$. The predicted trajectories and some of the property maps can be seen in Figure 11. The average RMSE on the test dataset is $0.019$ for $S$ trajectory, $0.011$ for $I$ and $0.014$ for $R$. Note that the irreducible error on this dataset (caused by the added Gaussian noise) is $0.01$. The shown property maps let us draw the following insights about the model:

- The time when $I$ is at its maximum $t_{\max}(I)$ is on average just below $0.2$. $I_0$ has a relatively large impact on $t_{\max}(I)$ by increasing it by $0.1$ for very low $I_0$ or decreasing it by $0.05$ for very high $I_0$. The larger the $I_0$ the faster the maximum is achieved.
- $S_0$ also has a negative impact on $t_{\max}(I)$ but it is much smaller ($\pm 0.02$).
- The maximum of $I$ (denoted $I_{\max}$) increases linearly with both $S_0$ and $I_0$. This time $S_0$ has slightly bigger impact ($\pm 0.1$) compared to $I_0$ ($\pm 0.04$)
- In both $t_{\max}(I)$ and $I_{\max}$, the impact of $R_0$ is insignificant.
- The horizontal asymptote of $R$ increases linearly with all three initial conditions. In particular, the shape function associated with $R_0$ has unit slope as expected.

Interesting advantage of our approach is that even though the system is described by three variables, we do not need to observe all of them to fit the trajectory (similarly to the pharmacokinetic example in the paper). ODE discovery methods assume that all variables are observed which constrains their applicability in many settings.

## B.6 Duffing oscillator

Chaotic systems usually have some kind of oscillatory behavior (i.e., it cannot be described by a finite composition). As discussed in Appendix G.1, it means that chaotic systems are currently beyond the capabilities of Semantic ODEs as they would not be able to correctly predict beyond the seen time domain. However, we could use it for a prediction on a bounded time domain. We have compared different models on the Duffing oscillator. The results can be seen in Table 6.

Table 6: Comparison of Average RMSE obtained by different models on the Duffing oscillator datasets (with two noise settings). Average performance over 5 random seeds and data splits is shown with standard deviation in the brackets.

| Method | Duffing oscillator | |
| --- | --- | --- |
| | low noise | high noise |
| SINDy-5 | $0.278_{(.032)}$ | $0.389_{(.059)}$ |
| WSINDy-5 | $0.262_{(.033)}$ | $0.361_{(.072)}$ |
| PySR-20 | $0.312_{(.049)}$ | $0.396_{(.020)}$ |
| SINDy | $0.284_{(.026)}$ | $0.386_{(.022)}$ |
| WSINDy | $0.263_{(.027)}$ | $0.339_{(.037)}$ |
| Neural ODE | $0.212_{(.018)}$ | $0.291_{(.021)}$ |
| Neural Laplace | $0.176_{(.032)}$ | $0.299_{(.017)}$ |
| DeepONet | $0.429_{(.084)}$ | $0.528_{(.066)}$ |
| Semantic ODE | $\mathbf{0.096}_{(.023)}$ | $\mathbf{0.236}_{(.001)}$ |

## B.7 Real datasets

We compare the performance of Semantic ODE against other baselines on two real datasets. The tumor growth dataset is based on the dataset collected by Wilkerson et al. (2017) based on eight clinical trials. We follow the preprocessing steps by Qian et al. (2022). The drug concentration dataset is based on data collected by (Woillard et al., 2011). The results are shown in Table 7.

Table 7: Comparison of Average RMSE obtained by different models on two real datasets. Average performance over 5 random seeds and data splits is shown with standard deviation in the brackets. "Semantic ODE*" is a variant of Semantic ODE where we incorporated a semantic inductive bias about the shape of the trajectory. We specified the composition to always be $(s_{-+c}, s_{++u})$ for the tumor growth dataset and $(s_{+-c}, s_{--c}, s_{-+h})$ for the drug concentration dataset. Note, the implementation of WSINDy we used cannot work with trajectories as sparse as in the drug concentration dataset.

| | Tumor growth (real) | Drug concentration (real) |
| --- | --- | --- |
| SINDy-5 | $0.243_{(.019)}$ | $0.286_{(.021)}$ |
| WSINDy-5 | $0.237_{(.015)}$ | NaN |
| PySR-20 | $0.536_{(.346)}$ | $0.257_{(.022)}$ |
| SINDy | $0.249_{(.029)}$ | $0.286_{(.014)}$ |
| WSINDy | $0.236_{(.016)}$ | NaN |
| Neural ODE | $0.228_{(.018)}$ | $0.263_{(.032)}$ |
| Neural Laplace | $0.243_{(.029)}$ | $0.302_{(.022)}$ |
| DeepONet | $0.242_{(.016)}$ | $0.265_{(.020)}$ |
| Semantic ODE | $0.234_{(.019)}$ | $0.264_{(.021)}$ |
| Semantic ODE* | $0.229_{(.019)}$ | $0.243_{(.015)}$ |

## B.8 Bifurcations

We believe that our framework is uniquely positioned to perform quite well on systems exhibiting bifurcations (when a small change to the parameter value causes a sudden qualitative change in

the system's behavior). In our framework, bifurcation occurs when the composition map predicts a different composition. As discussed in Appendix G.2, in the future, the semantic predictor may take as input not only the initial conditions but also other auxiliary parameters. We can then represent the composition map as a decision tree that divides the input space into different compositions. This decision tree then informs us where bifurcations occur.

We hope the following proof of concept based on the current implementation demonstrates that it is a viable approach. Instead of predicting a trajectory from its initial condition, we fix the initial condition to be always the same and predict a trajectory based on the parameter $r$ that we observe in our dataset. We generate the trajectories given the following differential equation

$$\dot{x} = rx - x^2 \tag{10}$$

The initial condition $x(0) = 1$ and $r$ sampled uniformly from $(-1, 2)$. We choose the set of compositions to be $(s_{+-h}), (s_{-+h})$ and record the position of the bifurcation point found by our algorithm (as opposed to the ground truth). The mean absolute error for different noise settings can be seen in Figure 12. Note that the range of values of the trajectory is $(0, 2)$, and even in high noise settings, the location of the bifurcation point can be identified.

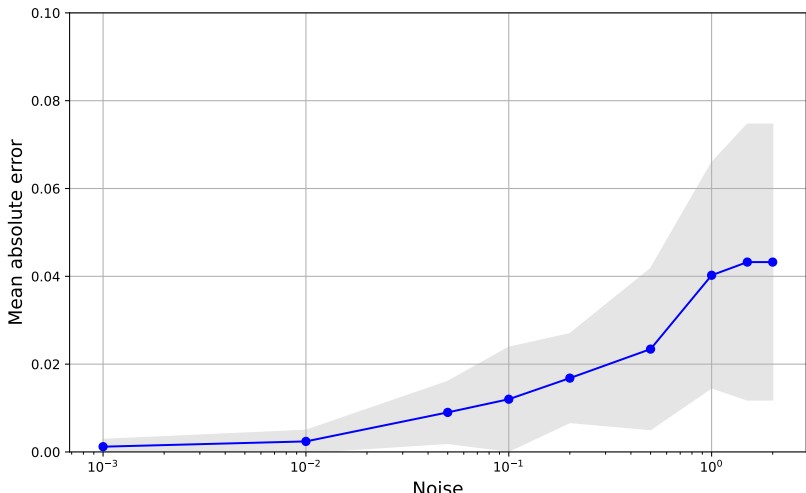

Figure 12: Mean absolute error on the predicted bifurcation point for different noise settings.

## B.9 GENERALIZATION

We show in Section 6.3 how our model can extrapolate well to unseen time domains. In this section, we show how we can make Semantic ODE generalize to previously unseen data points (initial conditions) by extrapolating the property map. We observe that each property function in Figure 8 looks approximately like a linear function. Thus, we fit a linear function to each of these functions and then evaluate our model on initial conditions from range $(1.0, 1.5)$. Note that our training set only contained initial conditions from $(0, 1)$. The semantic representation of the resulting model can be seen in Figure 13. We compare the performance of this model to ODEs from Table 2. The results can be seen in Table 8. We can see that our model has suffered only a small drop in performance even though it has never seen a single sample from that distribution. It also performs much better than any other ODE tested.

## C TRAINING OF THE SEMANTIC PREDICTOR

Training of Semantic ODE requires fitting its semantic predictor (trajectory predictor is fixed). This is done in two steps. First, we train the composition map $F_{com}$ and then the property map $F_{prop}$. It

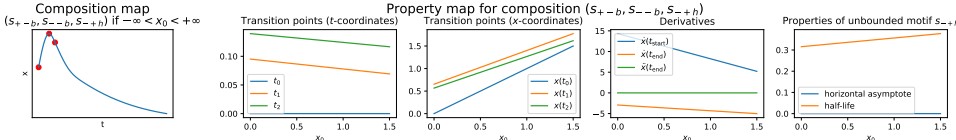

Figure 13: Semantic representation of the semantic ODE made to generalize to $x_0 \in (0, 1.5)$.

Table 8: Results of fitting Semantic ODE and SINDy to the pharmacokinetic dataset.

| Model | Syntactic biases | Semantic biases | Syntactic representation | Semantic representation | $x_0 \in (0,1)$ | $x_0 \in (1, 1.5)$ |
|---|---|---|---|---|---|---|
| SINDy | $\dot{x} = \sum_{i=1}^{n} \alpha_i g_i, n \le 1$ | NA | $\dot{x}(t) = -3.06x(t)t$ | NA | $0.222_{(0.041)}$ | $0.240_{(0.035)}$ |
| SINDy | $\dot{x} = \sum_{i=1}^{\hat{n}} \alpha_i g_i, n \le 2$ | NA | $\dot{x}(t) = 5.56 - 43.10x(t)t$ | NA | $0.112_{(0.027)}$ | $0.131_{(0.025)}$ |
| SINDy | $\dot{x} = \sum_{i=1}^{\hat{n}} \alpha_i g_i, n \le 5$ | NA | Equation (7) | NA | $0.101_{(0.023)}$ | $7.764_{(4.938)}$ |
| SINDy | $\dot{x} = \sum_{i=1}^{\hat{n}} \alpha_i g_i, n \le 10$ | NA | Equation (8) | NA | $0.029_{(0.005)}$ | $0.105_{(0.056)}$ |
| SINDy | $\dot{x} = \sum_{i=1}^{\hat{n}} \alpha_i g_i, n \le 15$ | NA | Equation (9) | NA | $0.020_{(0.004)}$ | $0.203_{(0.430)}$ |
| Semantic ODE | NA | $c_x : (s_{+-b}, s_{--b}, s_{-+h}), h = 0$ | NA | Figure 13 | $\mathbf{0.018}_{(.003)}$ | $\mathbf{0.023}_{(.005)}$ |

is possible to adjust the composition map before the property map is fitted or even provide your own composition map without fitting it. This constitutes one of the ways prior knowledge can be incorporated into the model. Then the dataset is divided into separate subsets, each for a different composition—according to the composition map. Then a separate property sub-map is trained on each of the subsets. A simple block diagram of training a semantic predictor is shown in Figure 6.

## C.1 COMPOSITION MAP

To fit a composition map we start with a *composition library* $\mathcal{C}' \subset \mathcal{C}$ which a set of compositions we want to consider. One can use a default set of compositions up to a certain length or filter out impossible ones to steer and accelerate the search. Then for every sample in our dataset, we measure how well each of the compositions fits the trajectory by fitting the properties that minimize prediction error. This gives us a matrix where each row is a sample and each column is a composition. An example of such matrix can be seen in Figure 10. We then use a dynamic programming algorithm to find the best split of $x_0$ into up to $I$ intervals, with different compositions on neighboring intervals, that minimize the overall prediction error for the whole dataset. We also make sure that each interval is not shorter than a prespecified threshold and contains a minimum number of samples. In our implementation, we choose the intervals to be at least 10% of the length of the entire domain and contain at least two samples. The number $I$ is chosen by the user (we use 3 in all our experiments). This procedure is described in Algorithm 1

## C.2 PROPERTY MAP

Property map consists of a few property sub-maps each trained on a different subset of data. Property sub-map predicts the properties, i.e., the coordinates of the transition points, necessary derivatives, and properties of the unbounded motif. Every single property is predicted by a different univariate function. To get these univariate functions we first choose a set of basis functions $\mathcal{B}$. By default, we choose B-Spline basis functions, identity and a constant. Then we can parameterize a univariate function using just $|\mathcal{B}|$ parameters and efficiently evaluate it using a single matrix-vector multiplication. This gives us, so-called, raw properties. In practice, we need to pass some of these functions through different transformations to ensure that the predicted properties make sense. For instance, transition points are in correct relation to one another or the derivatives have the correct sign. This procedure is summarized in Algorithm 2.

$t$-**coordinates**  Instead of predicting the $t$-coordinates directly, we predict the intervals between them and the $t$-coordinate of the last transition point. We use softmax to transform raw properties into positive values that add up to 1. We then multiply it by the $t$-coordinate of the last transition point (that was obtained by passing a raw property through a sigmoid function scaled to cover the interval of interest). We can then use a cumulative sum over the intervals to get the desired $t$-coordinates.

---

**Algorithm 1** Algorithm for learning the Composition Map $F_{\text{com}}$.

---

**Input:** Dataset $\{(x_0^{(d)}, \boldsymbol{t}^{(d)}, \boldsymbol{y}^{(d)})\}_{d=1}^D$, compositions library $\mathcal{C}'$, maximum number of branches $I$, trajectory predictor $F_{\text{traj}}^0 : \mathcal{C} \times \mathcal{P} \to C^0$

**Output:** Composition map $F_{\text{com}} : \mathbb{R} \to \mathcal{C}$

1: **procedure** COMPUTELOSS($d, c, p$)
2:     loss $\leftarrow \frac{1}{N_d} \sum_{n=1}^{N_d} \left( F_{\text{traj}}^0(c, p)(t_n^{(d)}) - y_n^{(d)} \right)^2$
3:     **return** loss
4: **end procedure**
5: Initialize loss table LossTable($d, c$), DP table $L(d, c, b)$ and backtracking table $B(d, c, b)$
6: **for** $d = 1$ to $D$ **do**
7:     **for all** $c \in \mathcal{C}'$ **do**
8:         LossTable($d, c$) $\leftarrow \min_{p \in \mathcal{P}_c}$ COMPUTELOSS($d, c, p$)
9:     **end for**
10: **end for**
11: Sort samples so that $x_0^{(1)} \le x_0^{(2)} \le \ldots \le x_0^{(D)}$
12: **for** $d = D$ down to 1 **do**
13:     **for all** $c \in \mathcal{C}'$ **do**
14:         **for** $i = 0$ to $I - 1$ **do**
15:             **if** $d == D$ **then**
16:                 $L(d, c, i) \leftarrow$ LossTable($d, c$)
17:             **else**
18:                 **if** $i == 0$ **then**
19:                     $L(d, c, 0) \leftarrow$ LossTable($d, c$) $+ L(d + 1, c, 0)$
20:                     $B(d, c, 0) \leftarrow c$
21:                 **else**
22:                     $L_{\text{stay}} \leftarrow$ LossTable($d, c$) $+ L(d + 1, c, i)$
23:                     $L_{\text{switch}} \leftarrow$ LossTable($d, c$) $+ \min\limits_{c' \ne c} L(d + 1, c', i - 1)$
24:                     **if** $L_{\text{stay}} \le L_{\text{switch}}$ **then**
25:                         $L(d, c, i) \leftarrow L_{\text{stay}}$
26:                         $B(d, c, i) \leftarrow c$
27:                     **else**
28:                         $L(d, c, i) \leftarrow L_{\text{switch}}$
29:                         $B(d, c, i) \leftarrow c'$ corresponding to $\min_{c' \ne c} L(d + 1, c', i - 1)$
30:                   **end if**
31:                 **end if**
32:             **end if**
33:         **end for**
34:     **end for**
35: **end for**
36: $(c^*, i^*) = \arg\min_{c \in \mathcal{C}', 0 \le i \le I-1} L(1, c, i)$
37: Initialize composition sequence $\{c_d^*\}_{d=1}^D$
38: $c_1^* \leftarrow c^*; i \leftarrow i^*$
39: **for** $d = 1$ to $D - 1$ **do**
40:     $c_{d+1}^* \leftarrow B(d, c_d^*, i)$
41:     **if** $c_{d+1}^* \ne c_d^*$ **then**
42:         $i \leftarrow i - 1$
43:     **end if**
44: **end for**
45: Define composition map $F_{\text{com}}(x)$ using switching points at $x_0^{(d)}$ where $c_d^* \ne c_{d+1}^*$

---

$x$**-coordinates** Instead of predicting the $x$-coordinates directly, we predict the absolute value of the difference between $x$-coordinates of the consecutive transition points. We do that by passing the raw property through a softplus function. Then based on the monotonicity of a given motif, we either

---

**Algorithm 2** Algorithm for Learning the Property Map $F_{\text{prop}}$.

---

**Input:** Dataset $\{x_0^{(d)}, \boldsymbol{t}^{(d)}, \boldsymbol{y}^{(d)}\}_{d=1}^{D}$, trajectory predictor $F_{\text{traj}}^0 : \mathcal{C} \times \mathcal{P} \to C^0$, composition map
  $F_{\text{com}} : \mathbb{R} \to \mathcal{C}$, set of basis functions $\mathcal{B}$
**Output:** Property map $F_{\text{prop}} : \mathbb{R} \to \mathcal{P}$
 1: **for** $d = 1$ to $D$ **do**
 2:   Evaluate basis functions: $\boldsymbol{B}^{(d)} = [b_1(x_0^{(d)}), b_2(x_0^{(d)}), \ldots, b_{|\mathcal{B}|}(x_0^{(d)})]$, where $b_i \in \mathcal{B}$
 3: **end for**
 4: Form matrix $\boldsymbol{B} \in \mathbb{R}^{D \times |\mathcal{B}|}$ with rows $\boldsymbol{B}^{(d)}$
 5: **for** $d = 1$ to $D$ **do**
 6:   Compute composition: $c^{(d)} = F_{\text{com}}(x_0^{(d)})$
 7: **end for**
 8: Partition dataset into subsets $\{\mathcal{D}_c\}$ where $\mathcal{D}_c = \{d \mid c^{(d)} = c\}$
 9: **for** each subset $\mathcal{D}_c$ **do**
10:   Initialize parameter matrix $\boldsymbol{W}^{(c)} \in \mathbb{R}^{|\mathcal{B}| \times P}$, where $P$ is the number of raw properties
11:   Extract corresponding basis evaluations $\boldsymbol{B}^c = \boldsymbol{B}[\mathcal{D}_c, :]$
12:   **repeat**
13:     Compute raw properties: $\tilde{\boldsymbol{P}}_c = \boldsymbol{B}^c \boldsymbol{W}^{(c)}$
14:     Use $\tilde{\boldsymbol{P}}_c$ to predict the properties $\{F_{\text{prop}}^{(c)}(x_0^{(d)})\}_{d \in \mathcal{D}}$ i.e.,
15:       Coordinates of the transition points
16:       Derivatives
17:       Properties of unbounded motif
18:     $L \leftarrow \frac{1}{|\mathcal{D}_c|} \sum_{d \in \mathcal{D}_c} \frac{1}{N_d} \sum_{n=1}^{N_d} \left( F_{\text{traj}}^0(c, F_{\text{prop}}^{(c)}(x_0^{(d)}))(t_n^{(d)}) - y_n^{(d)} \right)^2$
19:     Update parameters $\boldsymbol{W}^{(c)}$ to minimize $L$ (e.g., L-BFGS)
20:   **until** Convergence of optimization
21: **end for**
22: Define $F_{\text{prop}}$ as $F_{\text{prop}}(x_0) = F_{\text{prop}}^{F_{\text{com}}(x_0)}(x_0)$

---

add or subtract this number from the previous $x$-coordinate to obtain all $x$-coordinates respecting the composition.

**Derivative at the first transition point**  We pass the raw property through a sigmoid function and then we scale and translate it to obtain a value in a specific range as described in Table 9.

**Derivatives at the last transition point**  If there is at least one bounded motif in the composition, then the last transition point needs to be either an inflection point or a local extremum. As such, either the first or the second derivative vanishes and does not need to be trained. The trajectory during training is predicted through $F_{\text{traj}}^0$ that can only accept one derivative constraint at the last transition point. That is why the other derivative (the one that does not vanish) is implicitly determined by this trajectory predictor instead of being trained explicitly. We first find the trajectory and then look at the non-vanishing derivative and use it as the value predicted by the property map.

**Properties of the unbounded motif**  All raw properties for the unbounded motifs are passed through the softmax to make them positive. We then adjust them on a case-by-case basis to make sure that the motif properties make sense. For instance, instead of predicting the value of the horizontal asymptote in $s_{-+h}$ directly, we predict the distance between the asymptote and $x(t_{\text{end}})$. As the predicted distance is positive, after subtracting it from $x(t_{\text{end}})$, we get a valid value for the horizontal asymptote.

After we get all the properties, we pass it to $F_{\text{traj}}^0$ to predict the trajectory. We calculate mean squared error between the predicted trajectories and the observed ones. We also add two additional penalty terms. One discourages too much difference between derivatives at the transition points and the other discourages too large derivatives at the end. We calculate the total loss and update the parameters. In our implementation, we use the L-BFGS algorithm (Liu & Nocedal, 1989).

# D   TRAJECTORY PREDICTOR

## D.1   BOUNDED PART OF THE TRAJECTORY

We decide to define $\mathcal{X}_|$ as a set of cubic splines. They are piecewise functions where each piece is defined as a cubic polynomial. The places where two cubics are joined are called *knots*. Cubic splines require that the first and second derivatives at the knots be the same for neighboring cubics so that the cubic spline is guaranteed to be twice continuously differentiable. Cubic splines are promising because they are flexible, and for a fixed set of knots, the equations for their values and derivatives are linear in their parameters. Thus by just solving a set of linear equations, it is straightforward to find a function $x_|$ that *seemingly matches* $(c_|, p_|)$ (denoted as $x_| \sim (c_|, p_|)$ and formally defined in Definition 5 in Appendix A.2), i.e., passes through the transition points in $p_|$, and has the correct first derivative values at local extrema and boundary points $(t_0, t_{\text{end}})$ as well as correct second derivatives at inflection points and the endpoint $(t_{\text{end}})$. The challenge arises because $x_| \sim (c_|, p_|)$ may not imply $x_| \equiv (c_|, p_|)$. This is illustrated in Figure 14 in Appendix D.1. However, it is possible to make the implication hold by imposing additional conditions. We prove this in Theorem 1.

**Theorem 1.** *Let $u_|, v_|$ be bounded trajectory parts on $(t_0, t_{end})$ and assume $v_| \equiv (c_|, p_|)$, where $(c, p)$ is a semantic representation. If both of the following hold for every pair of consecutive transition points $(a, b)$ of $v_|$ then $u_| \sim (c_|, p_|) \implies u_| \equiv (c_|, p_|)$.*

- *$\forall t \in (a, b)\ \text{sign}(\ddot{u}_|(t)) = \text{sign}(\ddot{v}_|(t))$, and*
- *if neither of $\{a, b\}$ is a local extremum, $\text{sign}(\dot{u}_|(t)) = \text{sign}(\dot{v}_|(t))$ for $t \in \{a, b\}$*

*Proof.* See Appendix D.1.1. $\qquad\square$

We come up with two different ways of imposing these conditions that lead us to develop two trajectory predictors: $F_{\text{traj}}^0 : \mathcal{C} \times \mathcal{P} \to C^0$ and $F_{\text{traj}}^2 : \mathcal{C} \times \mathcal{P} \to C^2$. We use $F_{\text{traj}}^0$ during training of $F_{\text{sem}}$ as it is fast and differentiable, but the found trajectory may not be in $C^2$ (only continuous). At inference, we use $F_{\text{traj}}^2$ that is slower and not differentiable but ensures that the trajectory is in $C^2$.

### D.1.1   PROOF OF THEOREM 1

We provide proof of Theorem 1 below. But before we prove it, we need the following lemma.

**Lemma 1.** *Let $u_|, v_|$ be bounded trajectory parts on $(t_0, t_{end})$ and assume $v_| \equiv (c_|, p_|)$, where $(c, p)$ is a semantic representation, $u_| \sim (c_|, p_|)$, and the composition of $u_|$ is different from $c_|$. Then there exists a pair of consecutive transition points $(a, b)$ of $v_|$ such that*

- *$\exists t \in (a, b)$ such that $\text{sign}(\ddot{u}_|(t)) \neq \text{sign}(\ddot{v}_|(t))$ or;*
- *if neither of $\{a, b\}$ is a local extremum, $\exists t \in \{a, b\}$ such that $\text{sign}(\dot{u}_|(t)) \neq \text{sign}(\dot{v}_|(t))$*

*Proof.* First, we will show that there needs to be two consecutive transition points of $v_|$, $a, b$, where $v_| \sim s|[a, b]$ and $u_| \not\sim s|[a, b]$. Suppose for every pair of consecutive transition points of $v_|$, denoted $a, b$, $v_| \sim s|[a, b]$ and $u_| \sim s|[a, b]$ for some bounded motif $s$. Then necessarily $u_|$ has the same composition as $v_|$ (as they are defined on the same interval). Therefore if the composition of $u_|$ is different from $c_|$ then there needs to be two consecutive transition points of $v_|$, $a, b$, where $v_| \sim s|[a, b]$ and $u_| \not\sim s|[a, b]$.

We now consider two cases. Either there is a motif $s' \neq s$ such that $u_| \sim s'|[a, b]$ or there is no such single motif.

*Case 1.*

Let us assume that $u_| \sim s'|[a, b]$, where $s' \neq s$ is a different bounded motif. Then, as $u_| \sim (c_|, p_|)$, we know that $s$ and $s'$ have the same monotonicity. If $v_|$ is increasing on $[a, b]$ then $v_|(b) > v_|(a)$, which implies $u_|(b) > u_|(a)$. Thus $u_|$ is also increasing on $[a, b]$. Similarly, if $v_|$ is decreasing on $[a, b]$. Let us assume, without loss of generality, that $v_|$ is increasing on $[a, b]$. If $v_|$ is also convex then $u_|$ needs to necessarily be concave (as $s' \neq s$). That means that $\forall t \in (a, b)\ \text{sign}(\ddot{u}_|(t)) \neq \text{sign}(\ddot{v}_|(t))$. Thus one of the conditions in Lemma 1 is satisfied. Similarly, if $v_|$ is decreasing or concave.

*Case 2.*

Let us assume that there is no single motif $s' \neq s$ such that $u_| \sim s'|[a, b]$. That means that $u_|$ has a non-empty set of transition points $Q \subset (a, b)$.

If $a$ or $b$ is a local extremum of $v_|$ then there exists $q \in Q$ such that $q$ is an inflection point of $u_|$ because we cannot have two local extrema as consecutive transition points. Thus $u_|$ changes curvature at $q$ and necessarily there exists a point $t \in (a, b)$ such that $\text{sign}(\ddot{u}_|(t)) \neq \text{sign}(\ddot{v}_|(t))$.

If neither $a$ nor $b$ is a local extremum of $v_|$, then either there is an inflection point in $Q$ and we arrive at the same conclusion as before or there are no inflection points in $Q$. In that case, $Q$ contains only local extrema. However, we cannot have two local extrema as consecutive transition points. Therefore $Q$ contains only one local extremum. That means that $\text{sign}(\dot{u}_|(a)) \neq \text{sign}(\dot{u}_|(b))$. But we have $\text{sign}(\dot{v}_|(a)) = \text{sign}(\dot{v}_|(b))$ or $\dot{v}_|(a) = 0$ or $\dot{v}_|(b) = 0$. In all cases, either $\text{sign}(\dot{u}_|(a)) \neq \text{sign}(\dot{v}_|(a))$ or $\text{sign}(\dot{v}_|(b)) \neq \text{sign}(\dot{v}_|(b))$. Which is what we needed to show. □

We can now prove Theorem 1.

*Proof.* Let us assume that $u_| \sim (c_|, p_|)$. To show $u_| \equiv (c_|, p_|)$ it is sufficient to show that $u_|$ has composition $c_|$. For contradiction, let us assume that the composition of $u_|$ is different than $c_|$. By Lemma 1, there exist two consecutive transition points of $v_|$, denoted $a$ and $b$, such that $\exists t \in (a, b)$ such that $\text{sign}(\ddot{u}_|(t)) \neq \text{sign}(\ddot{v}_|(t))$ or, if neither of $\{a, b\}$ is a local extremum, $\text{sign}(u'(a)) \neq \text{sign}(v'(a))$ or $\text{sign}(u'(b)) \neq \text{sign}(v'(b))$. This contradicts the assumptions of our theorem. Namely, that for every pair of consecutive transition points $(a, b)$ of $v_|$, $\forall t \in (a, b)$ $\text{sign}(\ddot{u}_|(t)) = \text{sign}(\ddot{v}_|(t))$, and if neither of $\{a, b\}$ is a local extremum, $\text{sign}(\dot{u}_|(t)) = \text{sign}(\dot{v}_|(t))$ for $t \in \{a, b\}$. Thus the composition of $u_|$ is $c_|$ and, therefore, $u_| \equiv (c_|, p_|)$. □

### D.1.2 $C^0$ TRAJECTORY PREDICTOR

**Range of values for the derivative at $t_0$**  To ensure that the trajectory found by $F_{\text{traj}}^0$ has the correct semantic representation, we need to constrain the values of the first derivative of $x$ at $t_0$. These values depend on the slope of the line connecting the first transition point with the second one, i.e., we define slope $\kappa$ as $\kappa = \frac{x(t_1) - x(t_0)}{t_1 - t_0}$. The values also depend on the motif and the nature of the second transition point. They are presented in the table below.

| Motif | $t_1$ | Range |
|-------|-------|-------|
| $s_{++b}$ | inflection | $(0, \kappa)$ |
| $s_{+-b}$ | maximum | $(1.5\kappa, 3\kappa)$ |
| $s_{+-b}$ | inflection | $(\kappa, 3\kappa)$ |
| $s_{-+b}$ | minimum | $(3\kappa, 1.5\kappa)$ |
| $s_{-+b}$ | inflection | $(3\kappa, \kappa)$ |
| $s_{--b}$ | inflection | $(\kappa, 0)$ |

Table 9: Allowed values for $\dot{x}(t_0)$ enforced by $F_{\text{traj}}^0$.

**Trajectory conforms to the semantic representation**  In this section, we slightly relax the definition of conformity by allowing the part of the trajectory between two inflection points to be modeled as a straight line with an appropriate slope. Even though it does not match any of the defined motifs, in this section, we assume a straight line matches both the convex and concave variants of motifs with the corresponding monotonicity. That means we can relax the first assumption in Theorem 1 to $\forall t \in (a, b)$ $\text{sign}(\ddot{u}_|(t)) = \text{sign}(\ddot{v}_|(t))$ or $\text{sign}(\ddot{u}_|(t)) = 0$ if both $a, b$ are inflection points. To prove that $F_{\text{traj}}^0(c_|, p_|) \equiv (c_|, p_|)$, we first show that $F_{\text{traj}}^0(c_|, p_|) \sim (c_|, p_|)$.

**Lemma 2.** *Let $(c, p) \in \mathcal{C} \times \mathcal{P}$ be a semantic representation predicted by $F_{sem}$. Then $F_{traj}^0(c_|, p_|) \sim (c_|, p_|)$.*

*Proof.* As each motif is described by a separate cubic polynomial, it is defined by four parameters. To predict the trajectory, we set four constraints. Two for the $x$-coordinates of the transition points, and two for the derivatives (one for each transition point). If the transition point is $t_0$ then we set the first derivative at $t_0$ to the value specified in $p_|$. If it is a local extremum then we set the first derivative at it to 0. If it is an inflection point then we set the second derivative at it to 0. The only condition left to satisfy is the second derivative at $t_{\text{end}}$, if $t_{\text{end}}$ is a local extremum, or the first derivative at $t_{\text{end}}$ if it is an inflection point. However, as we use $F_{\text{traj}}^0$ for training the semantic predictor $F_{\text{sem}}$, we are guaranteed that the "other" automatically matches the one specified in $p_|$. Therefore, by Definition 5, $F_{\text{traj}}^0(c_|, p_|) \sim (c_|, p_|)$. $\square$

We will know prove that $F_{\text{traj}}^0(c_|, p_|) \equiv (c_|, p_|)$.

**Theorem 2.** *Let $(c, p) \in \mathcal{C} \times \mathcal{P}$ be a semantic representation predicted by $F_{sem}$. Then $F_{traj}^0(c_|, p_|) \equiv (c_|, p_|)$.*

*Proof.* By Lemma 2, we get that $F_{\text{traj}}^0(c_|, p_|) \sim (c_|, p_|)$. We will now use Theorem 1 to show that $F_{\text{traj}}^0(c_|, p_|) \equiv (c_|, p_|)$. Let $v_| \equiv (c_|, p_|)$ and let us denote $F_{\text{traj}}^0(c_|, p_|)$ as $u_|$. Let us take any pair of consecutive transition points of $v_|$ and denote them as $a, b$.

Let us consider two cases. First, when $a \neq t_0$ and second, where $a = t_0$.

*Case 1. $a \neq t_0$*

At least one of $\{a, b\}$ is an inflection point, as two local extrema cannot be two consecutive transition points. Denote this point as $q$. That means that $\ddot{u}_|(q) = \ddot{v}_|(q) = 0$. By definition of $F_{\text{traj}}^0$, $u_|$ is a cubic on $[a, b]$. That means $\ddot{u}_|$ is a straight line on $[a, b]$, passing through 0 at $q$. Now we are going to consider two subcases. Either the other transition point is a local extremum or it is another inflection point.

*Case 1.1* The other transition point is a local extremum.

Without loss of generality, let us assume that $b = q$ is an inflection point, $a$ is a local extremum, and that $u_|(a) < u_|(b)$. That means that at some point $t \in (a, b)$, $\dot{u}_|(t) > 0$. We also know that $\dot{u}_|(a) = 0$. That means that at some point $t \in (a, b)$, $\ddot{u}_| > 0$. As $\ddot{u}_|$ is a straight line passing through 0 at $b$, we get that $\forall t \in (a, b)$ $\ddot{u}_|(t) > 0$. This means that $\forall t \in (a, b)$ $\text{sign}(\ddot{u}_|(t)) = \text{sign}(\ddot{v}_|(t))$. Similarly for other cases (where $a$ is an inflection point, or $u_|(a) > u_|(b)$).

*Case 1.2* Both transition points are inflection points.

If $a$ and $b$ are both transition points, then $\forall t \in (a, b)$ $\ddot{u}_|(t) = 0$. This means $u_|$ is a straight line passing through $(a, u(a))$ and $(b, u(b))$. As such $\text{sign}(\dot{u}_|(a)) = \text{sign}(\dot{u}_|(b)) = \text{sign}(u_|(b) - u_|(a)) = \text{sign}(v_|(b) - v_|(a)) = \text{sign}(\dot{v}_|(a)) = \text{sign}(\dot{v}_|(b))$.

*Case 2. $a = t_0$*

If $a = t_0$ then $\dot{u}(a) = \dot{u}(t_0)$ is in the range described in Table 9. Without loss of generality, let us assume that $t_0 = 0$. Let us describe $u_|$ on $[a, b]$ as $\beta_3 t^3 + \beta_2 t^2 + \beta_1 t + \beta_0$. Let us denote $u_|(0) = u_0$ and $u_|(t_1) = u_1$. Let us also denote the required first derivative at 0 as $u_0'$. As $u_|$ needs to pass through both $(0, u_0)$ and $(t_1, u_1)$ and needs to have $\dot{u}_|(0) = u_0'$, we get the following equations.

$$\beta_0 = u_0 \tag{11}$$

$$\beta_3 t_1^3 + \beta_2 t_1^2 + \beta_1 t_1 + \beta_0 = u_1 \tag{12}$$

$$\beta_1 = u_0' \tag{13}$$

As in Table 9, we denote $\frac{u(t_1) - u(t_0)}{t_1 - t_0}$ as $\kappa$. Then $u_1 = u_0 + \kappa \times t_1$. The second equation then reduces to

$$\beta_3 t_1^2 + \beta_2 t_1 + u_0' = \kappa \tag{14}$$

Now we have to go by all six cases in Table 9.

*Case 2.1 $s_{++b}$, $t_1$ is an inflection point.*

As $t_1$ is an inflection point, we get that

$$6\beta_3 t_1 + 2\beta_2 = 0 \tag{15}$$

That means

$$\beta_2 = -3\beta_3 t_1 \tag{16}$$

By substituting into the previous equation, we get

$$\beta_3 t_1^2 - 3\beta_3 t_1^2 + u_0' = \kappa \tag{17}$$

which gives us

$$\beta_3 = \frac{(u_0' - \kappa)}{2t_1^2} \tag{18}$$

$$\beta_2 = -\frac{3(u_0' - \kappa)}{2t_1} \tag{19}$$

To satisfy the conditions of Theorem 1, we need to ensure that $\ddot{u}_|(t) > 0$ for all $t \in (0, t_1)$. This holds if

$$\frac{3(u_0' - \kappa)}{t_1^2} t - \frac{3(u_0' - \kappa)}{t_1} > 0 \tag{20}$$

This is equivalent to

$$3(u_0' - \kappa)(t - t_1) > 0 \tag{21}$$

As $t < t_1$, this is equivalent to $u_0' - \kappa < 0$ which is equivalent to

$$u_0' < \kappa \tag{22}$$

As our motif is $s_{++b}$, $u_0' \geq 0$. Thus $0 \leq u_0' < \kappa$ as specified in Table 9.

Moreover, as neither $t_0$ nor $t_1$ is a local extremum, we need to check the signs of the first derivatives. We do not need to check $t_0$, but we need to ensure that $\dot{u}_|(t_1) > 0$. But this follows from the fact that $\dot{u}_|(t_0) > 0$ and $\ddot{u}_|(t) > 0$ for all $t \in (t_0, t_1)$.

*Case 2.2* $s_{+-b}$, $t_1$ is a maximum.

As $t_1$ is a maximum, we get

$$3\beta_3 t_1^2 + 2\beta_2 t_1 + u_0' = 0 \tag{23}$$

That means

$$\beta_2 = \frac{-u_0' - 3\beta_3 t_1^2}{2t_1} \tag{24}$$

By substituting into Equation (14), we obtain

$$\beta_3 t_1^2 + 1/2 \times (-u_0' - 3\beta_3 t_1^2) + u_0' = \kappa \tag{25}$$

By rearranging, we get

$$\beta_3 = \frac{u_0' - 2\kappa}{t_1^2} \tag{26}$$

$$\beta_2 = \frac{3\kappa - 2u_0'}{t_1} \tag{27}$$

To satisfy the conditions of Theorem 1, we need to ensure that $\ddot{u}_|(t) < 0$ for all $t \in (0, t_1)$. That is we need

$$6\frac{u_0' - 2\kappa}{t_1^2} t + 2\frac{3\kappa - 2u_0'}{t_1} < 0 \tag{28}$$

which is equivalent to

$$(6u_0' - 12\kappa)t + (6\kappa - 4u_0')t_1 < 0 \tag{29}$$

After rearranging

$$\kappa(3t_1 - 6t) < u_0'(2t_1 - 3t) \tag{30}$$

Let us denote $t/t_1$ as $t'$. Then the previous equation is equivalent to

$$t'(3u_0' - 6\kappa) < 2u_0' - 3\kappa \tag{31}$$

As this is supposed to be true for all $t' \in (0, 1)$, two things need to be true.

$$2u_0' - 3\kappa > 0 \tag{32}$$
$$3u_0' - 6\kappa < 2u_0' - 3\kappa \tag{33}$$

This gives us

$$1.5\kappa < u_0' < 3\kappa \tag{34}$$

as specified in Table 9.

*Case 2.3 $s_{+-b}$, $t_1$ is an inflection.*

We follow a similar first step as in *Case 2.1*. We arrive at the conclusion that to satisfy the condition of Theorem 1, we need to ensure that $\ddot{u}_|(t) < 0$ for all $t \in (0, t_1)$. This holds if

$$u_0' > \kappa \tag{35}$$

However, we also need to make sure that $\dot{u}_|(t_1) > 0$. That is, we need to satisfy

$$3\frac{u_0' - \kappa}{2t_1^2}t_1^2 - 2\frac{3(u_0' - \kappa)}{2t_1}t_1 + u_0' > 0 \tag{36}$$

That is equivalent to

$$-u_0' + 3\kappa > 0 \tag{37}$$

Thus $u_0'$ needs to satisfy $\kappa < u_0' < 3\kappa$ as specified in Table 9.

*Case 2.4 $s_{-+b}$, $t_1$ is a minimum.*

The steps are very similar to those in *Case 2.2*. We arrive at equation Equation (31), but the inequality has a different direction. That is, we need

$$t'(3u_0' - 6\kappa) > 2u_0' - 3\kappa \tag{38}$$

to hold for all $t' \in (0, 1)$. This can only be true if

$$2u_0' - 3\kappa < 0 \tag{39}$$
$$3u_0' - 6\kappa > 2u_0' - 3\kappa \tag{40}$$

which gives us

$$3\kappa < u_0' < 1.5\kappa \tag{41}$$

as specified in Table 9.

*Case 2.5 $s_{-+b}$, $t_1$ is an inflection point.*

The steps are analogous to those in *2.3*. However, we need to ensure that $\ddot{u}_|(t) > 0$ for all $t \in (0, t_1)$ and that $\dot{u}_|(t_1) < 0$. This gives us

$$3\kappa < u_0' < \kappa \tag{42}$$

as in Table 9.

*Case 2.6 $s_{--b}$, $t_1$ is an inflection point.*

Analogously to *Case 2.1*, we get

$$\kappa \leq u_0' < 0 \tag{43}$$

$\square$

### D.1.3 $C^2$ TRAJECTORY PREDICTOR

In this section, we describe $F_{\text{traj}}^2$ that predicts a trajectory that is twice continuously differentiable.

**Why is it challenging?** A natural idea to find a cubic spline with the corresponding composition and properties would be to follow a similar approach as in $F_{\text{traj}}^0$ by choosing knots at the transition points. However, then the problem turns out to be overdetermined, i.e., we are not guaranteed that the needed cubic spline exists. Indeed, given $S$ motifs in the bounded part of the composition, we have 5 conditions for each of the internal knots (the values for the two cubic, matching first and second derivative, and the value of one of the derivatives). This is because each internal knot is either a local extremum (first derivative vanishes) or an inflection point (second derivative vanishes). We also specify the values of the derivatives at the endpoints, which gives us overall $5(S-1) + 2 + 3 = 5S$ conditions, whereas we only have $4S$ parameters.

The conclusion is simple: we need more knots. In fact, the general formula for the number of constraints is $3(K-1) + 2(S-1) + 2 + 3 = 3K + 2S$ where $K$ is the number of cubics, and $S$ is the number of motifs. By equating this number to $4K$, we get the optimal number of cubics is: $2S$. That means that we will need knots between transition points. This, however, poses a new challenge. Although the resulting cubic will have all the transition points and derivatives as required, i.e., it will seemingly match the semantic representation, we are in no way guaranteed that it will have the correct composition! Although we would like the function between two transition points to have a fixed sign of both derivatives, nothing is stopping our function from adding additional trend changes in between. This is visualized in Figure 14.

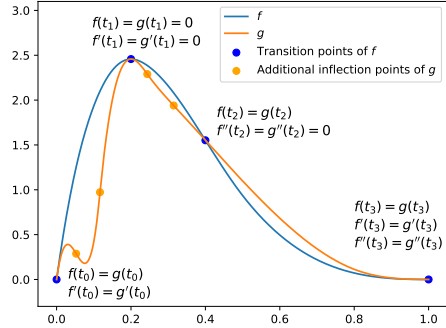

Figure 14: $f \equiv (c_|, p_|)$, $g \sim (c_|, p_|)$ but $g \not\equiv (c_|, p_|)$.

To solve this problem, we propose a completely different approach to fitting cubic splines.

**From cubic spline to a piecewise linear function and back** In this subsection, we show how we can describe a cubic spline in terms of its second derivative to reduce the number of parameters and control its second derivative. A cubic spline consisting of $K$ cubics is uniquely described by its $K+1$ knots and $4K$ coefficients (four per every cubic). However, the actual number of degrees of freedom is much smaller because the spline needs to be twice continuously differentiable at the knots. We observe that we can decrease the number of parameters by fitting the second derivative of the spline instead and then integrating it twice to get the required function (Figure 15). The second derivative of a cubic is a linear function (which can be integrated analytically), so each piece is described by just two parameters. In fact, we can describe a piecewise linear function by just the values at the knots, ensuring continuity. The values at two consecutive knots then uniquely determine the linear functions. Together with the additional two parameters for the integration constants ($c$ and $d$), we reduced the number of parameters to $K+3$. Formally, we want to find $\ddot{u} \in C^0$ and then define $u \in C^2$ as

$$u(t) = \int_{t_0}^{t} \left( \int_{t_0}^{t'} \ddot{u}(t'') \, dt'' + c \right) dt' + d, \tag{44}$$

where $\ddot{u}$ is described by $(t_0, \ldots, t_K, v_0, \ldots, v_K)$. This not only helps us decrease the number of parameters and constraints we need to impose but, crucially, allows us to exactly control $\text{sign}(\ddot{u}(t))$ by just making sure each $v_k$ is either positive or negative.

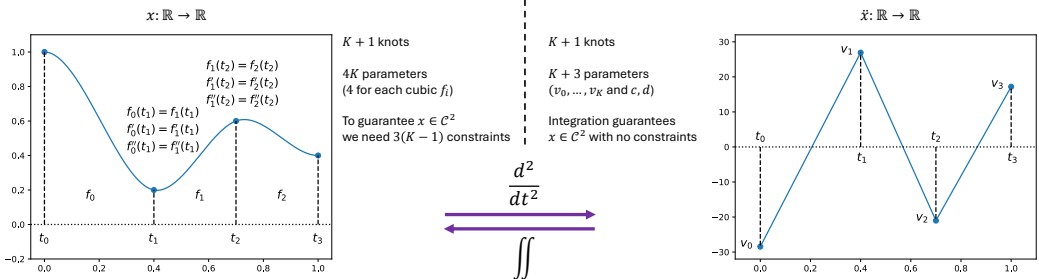

Figure 15: Traditionally, a cubic spline over $K + 1$ knots is described by $4K$ parameters and $3(K - 1)$ constraints that guarantee that the function is twice continuously differentiable. Instead, we describe it using its second derivative, which can be parametrized by $K + 1$ parameters and two integration constants. Integration guarantees the function is twice continuously differentiable without any additional constraints.

**Fitting the second derivative**  We observed that the second derivative of a cubic spline is a piecewise linear function that is entirely defined by its values at the knots. We choose the knots as the transition points and add one knot between every two of them. The positions of these knots are additional parameters of our model. Given $n$ transition points, we have $2n - 1$ knots (denoted $t_0, \ldots, t_{2n-2}$) and their associated values denoted $v_0, \ldots, v_{2n-2}$. As each antiderivative is determined up to a constant, we need additional parameters $c$ and $d$ as integration constants. Ultimately, $\ddot{u}$ is described by $(t_0, \ldots, t_{2n-2}, v_0, \ldots, v_{2n-2}, c, d)$. Overall, this gives $4n$ parameters. However, many of these parameters are directly determined by the semantic representation of $x$. In particular, the following parameters are predetermined.

- $t_{2i}$ for all $i \in [0 : n - 1]$ ($t$-coordinates of the transition points)
- $v_{2n-2}$ and all $v_{2i}$ where $t_{2i}$ is an inflection point
- $c = \dot{x}(t_0)$
- $d = x(t_0)$

In addition, for every transition point $t_{2i}$ with a specified first derivative (either a local extremum or the endpoint), the value of $v_{2i-1}$ is chosen to enforce the correct value of $\dot{u}(t_{2i})$. This can be done by observing that the value of $\dot{u}(t_{2i})$ is equal to the sum of $\dot{u}(t_{2j})$, where $t_{2j}$ is a previous transition point with a specified first derivative, and the integral of $\ddot{u}$ between those points. Namely,

$$\dot{u}(t_{2i}) = \int_{t_{2j}}^{t_{2i}} \ddot{u}(t)\, \mathrm{d}t + \dot{u}(t_{2j}) \tag{45}$$

This integral can be calculated analytically as $\ddot{u}$ is a piecewise linear function. In the end, we are left with $2n - 2$ parameters obeying the following constraints.

- $t_{2i} < t_{2i+1} < t_{2i+2}$ for all $i \in [0 : n - 2]$
- $\mathrm{sign}(v_i) = \mathrm{sign}(\ddot{x}(t_i))$

The goal is to find these parameters such that the resulting $u(t_{2i}) = x(t_{2i})$ and $\mathrm{sign}(\dot{u}(t_{2i})) = \mathrm{sign}(\dot{x}(t_{2i}))$ and this can be formed as the following objective.

$$\sum_{i=0}^{n-1} (u(t_{2i}) - x(t_{2i}))^2 + \lambda \min(\dot{u}(t_{2i}) \times \dot{x}(t_{2i}), 0) \tag{46}$$

where $\lambda$ is chosen to be very large. We then use L-BFGS-B and Powell to optimize this objective three times for different initial guesses. If the maximum error on the transition points $(\max_i |u(t_{2i}) - x(t_{2i})|)$ and on the derivatives $(\max_i |\dot{u}(t_{2i}) - \dot{x}(t_{2i})|)$ is lower than some user-defined threshold (we use $0.001$) then we have found a function with the correct semantic representation. If not then we default to using $F_{\mathrm{traj}}^0$. The threshold allows the user to choose a trade-off between how smooth the trajectory needs to be and how faithful it needs to be to its semantic representation.

To make this concrete, we consider an example where the finite part of the composition is $(s_{+-b}, s_{--b}, s_{-+b}, s_{++b})$ (see Figure 16). It is described by 20 parameters

$(t_0, \ldots, t_8, v_0, \ldots, v_8, c, d)$, but most of them are fixed (denoted by the red color in Figure 16). In particular, $t_0, t_2, t_4, t_6, t_8$ are the $t$-coordinates of the transition points. $v_4 = 0$ as an inflection point and $v_8 = 0$ as the last transition point. Moreover, $v_1, v_5, v_7$ are always chosen to ensure the first derivatives at $t_2, t_6, t_8$ have the correct values and $c = \dot{x}(t_0) = 1$, $d = x(t_0) = 0$. That leaves 8 trainable parameters $(v_0, t_1, v_2, t_3, v_3, t_5, v_6, t_7)$ constrained such that $t_0 < t_1 < t_2 < t_3 < t_4 < t_5 < t_6 < t_7 < t_8$, $v_0, v_2, v_3 < 0$, $v_6 > 0$.

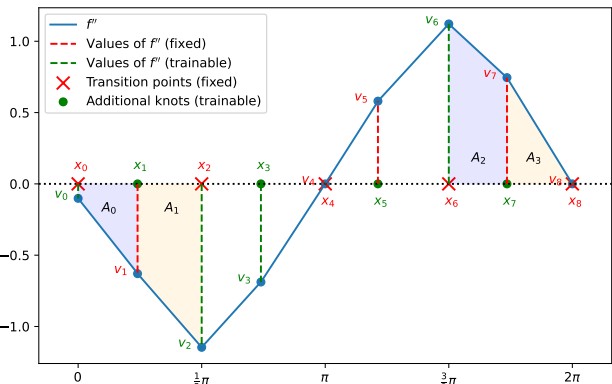

Figure 16: Example parametrization of $u$ through its second derivative. $x$ has the composition $(s_{+-b}, s_{--b}, s_{-+b}, s_{++b})$. It is described by 20 parameters $(t_0, \ldots, t_8, v_0, \ldots, v_8, c, d)$ where 12 of them are fixed and 8 are trainable.

## D.2 Unbounded part of the trajectory

In contrast to the bounded part of the trajectory, for the unbounded part we have only one predictor. It takes the unbounded motif and its properties, as well as the coordinates of the last transition point and both derivatives at this point, and predicts a trajectory $_| x$. We need the derivatives to guarantee that the trajectory is twice continuously differentiable at $t_{\text{end}}$. The first challenge is to come up with useful properties that sufficiently describe the unbounded motif.

### D.2.1 Properties of unbounded motifs

Below, we describe the properties of unbounded motifs one by one.

$s_{++u}$  This motif represents an increasing, convex function. We decided to model it as a function exhibiting an exponential-like behavior. That is, we expect it to double after a fixed time interval. As it is true for a function such as $u(t) = 3 \times 2^{\frac{t}{5}}$. In this case $u(t + 5) = 2 \times u(t)$. However, we need to generalize the notion of this doubling time to settings with arbitrary initial conditions, including negative ones. In that case, the doubling time depends on $t$. However, we can make it approach a fixed value asymptotically. Let $\gamma(t)$ be defined as value such that $_| x(t + \gamma(t)) = 2_| x(t)$. As $_| x$ is increasing and convex, $\gamma(t)$ is well-defined. Then we define our property as $\lim_{t \to \infty} \gamma(t)$ and call it "asymptotic doubling time".

$s_{--u}$  We describe $s_{--u}$ using an analogous property that we call "negative asymptotic doubling time" defined in exactly the same way. The only difference is that, in this case, doubling makes the trajectory more negative and thus the value decreases.

$s_{+-u}$  This motif represents an increasing, concave function. We decided to model it as a function that resembles a logarithm. Thus, we expect it to have an inverse behavior to the exponential function. As we double the input, we expect a constant increase in the function's value. As with the previous examples, in general, this increase would depend on the current value of the trajectory, but we can make it converge as $t \to \infty$. Formally, we define $\gamma(t)$ as a number such that $_| x(2t) = _| x(t) + \gamma(t)$. As

$_|x$ increases, this is well defined. Thus, we define the property as $\lim_{t\to\infty}\gamma(t)$ and call it "asymptotic incrementing factor".

$s_{-+u}$   We describe $s_{-+u}$ using an analogous property that we call "asymptotic decrementing factor". It is defined as $\lim_{t\to\infty}\gamma(t)$, where $\gamma(t)$ is defined as number such that $_|x(2t) = {}_|x(t) - \gamma(t)$.

$s_{-+h}$   This motif represents a decreasing, convex function with a horizontal asymptote. A natural property is the horizontal asymptote $h$. However, we decided to include one more property that describes how quickly the trajectory approaches that asymptote. We define $t_{1/2}$ as the value of $t$ where $_|x$ is in the middle between the value of the last transition point and the asymptote. Formally, $_|x(t_{1/2}) = \frac{_|x(t_{\text{end}})+h}{2}$. We call it "half-life" as it would correspond to half-life if modeled as an exponential decay.

$s_{+-h}$   We describe this motif using the same properties as $s_{-+h}$. However, as the function increases, instead of "half-life" we call it "inverse half-life".

### D.2.2   PARAMETRIZATION

Having defined the properties of interest in the previous section, we need to find a parametrization for each of these motifs that would allow us to choose an arbitrary property as well as the last transition point and the two derivatives at this point.

$s_{++u}$   We parametrize this motif as

$$_|x(t) = \theta_1 e^{\theta_2(t-t_{\text{end}})} + \theta_3(t - t_{\text{end}}) + \theta_4(t - t_{\text{end}}) + \theta_5 \tag{47}$$

We can quickly calculate that

$$_|x(t_{\text{end}}) = \theta_1 + \theta_5 \tag{48}$$

$$_|\dot{x}(t_{\text{end}}) = \theta_1\theta_2 + \theta_4 \tag{49}$$

$$_|\ddot{x}(t_{\text{end}}) = \theta_1\theta_2^2 + 2\theta_3 \tag{50}$$

Now let us define $\gamma(t)$ as earlier, i.e., $\gamma(t)$ is the value such that $_|x(t + \gamma(t)) = 2_|x(t)$. We are now going to prove that $\lim_{t\to\infty} g(t) = \frac{\log(2)}{\theta_2}$ which we denote as simply $\gamma_*$. To do it we first prove that $\lim_{t\to\infty} 2_|x(t) - {}_|x(t + \gamma_*) = 0$.

$$
\begin{aligned}
\lim_{t\to\infty} \frac{2_|x(t)}{_|x(t+\gamma_*)} &= \lim_{t\to\infty} \frac{2\left[\theta_1 e^{\theta_2(t-t_{\text{end}})} + \theta_3(t - t_{\text{end}}) + \theta_4(t - t_{\text{end}}) + \theta_5\right]}{\theta_1 e^{\theta_2(t+\gamma-t_{\text{end}})} + \theta_3(t + \gamma - t_{\text{end}}) + \theta_4(t + \gamma_* - t_{\text{end}}) + \theta_5} \\
&= \lim_{t\to\infty} \frac{2\theta_1 e^{\theta_2(t-t_{\text{end}})}}{\theta_1 e^{\theta_2(t-t_{\text{end}})} e^{\theta_2\gamma_*}} \\
&= \frac{2}{e^{\theta_2\gamma_*}} \\
&= \frac{2}{e^{\theta_2\left(\frac{\log 2}{\theta_2}\right)}} \\
&= \frac{2}{2} = 1
\end{aligned} \tag{51}
$$

From that it follows $\lim_{t\to\infty} 2_|x(t) - {}_|x(t + \gamma_*) = 0$. As the function $\frac{2_|x(t)}{_|x(t+\gamma)}$ is continuous in both $t$ and $\gamma$ and for every $t > t_{\text{end}}$ it is bounded for all $\gamma > 0$, it is absolutely continuous in $\gamma$. From that it follows that $\lim_{t\to\infty} g(t) = g_* = \frac{\log(2)}{\theta_2}$.

So we can set the parameters as follows.

$$\theta_2 = \frac{\log(2)}{\gamma_*} \tag{52}$$

$$\theta_3 = (_|\ddot{x}(t_{\text{end}}) - \theta_1\theta_2^2)/2 \tag{53}$$

$$\theta_4 = {}_|\dot{x}(t_{\text{end}}) - \theta_1\theta_2 \tag{54}$$

$$\theta_5 = {}_|x(t_{\text{end}}) - \theta_1 \tag{55}$$

We have some freedom in choosing $\theta_1$, but we need to make sure that the function is indeed increasing and convex for all $t > t_{\text{end}}$. This is true if $\theta_3 > 0$ and $\theta_4 > 0$. So we choose $\theta_1$ to be

$$\theta_1 = \min(_|\dot{x}(t_{\text{end}})/2\theta_2, \, _|\ddot{x}(t_{\text{end}})/2\theta_2^2, \, 1) \tag{56}$$

$s_{--b}$   We parametrize this motif analogously to the previous one.

$$_|x(t) = -\theta_1 e^{\theta_2(t-t_{\text{end}})} - \theta_3(t - t_{\text{end}}) - \theta_4(t - t_{\text{end}}) + \theta_5 \tag{57}$$

where

$$\theta_2 = \frac{\log(2)}{\gamma_*} \tag{58}$$

$$\theta_3 = -(_|\ddot{x}(t_{\text{end}}) - \theta_1\theta_2^2)/2 \tag{59}$$

$$\theta_4 = -_|\dot{x}(t_{\text{end}}) - \theta_1\theta_2 \tag{60}$$

$$\theta_5 = {_|x}(t_{\text{end}}) + \theta_1 \tag{61}$$

and $\theta_1$ needs to be chosen to make sure that $\theta_3$ and $\theta_4$ are positive. We choose it to be

$$\theta_1 = \min(-_|\dot{x}(t_{\text{end}})/2\theta_2, \, -_|\ddot{x}(t_{\text{end}})/2\theta_2^2, \, 1) \tag{62}$$

$s_{+-u}$   We parametrize this motif as

$$_|x(t) = \theta_1 \log(\theta_2(t - t_{\text{end}})^2 + \theta_3(t - t_{\text{end}}) + 1) + \theta_4 \tag{63}$$

where

$$\theta_1 = \gamma_*/\log(4) \tag{64}$$

$$\theta_3 = {_|\dot{x}}(t_{\text{end}})/\theta_1 = \log(4)_|\dot{x}(t_{\text{end}})/\gamma_* \tag{65}$$

$$\theta_2 = \theta_3^2/2 = (\log(4)_|\dot{x}(t_{\text{end}})/\gamma_*)^2/2 \tag{66}$$

$$\theta_4 = {_|x}(t_{\text{end}}) \tag{67}$$

where $\gamma_*$ is the property of $s_{+-u}$ that we call asymptotic incrementing factor.

It is easy to verify that this function is indeed increasing and concave. If there is a bounded motif before it, it has to be $s_{++c}$, and then $t_{\text{end}}$ is an inflection point. Thus

$$_|\ddot{x}(t_{\text{end}}) = \theta_1(2\theta_2 - \theta_3^2) = 0 \tag{68}$$

as required. If there is no bounded motif before it, there is no way to set the second derivative at $t_{\text{end}}$ to any other value than $0$. This property is fixed and not trained in the property map.

Let $\gamma(t)$ be defined as earlier, i.e.,

$$\gamma(t) = {_|x}(2t) - {_|x}(t) \tag{69}$$

We can compute it as

$$
\begin{aligned}
\gamma(t) &= {_|x}(2t) - {_|x}(t) \\
&= \theta_1 \log(\theta_2(2t - t_{\text{end}})^2 + \theta_3(2t - t_{\text{end}}) + 1) - \theta_1 \log(\theta_2(t - t_{\text{end}})^2 + \theta_3(t - t_{\text{end}}) + 1) \\
&= \theta_1 \log\left(\frac{\theta_2(2t - t_{\text{end}})^2 + \theta_3(2t - t_{\text{end}}) + 1}{\theta_2(t - t_{\text{end}})^2 + \theta_3(t - t_{\text{end}}) + 1}\right)
\end{aligned}
\tag{70}
$$

Thus

$$
\begin{aligned}
\lim_{t\to\infty} \gamma(t) &= \theta_1 \log\left(\frac{4\theta_2}{\theta_2}\right) \\
&= \theta_1 \log(4) \\
&= \gamma_*
\end{aligned}
\tag{71}
$$

as required.

$s_{-+u}$    This motif is parametrized analogously to $s_{+-u}$ as

$$_|x(t) = -\theta_1 \log(\theta_2(t - t_{\text{end}})^2 + \theta_3(t - t_{\text{end}}) + 1) + \theta_4 \tag{72}$$

where

$$\theta_1 = \gamma_*/\log(4) \tag{73}$$
$$\theta_3 = -_|\dot{x}(t_{\text{end}})/\theta_1 = -\log(4)_|\dot{x}(t_{\text{end}})/\gamma_* \tag{74}$$
$$\theta_2 = \theta_3^2/2 = (\log(4)_|\dot{x}(t_{\text{end}})/\gamma_*)^2/2 \tag{75}$$
$$\theta_4 = _|x(t_{\text{end}}) \tag{76}$$

where $\gamma_*$ is the property of $s_{+-u}$ that we call asymptotic decrementing factor.

Let us define $\gamma$ as earlier, i.e.,

$$\gamma(t) = _|x(t) - _|x(2t) \tag{77}$$

Then

$$\begin{aligned}\lim_{t\to\infty} \gamma(t) &= -\theta_1 \log\left(\frac{1}{4}\right) \\ &= \theta_1 \log(4) \\ &= \gamma_* \end{aligned} \tag{78}$$

as required.

$s_{-+h}$    We parametrize this motif as

$$_|x(t) = h(g(t - t_{\text{end}})) \tag{79}$$

where

$$h(t) = \frac{\theta_1}{1 + e^t} + \theta_2 \tag{80}$$

and $g(t)$ is appropriately defined cubic spline.

First, we need to make sure that $_|x$ is actually decreasing and convex.

$$_|\dot{x}(t) = \dot{h}(g(t - t_{\text{end}}))\dot{g}(t - t_{\text{end}}) \tag{81}$$

$$\dot{h}(t) = -\frac{\theta_1 e^t}{(e^t + 1)^2} < 0 \tag{82}$$

So to satisfy $_|x(t) < 0$ for $t \geq t_{\text{end}}$, we need $g'(t) > 0$ for $t \geq 0$. Let us now look at the second derivative.

$$_|\ddot{x}(t) = \ddot{h}(g(t - t_{\text{end}}))\dot{g}(t - t_{\text{end}})^2 + \dot{h}(g(t - t_{\text{end}}))\ddot{g}(t - t_{\text{end}}) \tag{83}$$

$$\ddot{h}(t) = \frac{\theta_1(e^t - 1)e^t}{(e^t + 1)^3} \tag{84}$$

For $t > 0$, this is always positive. Let us assume $g(0) = 0$, then $g(t) > 0$ for $t > 0$. To satisfy $_|\ddot{x}(t) > 0$ for $t > t_{\text{end}}$, we need $\ddot{g}(t) < 0$ for $t > 0$. Let us now look at $t_{\text{end}}$.

$$_|x(t_{\text{end}}) = h(g(0)) = h(0) = \theta_1/2 + \theta_2 \tag{85}$$

$$_|\dot{x}(t_{\text{end}}) = \dot{h}(0)\dot{g}(0) = -\frac{\theta_1}{4}\dot{g}(0) \tag{86}$$

$$_|\ddot{x}(t_{\text{end}}) = \ddot{h}(0)\dot{g}(0)^2 + \dot{h}(0)\ddot{g}(0) = -\frac{\theta_1}{4}\ddot{g}(0) \tag{87}$$

In addition, we also need to satisfy the properties, i.e.,

$$\lim_{t\to\infty} {}_|x(t) = h \tag{88}$$

$$_|x(t_{1/2}) = (_|x(t_{\text{end}}) + h)/2 \tag{89}$$

As $g$ is increasing,

$$\lim_{t\to\infty} {}_|x(t) = \theta_2 \tag{90}$$

Thus $\theta_2 = h$. From that follows that $\theta_1 = 2(x(t_\text{end}) - h)$. From the "half-life" property, we get

$$\frac{2(_| x(t_\text{end}) - h)}{1 + e^{g(t_{1/2} - t_\text{end})}} + h = (_| x(t_\text{end}) + h)/2 \tag{91}$$

From that, we get that $g(t_{1/2} - t_\text{end}) = \log(3)$.

To summarize, we need to find $g$ such that

$$g(0) = 0 \tag{92}$$

$$\dot{g}(0) = -\frac{2_| \dot{x}(t_\text{end})}{x(t_\text{end}) - h} \tag{93}$$

$$\ddot{g}(0) = 0 \tag{94}$$

$$g(t_{1/2} - t_\text{end}) = \log(3) \tag{95}$$

We impose $\ddot{g}(0) = 0$, so that $_| \ddot{x}(t_\text{end}) = 0$, which is always the case if there is a bounded motif before. If it is the only motif, then one of the property maps is fixed and not trained.

To make sure that $\ddot{g}(t) \geq 0$ for all $t \geq 0$ we define it as a piecewise function composed of three cubic and a straight line. Similar to our approach in Appendix D.1.3, we describe the two cubics using second derivatives. Thus $\ddot{g}$ is a piecewise linear function with knots at $0, t_1, t_2$ such that

$$\ddot{g}(0) = 0 \tag{96}$$

$$\ddot{g}(t_1) = v_1 \tag{97}$$

$$\ddot{g}(t_2) = 0 \tag{98}$$

$$\tag{99}$$

and $\ddot{g}(t) = 0$ for all $t \geq t_2$. The goal is to now find $t_1, t_2, v_1$ such that $g$ is increasing, concave, and $g(t_{1/2} - t_\text{end}) = \log(3)$.

$g$ is always going to be concave if $v_1 < 0$. However, to make sure it stays increasing, we need to have $\dot{g}(t_2) > 0$. This requires $\dot{g}(0) + (t_2 \times v_1)/2 > 0$. If $t_{1/2} - t_\text{end} < 1.5 \log(3)/\dot{g}(0)$ then we set $t_2 = t_{1/2} - t_\text{end}$ otherwise we set $t_2 = 1.5 \log(3)/\dot{g}(0)$. In both cases, we choose $t_1 = t_2/2$. We used a symbolic Python library `sympy` to arrive at the following conclusions. In the first case, we set $v_1$ as follows.

$$v_1 = 4\frac{-(t_{1/2} - t_\text{end})\dot{g}(0) + \log(3)}{(t_{1/2} - t_\text{end})^2} \tag{100}$$

In the second case, we set it as follows.

$$v_1 = v_1 = 4\frac{-(t_{1/2} - t_\text{end})\dot{g}(0) + \log(3)}{2 * (t_{1/2} - t_\text{end}) * (1.5 \log(3)/\dot{g}(0)) - (1.5 \log(3)/\dot{g}(0))^2} \tag{101}$$

From that, we calculate the coefficients of the cubics and the slope and intercept of the straight line.

$s_{+-h}$ This one is analogous to $s_{-+h}$.

# E  EXPERIMENTAL DETAILS

All experimental code can be found at `https://github.com/krzysztof-kacprzyk/SemanticODE`.

## E.1  DATASETS

**Pharmacokinetic model**   The pharmacokinetic dataset is based on the pharmacokinetic model developed by Woillard et al. (2011) to model the plasma concentration of Tacrolimus. This model

consists of a system of ODEs described below.

$$\frac{dC_{\text{depot}}}{dt} = -k_{\text{tr}}C_{\text{depot}} \tag{102}$$

$$\frac{dC_{\text{trans1}}}{dt} = k_{\text{tr}}C_{\text{depot}} - k_{\text{tr}}C_{\text{trans1}} \tag{103}$$

$$\frac{dC_{\text{trans2}}}{dt} = k_{\text{tr}}C_{\text{trans1}} - k_{\text{tr}}C_{\text{trans2}} \tag{104}$$

$$\frac{dC_{\text{trans3}}}{dt} = k_{\text{tr}}C_{\text{trans2}} - k_{\text{tr}}C_{\text{trans3}} \tag{105}$$

$$\frac{dC_{\text{cent}}}{dt} = k_{\text{tr}}C_{\text{trans3}} - ((CL + Q) * C_{\text{cent}}/V_1) + (Q * C_{\text{peri}}/V_2) \tag{106}$$

$$\frac{dC_{\text{peri}}}{dt} = (Q * C_{\text{cent}}/V_1) - (Q * C_{\text{peri}}/V_2) \tag{107}$$

The values of the parameters and the initial conditions are presented in Table 10.

Table 10: Parameters of the pharmacokinetic dataset.

| Parameter | Value |
|---|---|
| $CL$ | 80.247 |
| $V1$ | 486.0 |
| $Q$ | 79 |
| $V2$ | 271 |
| $k_{\text{tr}}$ | 3.34 |
| $C_{\text{depot}}(t_0)$ | 10 |
| $C_{\text{cent}}(t_0)$ | $x_0 \times (V_1/1000)$ |
| $C_{\text{peri}}(t_0)$ | $V_2/V_1 \times C_{\text{cent}}(t_0)$ |
| $C_{\text{trans1}}(t_0)$ | 0.0 |
| $C_{\text{trans2}}(t_0)$ | 0.0 |
| $C_{\text{trans3}}(t_0)$ | 0.0 |

We create 100 samples of $x_0$ equally spaced between 0 and 20. We then solve the initial value problem to obtain a trajectory of $C_{\text{cent}}$. We then scale it back to appropriate units by multiplying by 1000 and dividing by $V_1$. We observe each trajectory at 20 equally spaced time points between 0 and 24. Then we scale the dataset by dividing the concentrations by 20 and the time points by 24. Finally, we add a Gaussian noise with a standard deviation $\sigma = 0.01$ for the low noise setting and $\sigma = 0.2$ for the high noise setting.

The out-domain dataset is created similarly, but each trajectory is observed only at $t_0 = 0$ and at 20 time points between 24 and 48 (1 and 2 after dividing by 24).

**Logistic growth** The logistic growth dataset is described by the following equation (Verhulst, 1845).

$$\dot{x}(t) = x(t)(1 - \frac{x(t)}{2}) \tag{108}$$

We create 200 samples of $x_0$ equally spaced between 0.2 and 4. We then solve the initial value problem to obtain a trajectory for each. We observe each trajectory at 20 equally spaced time points between 0 and 5. Finally, we add a Gaussian noise with a standard deviation $\sigma = 0.01$ for the low noise setting and $\sigma = 0.2$ for the high noise setting.

**General ODE** The general ODE dataset is described by the equation

$$\dot{x}(t) = f(x(t), t) \tag{109}$$

where $f$ is not described by a compact closed form expression. We describe $f$ using a probability density function of a 2-dimensional mixture of 3 Gaussians. The means, covariances, and weights for each Gaussian are shown in Table 11.

Table 11: Parameters of the Gaussians used to define the general ODE dataset.

| Weight | Mean | Covariance matrix |
|--------|------|-------------------|
| 0.4 | $[0, 0]$ | $[[1, 0], [0, 1]]$ |
| $-0.3$ | $[3, 3]$ | $[[1, 0.0], [0.0, 1]]$ |
| 0.3 | $[-1, -2]$ | $[[2, 0], [0, 2]]$ |

We then define $f(x, t)$ as the value of the probability density function of this mixture multiplied by 50. We create 200 samples of $x_0$ equally spaced between $-3$ and 3. We then solve the initial value problem to obtain a trajectory for each. We observe each trajectory at 20 equally spaced time points between 0 and 5. Then we scale the dataset by dividing $x$ by 3 and the time points by 5. Finally, we add a Gaussian noise with a standard deviation $\sigma = 0.01$ for the low noise setting and $\sigma = 0.2$ for the high noise setting.

**Mackey-Glass** The Mackey-Glass dataset is described by the following Mackey-Glass equation (Mackey & Glass, 1977).

$$\dot{x}(t) = \frac{\beta_0 \theta^n x(t - \tau)}{\theta^n + x(t - \tau)^n} - \gamma x(t) \tag{110}$$

We choose the following parameters: $\theta = 1$, $\beta = 0.4$, $\tau = 4.0$, $n = 4$. We create 200 samples of $x_0$ equally spaced between 1.0 and 3.0. We then solve the initial value problem to obtain a trajectory for each. We observe each trajectory at 20 equally spaced time points between 0 and 30. Then we scale the dataset by dividing $x$ by 3 and the time points by 30. Finally, we add a Gaussian noise with a standard deviation $\sigma = 0.01$ for the low noise setting and $\sigma = 0.2$ for the high noise setting.

**Integro-DE** The integro-differential equation dataset is described by the following equation (Holt et al., 2022; Bourne, 2018).

$$\dot{x}(t) = -2x(t) - 5 \int_0^t x(s) ds \tag{111}$$

We create 100 samples of $x_0$ equally spaced between $-1.0$ and 1.0. We then solve the initial value problem to obtain a trajectory for each. We observe each trajectory at 20 equally spaced time points between 0 and 5. Finally, we add a Gaussian noise with a standard deviation $\sigma = 0.01$ for the low noise setting and $\sigma = 0.2$ for the high noise setting.

### E.2 METHODS

**SINDy** We use SINDy (Brunton et al., 2016b) as implemented in the PySINDy package (de Silva et al., 2020; Kaptanoglu et al., 2022). We pass the variable $t$ as an additional dimension of the trajectory to allow for a time-dependent solution (not just autonomous systems). We use the following library of functions:

$$1, x, t, x^2, xt, t^2, e^x, e^t, \sin(x), \sin(t), \cos(x), \cos(t),$$
$$\sin(2x), \sin(2t), \cos(2x), \cos(2t), \sin(3x), \sin(3t)$$

We use Mixed-Integer Optimized Sparse Regression (MIOSR) (Bertsimas & Gurnee, 2023) for optimization as it allows us to choose a sparsity level—the number of terms in the equation. In our experiments, we consider two variants of SINDy, that we denote SINDy and SINDy-5. SINDy-5 enforces the maximum number of terms to be 5 which would hopefully allow for analyzing the equation. In the SINDy variant, we choose the maximum number of terms during hyperparameter tuning (between 1 and 20). For both SINDy and SINDy-5, we tune the parameter $\alpha$ of MIOSR that describes the strength of L2 penalty (between $1e - 3$ and 1), and the derivative estimation algorithm. We choose between the following techniques: finite difference, spline, trend filtered, and smoothed finite difference as available in PySINDy. The parameter ranges we consider for each of them are shown in Table 12.

**WSINDy** We use WSINDy (Reinbold et al., 2020; Messenger & Bortz, 2021a) based on (Reinbold et al., 2020) as implemented in PySINDy package (de Silva et al., 2020; Kaptanoglu et al., 2022). We

Table 12: Hyperparameter ranges for each of the derivative estimation methods.

| Method | Hyperparameter ranges |
|---|---|
| finite difference | $k \in \{1, \ldots, 5\}$ |
| spline | $s \in (1e-3, 1)$ |
| trend filtered | $\text{order} \in \{0, 1, 2\}, \alpha \in (1e-4, 1)$ |
| smoothed finite difference | $\text{window\_length} \in \{1, \ldots, 5\}$ |

pass the variable $t$ as an additional dimension of the trajectory to allow for a time-dependent solution (not just autonomous systems). We use the following library of functions:

$$1, x, t, x^2, xt, t^2, e^x, e^t, \sin(x), \sin(t), \cos(x), \cos(t),$$
$$\sin(2x), \sin(2t), \cos(2x), \cos(2t), \sin(3x), \sin(3t)$$

We use Mixed-Integer Optimized Sparse Regression (MIOSR) (Bertsimas & Gurnee, 2023) for optimization as it allows us to choose a sparsity level—the number of terms in the equation. In our experiments, we consider two variants of WSINDy, that we denote WSINDy and WSINDy-5. WSINDy-5 enforces the maximum number of terms to be 5 which would hopefully allow for analyzing the equation. In the WSINDy variant, we choose the maximum number of terms during hyperparameter tuning (between 1 and 20). For both WSINDy and WSINDy-5, we tune the parameter $\alpha$ of MIOSR that describes the strength of the L2 penalty (between $1e-3$ and 1). We choose the parameter $K$ (the number of domain centers) to be 200.

**PySR** We adapt PySR (Cranmer, 2020), a well-known symbolic regression method to ODE discovery by first estimating the derivative and then treating it as a label. We choose the derivative estimation technique and its parameters by hyperparameter tuning (as with SINDy) using the methods and parameter ranges in Table 12. We use the following operators and functions.

$$+, -, \times, \div, \sin, \exp, \log(1 + |x|)$$

Following the advice in the documentation, we also put a constraint to prevent nesting of sin functions. During hyperparameter tuning we allow PySR to search for 15 seconds and then we train the final model for 1 minute. This is a bit larger but comparable time budget to Sematic ODE and a much bigger time budget than SINDy and WSINDy require.

**NeuralODE** We implement a NeuralODE (Chen et al., 2018) model using `torchdiffeq` library. We parametrize the ODE as a fully connected neural network. The data is standardized before fitting. We set the batch size to 32 and train for 200 epochs using Adam optimizer (Kingma & Ba, 2017). We tune hyperparameters using `Optuna` (Akiba et al., 2019) for 20 trials. Ranges for the hyperparameters are shown in Table 13.

Table 13: Hyperparameter ranges used for tuning Neural ODE.

| Hyperparameter | Range |
|---|---|
| learning rate | (1e-5,1e-1) |
| number of layers | (1,3) |
| units in each layer (separately) | (16,128) |
| dropout rate | (0.0,0.5) |
| weight decay | (1e-6,1e-2) |
| activation function | ELU, Sigmoid |

**DeepONet** We implement DeepONet (Lu et al., 2020) using a fully connected neural network. The data is standardized before fitting. We train for 200 epochs using Adam optimizer (Kingma & Ba, 2017). We tune hyperparameters using `Optuna` (Akiba et al., 2019) for 20 trials. Ranges for the hyperparameters are shown in Table 14.

Table 14: Hyperparameter ranges used for tuning DeepONet.

| Hyperparameter | Range |
|---|---|
| learning rate | (1e-5,1e-1) |
| number of layers | (1,5) |
| number of hidden states | (10,100) |
| dropout rate | (0.0,0.5) |
| weight decay | (1e-6,1e-2) |
| batch size | $\{8, 16, 32\}$ |

**DeepONet** We implement Neural Laplace (Holt et al., 2022) using a fully connected neural network. The data is standardized before fitting. We train for 200 epochs using Adam optimizer (Kingma & Ba, 2017). We tune hyperparameters using `Optuna` (Akiba et al., 2019) for 20 trials. Ranges for the hyperparameters are shown in Table 15.

Table 15: Hyperparameter ranges used for tuning Neural Laplace.

| Hyperparameter | Range |
|---|---|
| learning rate | (1e-5,1e-1) |
| number of layers | (2,5) |
| number of hidden states | (10,100) |
| latent dimension | (2,10) |
| dropout rate | (0.0,0.5) |
| weight decay | (1e-6,1e-2) |
| batch size | $\{8, 16, 32\}$ |

**Semantic ODE** In all experiments in Section 6.4 we use a full composition library containing all compositions up to 3 motifs (with the exception of logistic growth dataset where we only consider compositions up to 2 motifs). We choose the maximum number of branches for the composition map to be $I = 3$. Each univariate function in the property maps is described as a linear combination of 6 basis functions: constant, linear, and four B-Spline basis functions of degree 3. The property maps are trained using L-BFGS as implemented in PyTorch. We fix the penalty term for the difference between derivatives to be $0.01$ and we perform hyperparameter tuning of each property sub-map to find the optimal learning rate (between 1e-4 and 1.0) and the penalty term for the first derivative at the last transition point (between 1e-9 and 1e-1).

### E.3 BENCHMARKING PROCEDURE

The experimental results presented in Table 3 are a result of the following procedure. For 5 different seeds, the dataset is randomly split into training, validation, and test datasets with ratios $0.7 : 0.15 : 0.15$. For each method, we perform hyperparameter tuning for 20 trials and then report the performance on the test set. Each seed results in both a different split and a different random state of the algorithm (apart from SINDy, which is deterministic). As Semantic ODE splits the training dataset and trains separate property sub-maps, we pass the combined train and validation set to it. For each property sub-map, the subset of this set with the corresponding composition is once again split into training and validation subsets. The validation set is used for hyperparameter tuning and for early stopping.

## F    EXTENDED RELATED WORKS

**Symbolic regression and discovery of differential equations** The discovery of differential equations is usually considered a part of a broader area called symbolic regression. Symbolic regression is the area of machine learning whose task is to describe data using a closed-form expression. Traditional symbolic regression has used genetic programming (Stephens, 2022; Cranmer, 2020) for this take but recently neural network has also been utilized for that task. That includes representing the equation

directly as a neural network by adapting the activation functions (Martius & Lampert, 2017; Sahoo et al., 2018), using neural networks to prune the search space (Udrescu & Tegmark, 2020; Udrescu et al., 2021), searching for equations using reinforcement learning (Petersen et al., 2021), or using large pre-trained transformers (Biggio et al., 2021; D'Ascoli et al., 2022). Designing complexity metrics and constraints that make sure the equations are simple enough to analyze is itself a challenging research problem (Kacprzyk & van der Schaar, 2025). Standard symbolic regression can be adapted to ODE discovery by just estimating the derivative from data and treating it as a target (Quade et al., 2016). However, many dedicated ODE discovery techniques have been proposed. Among them, the most popular is SINDy (Brunton et al., 2016b) that describes the derivative as a linear combination of functions from a prespecified library. This was followed by numerous extensions, including implicit equations (Kaheman et al., 2020), equations with control (Brunton et al., 2016b), and longitudinal treatment effect estimation (Kacprzyk et al., 2024a). Approaches based on weak formulation of ODEs that allow to circumvent derivative estimation have also been proposed (Messenger & Bortz, 2021a; Qian et al., 2022). ODE discovery methods usually cannot be directly used to discover partial differential equations (PDEs) and thus many dedicated PDE discovery algorithms have been developed (Rudy et al., 2017; Raissi & Karniadakis, 2018; Messenger & Bortz, 2021b; Kacprzyk et al., 2023). The challenge of finding compact and well-fitting closed-form expressions inspired Shape Arithmetic Expressions (Kacprzyk & van der Schaar, 2024) that extend the prespecified set of well-known functions (e.g., $\exp$ or trigonometric functions) in symbolic regression by flexible and learnable univariate functions that do not have a compact symbolic representation but can be comprehended by looking at their graph.

**Time series representation** Semantic representation is closely related to the topic of time series representation. In particular, apart from the work by Kacprzyk et al. (2024b), there have been other works that try to symbolically describe the time series. In particular, Symbolic Aggregate approXimation (Lonardi & Patel, 2002), Shape Description Alphabet (André-Jönsson & Badal, 1997), or the triangular representation of process trends (Cheung & Stephanopoulos, 1990) represent a time series as a sequence of symbols that resembles the definition of a composition. However, these methods are mostly used for data mining or classification rather than forecasting. Similarly, although shapelet-based methods (Ye & Keogh, 2009) and motif discovery (Torkamani & Lohweg, 2017) are concerned with the trajectory's shape, they are usually aimed at finding subsequences of a time series that represent the most important or repeating patterns. There are also numerous other nonsymbolic time series representation techniques ranging from Piecewise Aggregate Approximation (Yi & Faloutsos, 2000) to signatures (Lyons, 2014).

**Shape preserving splines** As we describe the bounded part of our trajectory as a cubic spline, and we want it to have a specific shape, this may seem related to an area of machine learning called shape-preserving splines. Its main goal is to interpolate data while maintaining essential characteristics such as monotonicity, convexity, or non-negativity (Fritsch & Carlson, 1980; Pruess, 1993). Although their goal is to interpolate rather than reconstruct, and they rarely consider the exact positions of the local extrema and inflection points, ideas from this field may prove useful in designing better and more efficient trajectory predictors in the future.

**Neural ODEs** Neural ODEs (Chen et al., 2018) provide a flexible way to model continuous dynamical systems by parameterizing the derivative function of an ODE with a neural network. While this allows Neural ODEs to capture complex system behaviors, they often operate as black-box models, making the task of analyzing to obtain the semantic representation even more difficult than in the case of closed-form ODEs.

# G ADDITIONAL DISCUSSION

## G.1 LIMITATIONS

**Finite compositions** Throughout our work we implicitly assume that the compositions are finite. However, that is not always the case. In particular, any trajectory that has an oscillatory/periodic behavior has an infinite composition. For instance, $\sin$. Although Semantic ODE can fit such a trajectory on any bounded interval, it cannot yet predict it into the future. This could possibly be addressed by extending the definition of semantic representation by an additional layer that

describes periodic trajectories. The idea is to take the infinite composition (infinite sequence of motifs) and represent it using some finite representation. For instance, a trajectory $\sin(t)$ would be described as a *meta-motif* $(s_{+-c}, s_{--c}, s_{-+c}, s_{++c})$ repeating forever, where the *meta-properties* may include the "frequency" and "amplitude" that may also vary with time and be itself described using compositions. This extension still assumes there is some underlying pattern behind the infinite sequence of motifs that can be "compressed" into a shorter representation. If the infinite sequence of motifs is algorithmically random, then motif-based representation may not be useful.

**Long compositions**    The current implementation may struggle if the ground-truth composition is finite but long. In that case, it may fall outside of the chosen set of compositions.

**Chaotic systems**    Chaotic systems usually have some kind of oscillatory behavior (i.e., it cannot be described by a finite composition). As discussed above, it means that chaotic systems are currently beyond the capabilities of Semantic ODEs. They would not be able to correctly predict beyond the seen time domain. However, we could use it for a prediction on a bounded time domain. We have included such an experiment in Appendix B.6. Although our method achieves decent results, this performance would drop drastically if we tried to predict beyond the training time domain. In general, Semantic ODE can model big changes in the trajectory following a small change in the initial condition (as is characteristic for chaotic systems). As the composition map is discontinuous at points where the composition changes, it can model a sudden change in the shape of the trajectory.

**Training of the composition map**    The composition map determines how the dataset is split and which property maps are trained. Thus, it is very important to do it well. Currently, we evaluate how well a particular composition fits a trajectory individually for every sample. However, then we fit property sub-maps where these properties vary smoothly between samples. Thu,s the neighboring samples should be taken into consideration when evaluating how well a particular composition fits a trajectory. If the same composition fits two neighboring samples well but for very different property values, this may not be a good match.

**Ubounded motifs**    The type of unbounded motifs we choose and how we decide to parameterize them influences how well we can fit trajectories. For instance, we describe $s_{++u}$ as behaving like an exponential function in the long term. This may not work well if the actual trajectory behaves like a quadratic function, for instance. To address it, we should have more motifs available, so that we can be more certain that we can find a well-fitting motif. However, this also makes training the composition map more difficult. This emphasizes why finding an effective and efficient training procedure for the composition map is essential.

**One-dimensional systems**    The current implementation of Semantic ODE works only for one-dimensional systems. We describe a possible roadmap for realizing direct semantic modeling for multi-dimensional systems in Appendix G.2.

### G.2    ROADMAP FOR DIRECT SEMANTIC MODELING IN MULTIPLE DIMENSIONS

Although Semantic ODE realizes direct semantic modeling for one-dimensional trajectories, we believe direct semantic modeling can be successfully applied to trajectories with multiple dimensions. To realize that, we first need to allow for multi-dimensional inputs to the semantic predictor. As the composition map is just a classification algorithm and the property maps are just sets of static regression models, this should be possible. We could imagine a composition map represented as a decision tree and the properties being predicted by, for instance, generalized additive models (GAMs) (Hastie & Tibshirani, 1986). In that way, we can still have an understandable model representation. The major challenge is to come up with an efficient optimization procedure that would not only search this complex space but also make sure that every predicted semantic representation is indeed valid. An additional challenge is to keep the model in an appropriate form and design an interface to edit the property maps as it is done in a Semantic ODE.

The easiest way to model multiple trajectories (or multidimensional trajectories) is to model each trajectory dimension independently but conditioned on the multidimensional initial condition $x_0 \in \mathbb{R}^M$, i.e., for $M$ trajectories, we want to fit $M$ forecasting models, $F_1, \ldots, F_M$ such that for each

$m \in [M]$, $F_m : \mathbb{R}^M \to C^2$. Each $F_m$ can still be represented as $F_{\text{traj}} \circ F_{\text{sem}}^{(m)}$, where $F_{\text{traj}}$ is the same as described in the paper and $F_{\text{sem}}^m : \mathbb{R}^M \to \mathcal{C} \times \mathcal{P}$. $F_{sem}^m$ can similarly be represented using a composition map and property maps for each predicted composition. A proof of concept of this approach is demonstrated in Appendix B.5. This approach should scale to multiple dimensions as it is very modular. Each $F_{\text{sem}}^m$ can be analyzed independently. Each property map of $F_{\text{sem}}^m$ corresponding to a particular composition can be analyzed independently. And finally, each shape function for each of the predicted properties can be analyzed independently. We believe this modularity is one of the reasons why even systems with multiple dimensions can remain understandable and, more importantly, can be edited as each change has a localized impact. In contrast, changing a single parameter in a system of ODEs may result in a change of all the trajectories in ways that may be difficult to predict without a careful analysis.

Another extension would be to model some dimensions jointly. For instance, we could try to characterize the shape of a two-dimensional curve. This would require extending the current definition of semantic representation.

**Computational cost**   The computational cost of extending semantic modeling to $M$ dimensions depends on the exact approach taken. We will try to do our best guess assuming the approach remains as similar as possible to the one we describe in the paper.

With our current ideas, the added cost from having multiple dimensions comes mostly from the need to perform the same tasks $M$ times. For $M$ trajectories, we need to fit $M$ composition maps and $M$ property maps.

We do not expect the time to fit a single composition map to increase significantly. The main computational burden of the current implementation comes from fitting a composition to each sample to see how well it fits. For the same number of samples, the computation cost is going to be the same. Although there will be an added cost to find optimal decision boundaries in multiple dimensions (as opposed to one), we do not think this will be a significant burden given current efficient classification algorithms.

Fitting of each property map will require $M$ times the number of parameters. However, it is important to note that in most cases, the current property maps have just tens of parameters, so this number is still likely to be a few hundred at most. It is possible that the number of property maps needed to be fitted (for each of the compositions) is going to be larger. However, a user would still like to narrow it down to a reasonable number (likely less than 10) to make sure that the composition map is understandable.

Overall, we suspect the training time to increase at least linearly with the number of dimensions, but we do not expect any combinatorial explosion as often happens in symbolic regression, so for the number of trajectories we may be interested (probably less than 10), the computational costs should not be a major concern.

### G.3   GRANULARITY OF INFORMATION

We believe the granularity of information (in terms of the shape of the trajectory, its maxima, minima, and asymptotic behavior) should be comparable between direct semantic modeling and other approaches, as long as all approaches are framed as solutions to finding a forecasting model. The key difference is that direct semantic modeling gives immediate access to the semantic representation of the model without any further analysis and allows us to easily edit the model as well as incorporate semantic inductive biases during training.

There is, however, one way in which ODEs *may* offer more granular information by assuming a particular causal structure. *Note*: it may not be correct, and it makes them less flexible. For a discretized ODE we can write the underlying causal graph as $x(t_0) \to x(t_0 + \Delta t) \to \ldots \to x(t)$. Whereas our model assumes a more general $x(t_0) \to x(t)$. This generality is useful (as we show in Section 6.4) and it does not negatively impact the performance. However, this model does not allow us to do interventions where we change the value of $x$ at a particular $t$ and predict the future. ODEs do allow for this kind of inference but, of course, there is no guarantee that this inference is correct. For instance, a system governed by a delayed differential equation has a different causal structure.

We believe direct semantic modeling in the future can be extended to accommodate some kinds of intervention, where they can be described as additional features beyond the initial conditions.

## G.4 Partial Differential Equations

Partial differential equations (PDEs) no longer describe trajectories (a function defined on the real line) but rather *fields* which are defined on multidimensional surfaces such as planes. Extending direct semantic modeling to PDEs requires developing a new definition of the semantic representation of a *field*. Moreover, as the initial condition is no longer a single number, the semantic predictor needs to take whole functions as inputs (for both initial and boundary conditions).

## G.5 Flexibility of motif-based semantic representation

In this section, we will prove the following representation theorem.

**Theorem 3.** *Any function $x \in C^2(t_0, +\infty)$ whose second derivative vanishes at a finite number of points (i.e., $|\{t \; : \; \ddot{x}(t) = 0\}| < \infty$) can be described as a finite composition constructed from proposed motifs.*

*Proof.* We call all points in $\{t \; : \; \ddot{x}(t) = 0\}$ the vanishing points. Let $i_{\text{first}}$ and $i_{\text{last}}$ be the smallest and largest $t$ such that $\ddot{x}(t) = 0$. As $\ddot{x}$ vanishes only a finite number of times, such numbers exist. $\ddot{x}$ is positive or negative for all points in between consecutive vanishing points as well as for all $t < i_{\text{first}}$ and $t > i_{\text{last}}$. That means $\dot{x}$ is either strictly increasing or strictly decreasing in between consecutive vanishing points as well as for all $t < i_{\text{first}}$ and $t > i_{\text{last}}$. That means there is, at most, one point between consecutive vanishing points where $\dot{x} = 0$. Similarly for $t < i_{\text{first}}$ and $t > i_{\text{last}}$. That means there is a finite number of points where $\dot{x}$ vanishes. we call such points extrema and denote the smallest and largest extremum as $e_{\text{first}}$ and $e_{\text{last}}$. For all points between consecutive extrema $\dot{x}$ is either positive or negative (similarly for $t < e_{\text{first}}$ and $t > e_{\text{last}}$). That means that $x$ is either strictly increasing or decreasing between consecutive extrema (similarly for $t < e_{\text{first}}$ and $t > e_{\text{last}}$). We can now take the union of all vanishing points and extrema and divide $(t_0, +\infty)$ into intervals such that on each interval, $x$ is either strictly increasing or decreasing and is either convex or concave. Thus we can assign to each bounded interval one of the proposed bounded motifs. In the final (unbounded) interval, $x$ can be strictly increasing and convex ($s_{++u}$) or strictly decreasing and concave ($s_{--u}$). If it is strictly increasing and concave, then either it approaches a real number ($s_{+-h}$) or diverges to infinity ($s_{+-u}$). If it is strictly decreasing and convex, then either it approaches a real number ($s_{-+h}$) or diverges to infinity ($s_{-+u}$). Thus we are able to describe the whole trajectory as a finite composition constructed from proposed motifs. $\qquad\square$

## G.6 Sharing samples between property maps

Each $F_{\text{prop}}$ is trained separately for each composition. They do not share any information. However, we believe some amount of sharing may improve performance where the two adjacent compositions are similar, and the exact boundary is uncertain. We consider this an interesting extension for future work.

