# OpenReview forum: "No Equations Needed: Learning System Dynamics Without Relying on Closed-Form ODEs"
_ICLR.cc/2025/Conference — ICLR 2025 Poster_

### Official Review · Reviewer_BR8f · 2024-10-31

**Soundness:** 3
**Presentation:** 3
**Contribution:** 2
**Rating:** 6
**Confidence:** 3

**Summary:**

The authors propose direct semantic modeling of low-dimensional dynamical systems, which shifts away from the traditional two-step pipeline composed of first discovering a closed-form equation and then analyzing it.  The framework allows the incorporation of intuitive inductive biases into the optimization algorithm and the editing of the model’s behavior directly from data. The effectiveness of this framework is validated by extensive experiments including the logistic growth model, systems governed by a general differential equation, pharmacokinetic model, delay differential equation, and the integro-differential equation.

**Strengths:**

1. A novel approach, called direct semantic modeling, is proposed to learn the semantic representation of the dynamical system directly from data, eliminating the need for post-mathematical analysis.

2. The method operates directly on the semantic representation, enables intuitive adjustments, and seamlessly incorporates constraints that mirror the system’s operational behavior, which enhances the modeling flexibility and boosts performance, as it bypasses the need for a concise closed-form equation.

**Weaknesses:**

1. Compared to methods that rely on closed-form solutions, the interpretability and generalizability of direct semantic modeling approaches appear to be less apparent. As the author mentioned in the main text: " The primary objective of discovering a closed-form ODE, as opposed to using a black-box model, is to have a model representation that can be analyzed by humans to understand its behavior. Such understanding is necessary to ensure that the model behaves as expected.".

2. The author has showcased the method’s efficacy through a variety of examples of 1D ODE, such as the logistic growth model, systems governed by a general differential equation, pharmacokinetic model, delay differential equation, and the integro-differential equation, which is commendable. However, the lack of demonstration of more complex data such as experimental data with unknown dynamical equations may diminish its appeal to a broader audience and contribution.

3. Identifying transition points (critical points) in dynamical systems that exhibit bifurcation behavior is inherently challenging, which may constrain the application of the framework to such systems, as their semantic representations tend to be more complex.

**Questions:**

1. Please elaborate on the advantages and disadvantages of direct semantic modeling compared to closed-form solutions in terms of model interpretability and generalizability.

2. If possible, please demonstrate the model’s effectiveness on more complex data, such as the semantic representation learning and prediction of time series in experiments.

3. How to efficiently identify bifurcation points, perform semantic modeling, and predict bifurcation behavior within dynamical systems using this framework when bifurcation behavior is present？

4. If possible, the author could include a discussion on handling partial differential equations with this framework, noting that in addition to initial value problems, there are also boundary value problems.

5. By the way, there are some minor typing mistakes in the article. On the second page, the differential operator symbol ‘d’ should be set in upright type.

---

> ### Author Response · Authors · 2024-11-21
> **Response to Reviewer BR8f [1/2]**
>
> Dear Reviewer BR8f,
>
> Thank you very much for your review. We are glad that you found our approach to be novel. By addressing your questions and concerns, we truly believe we have strengthened our paper. We provide our responses below and point to relevant changes in the paper.
>
> ---
>
> ### (A) Interpretability and generalizability
>
> **Interpretability**
>
> The main advantage of our approach in terms of interpretability is the fact that we do not have to perform any mathematical analysis to obtain a semantic representation of the model (as it is defined in the paper). Obtaining the same information from a closed-form ODE is often possible (if the equation is not too complex) but may be time-consuming or require mathematical expertise.
>
> **Generalizability**
>
> Closed-form ODEs may generalize very well if the discovered equation is the correct one. However, finding the correct equation may be hard or sometimes even impossible if the phenomenon is not governed by an ODE. For instance, the dynamics may depend on history (e.g., delay and integral differential equations), involve unobserved variables (e.g., pharmacological model), or empirically determined functions (e.g., stress-strain curve). In Section 6.2, we showed how ODEs may fail to generalize outside of the training time domain and how, by editing and incorporating semantic inductive biases, we can increase the generalizability of Semantic ODE to the unseen time domain.
>
> We have included another experiment in Appendix B.10, showcasing how we can generalize the model in Section 6.2 to initial conditions $(x_0)$ not seen during training. We observe that each property function in Figure 8 looks approximately like a linear function. Thus, we fit a linear function to each of these functions and then evaluate our model on initial conditions from range $(1.0, 1.5)$. Note that our training set only contained initial conditions from $(0, 1)$. We compare the performance of this model to ODEs from Table 2. The results can be seen below. Our model has suffered only a small drop in performance even though it has never seen a single sample from that distribution. It also performs much better than any other ODE tested.
>
> | Model | $x_0 \in (0,1)$ | $x_0 \in (1,1.5)$ |
> | --- | --- | --- |
> | SINDy | $0.222_{(0.041)}$ | $0.240_{(0.035)}$ |
> | SINDy | $0.112_{(0.027)}$ | $0.131_{(0.025)}$ |
> | SINDy | $0.101_{(0.023)}$ | $7.764_{(4.938)}$ |
> | SINDy |$0.029_{(0.005)}$ | $0.105_{(0.056)}$ |
> | SINDy | $0.020_{(0.004)}$ | $0.203_{(0.430)}$ |
> | Semantic ODE | **$0.018_{(0.003)}$** | **$0.023_{(0.005)}$** |
>
> ---
>
> ### (B) Experiments on real datasets
> Thank you for your suggestion regarding additional experiments. We have now added experiments on two real datasets (on tumor growth and drug concentration). We added these results to Appendix B.8 and reproduced them below. "Semantic ODE*" is a variant of Semantic ODE where we incorporated a semantic inductive bias about the shape of the trajectory. We specified the composition to always be $(s_{-+c},s_{++u})$ for the tumor growth dataset and $(s_{+-c},s_{--c},s_{-+h})$ for the drug concentration dataset. Note that the implementation of WSINDy we used cannot work with trajectories as sparse as the drug concentration dataset.
>
> | | Tumor growth (real) | Drug concentration (real) |
> | --- | --- | --- |
> | SINDy-5 | $0.243_{(.019)}$ | $0.286_{(.021)}$ |
> | WSINDy-5 | $0.237_{(.015)}$ | N/A |
> | PySR-20 | $0.536_{(.346)}$ | $0.257_{(.022)}$ |
> | SINDy | $0.249_{(.029)}$ | $0.286_{(.014)}$ |
> | WSINDy | $0.236_{(.016)}$ | N/A |
> | Neural ODE | $0.228_{(.018)}$ | $0.263_{(.032)}$ |
> | Neural Laplace | $0.243_{(.029)}$ | $0.302_{(.022)}$ |
> | DeepONet | $0.242_{(.016)}$ | $0.265_{(.020)}$ |
> | Semantic ODE | $0.234_{(.019)}$ | $0.264_{(.021)}$ |
> | Semantic ODE* | $0.229_{(.019)}$ | $0.243_{(.015)}$ |

---

> ### Author Response · Authors · 2024-11-21
> **Response to Reviewer BR8f [2/2]**
>
> ### (C) Bifurcations
> We believe that our framework is uniquely positioned to perform quite well on systems exhibiting bifurcations (when a small change to the parameter value causes a sudden qualitative change in the system's behavior). In our framework, bifurcation occurs when the composition map predicts a different composition. As discussed (now more thoroughly) in Appendix G.1, in the future, the semantic predictor may take as input not only the initial conditions but also other auxiliary parameters. We can then represent the composition map as a decision tree that divides the input space into different compositions. This decision tree then informs us where bifurcations occur.
>
> We hope the following proof of concept based on the current implementation demonstrates that it is a viable approach. Instead of predicting a trajectory from its initial condition, we fix the initial condition to be always the same and predict a trajectory based on the parameter $r$ that we observe in our dataset. We generate the trajectories given the following differential equation
> $$ \dot{x} = rx - x^2 $$
> the initial condition $x(0)=1$ and $r$ sampled uniformly from $(-1,2)$. We choose the set of compositions to be $(s_{+-h}),(s_{-+h})$ and record the position of the bifurcation point found by our algorithm (as opposed to the ground truth). The mean absolute error for different noise settings can be seen in Appendix B.9 and the table below. Note that the range of values of the trajectory is $(0,2)$, and even in high noise settings, the location of the bifurcation point can be identified.
>
> | Noise level | Mean Absolute Error (Std) |
> |---|---|
> | 0.001 | 0.001 (0.002) |
> | 0.010 | 0.002 (0.003) |
> | 0.050 | 0.009 (0.007) |
> | 0.100 | 0.012 (0.012) |
> | 0.200 | 0.017 (0.010) |
> | 0.500 | 0.023 (0.018) |
> | 1.000 | 0.040 (0.026) |
> | 1.500 | 0.043 (0.031) |
> | 2.000 | 0.043 (0.031) |
>
> ---
>
> ### Questions
>
> **Q1**:  Please see our response in "Interpretability and generalizability"
>
> **Q2**: Please see our response in "Experiments on real datasets".
>
> **Q3**: Please see our response in "Bifurcations".
>
> **Q4**: Partial differential equations (PDEs) no longer describe trajectories (a function defined on the real line) but rather *fields* that are defined on multi-dimensional surfaces such as planes. Extending direct semantic modeling to PDEs requires developing a new definition of semantic representation of a *field*. Moreover, as the initial condition is no longer a single number, the semantic predictor needs to take whole functions as inputs (for both initial and boundary conditions). We have added this discussion to Appendix G.4.
>
> **Q5**: We have now updated all differential operator symbols to upright type.

---

> > ### Author Response · Authors · 2024-11-25
> >
> > Dear Reviewer BR8f,
> >
> > Thank you once again for your thoughtful feedback, which has significantly improved our paper. We have carefully addressed your comments in our rebuttal and hope you find our responses satisfactory.
> >
> > As the discussion period is nearing its end, we wanted to check if you have any remaining concerns or questions we can address. If our revisions meet your expectations, we hope you might consider increasing your score.
> >
> > We truly appreciate your time and invaluable insights!
> >
> > Kind regards,
> >
> > Authors

---

### Official Review · Reviewer_DbZG · 2024-11-01

**Soundness:** 2
**Presentation:** 3
**Contribution:** 4
**Rating:** 6
**Confidence:** 3

**Summary:**

This paper proposes a direct semantic modeling method, which is fundamentally different from the traditional two-step modeling, which first discovers the symbolic form of ODEs. This approach is evaluated through extensive experiments.

**Strengths:**

This algorithm is completely new and different from the previous works. The authors provide detailed definitions of many new concepts and provide many analyses of whether they are well-defined or not. With experiments on different tasks and equations, results show that this algorithm can outperform previous algorithms in many ways.

**Weaknesses:**

1. Note the capitalization of the title.
2. Except for the traditional two-step modeling methods, there are also direct modeling methods for ODE trajectory simulation, like FNO, DeepONet, and many others. Considering the comparisons between them and Semantic ODE will be meaningful.
3. This algorithm is quite novel in that it only provides semantics rather than closed-form expressions, but can you analyze more about what kinds of downstream tasks it has?
4. l86, l475: Lack of a period at the end of the title.
5.1210: (denoted $c x$)

**Questions:**

1. How to ensure that the training subset of $F_{prop}$, which coresponds to one composition, have enough data? lf the dataset is divided into many subsets because of maybe many different intervals or different motifs, can $F fpropl$ be trained well with small datasets?
2. Do different $F {propl$s of different compositions share weights? Or do you need to train as many $F fpropl$s as the compositions?
3. I wonder if training with $F^0_{traj}$ but inference with $F^1_{traj}$ will introduce misalignment and error of the output.

---

> ### Author Response · Authors · 2024-11-21
> **Response to Reviewer DbZG [1/2]**
>
> Dear Reviewer DbZG,
>
> Thank you very much for your review. We are glad you found our work to be completely new and different from previous approaches.  By addressing your questions and concerns, we truly believe we have strengthened our paper. We provide our responses below and point to relevant changes in the paper.
>
> ---
>
> ### (A) Capitalization of the title
> It seems to us that the current capitalization follows common capitalization guidelines, including AP Stylebook and APA Style. Please let us know if you believe we should change the capitalization of some words to make it consistent.
>
> ---
>
> ### (B) Comparison with black-box approaches
>
> We have now added a comparison with three black box approaches: NeuralODE [1], NeuralLaplace [2], and DeepONet [3]. All results are now included in Table 3 and can also be seen below. In nearly all settings, Semantic ODE outperforms the black-box method. This is likely because we operate in a low-data regime, where we always have fewer than 200 samples measured at most 20 times throughout the trajectory.
>
> | | Logistic Growth | Logistic Growth | General ODE | General ODE | Pharmacokinetic | Pharmacokinetic | Mackey-Glass | Mackey-Glass | Integro-DE | Integro-DE |
> | --- | --- | --- | --- | --- | --- | --- | --- | --- | --- | --- |
> | Noise Level | low | high | low | high | low | high | low | high | low | high |
> | Neural ODE | $0.023_{(.004)}$ | $0.197_{(.005)}$ | $0.029_{(.005)}$ | $0.075_{(.006)}$ | $0.036_{(.008)}$ | $0.203_{(.007)}$ | $0.177_{(.010)}$ | $0.194_{(.010)}$ | $0.073_{(.007)}$ | $0.215_{(.009)}$ |
> | Neural Laplace | $0.126_{(.036)}$ | $0.230_{(.017)}$ | $0.108_{(.030)}$ | $0.138_{(.023)}$ | $0.100_{(.022)}$ | $0.229_{(.013)}$ | $0.057_{(.006)}$ | $0.094_{(.009)}$ | $0.075_{(.044)}$ | $0.249_{(.014)}$ |
> | DeepONet | $0.184_{(.040)}$ | $0.306_{(.023)}$ | $0.160_{(.033)}$ | $0.195_{(.027)}$ | $0.058_{(.010)}$ | $0.212_{(.005)}$ | $0.107_{(.014)}$ | $0.132_{(.012)}$ | $0.100_{(.015)}$ | $0.230_{(.014)}$ |
> | Semantic ODE | $0.015_{(.005)}$ | $0.200_{(.009)}$ | $0.015_{(.001)}$ | $0.068_{(.002)}$ | $0.023_{(.014)}$ | $0.211_{(.015)}$ | $0.037_{(.003)}$ | $0.075_{(.003)}$ | $0.025_{(.003)}$ | $0.203_{(.007)}$ |
>
> ---
>
> ### (C) Downstream tasks
> The main difference between direct semantic modeling and traditional two-step modeling is the fact that the user can interact directly with the semantic representation of the model. That not only avoids (often demanding) the analysis process but, more importantly, allows for incorporating semantic inductive biases (Section 6.1) and model editing (Section 6.3), which can significantly increase its extrapolation properties to the unseen time domain (as shown in Section 6.3).
>
> We have included another experiment in Appendix B.10, showcasing how we can generalize the model in Section 6.2 to initial conditions $(x_0)$ not seen during training. We observe that each property function in Figure 8 looks approximately like a linear function. Thus, we fit a linear function to each of these functions and then evaluate our model on initial conditions from range $(1.0, 1.5)$. Note that our training set only contained initial conditions from $(0, 1)$. We compare the performance of this model to ODEs from Table 2. The results can be seen below. We can see that our model has suffered only a small drop in performance even though it has never seen a single sample from that distribution. It also performs much better than any other ODE tested.
>
> | Model | $x_0 \in (0,1)$ | $x_0 \in (1,1.5)$ |
> | --- | --- | --- |
> | SINDy | $0.222_{(0.041)}$ | $0.240_{(0.035)}$ |
> | SINDy | $0.112_{(0.027)}$ | $0.131_{(0.025)}$ |
> | SINDy | $0.101_{(0.023)}$ | $7.764_{(4.938)}$ |
> | SINDy |$0.029_{(0.005)}$ | $0.105_{(0.056)}$ |
> | SINDy | $0.020_{(0.004)}$ | $0.203_{(0.430)}$ |
> | Semantic ODE | **$0.018_{(0.003)}$** | **$0.023_{(0.005)}$** |

---

> ### Author Response · Authors · 2024-11-21
> **Response to Reviewer DbZG [2/2]**
>
> ### (D) Minor issues
> Thank you for spotting the typos. We have fixed them now.
>
> ---
>
> ### Questions
>
> **Q1**: Thank you for asking this question! We should have mentioned this in the implementation details. Indeed, in Algorithm 1, before we decide to switch from one composition to the other, we verify that both the number of samples and the interval length are above user-defined thresholds. In experiments, we require at least two samples and the interval length to be at least 10% of the size of the entire domain of the map. We have now included this in Appendix C.1. There is also a maximum number of splits decided by the user. In all experiments, we allow for at most 3 subsets/property maps. The property maps themselves are described as a linear combination of basis functions (each described by just a few parameters), so, in principle, they require fewer samples than models with a larger number of parameters.
>
> **Q2**: No, each $F_{\text{prop}}$ is trained separately for each composition. They do not share any information. However, we believe some amount of sharing may improve performance where the two adjacent compositions are similar, and the exact boundary is uncertain. We consider this an interesting extension for future work. We have included it in Appendix G.6.
>
> **Q3**: Before predicting with $F_{\text{traj}}^2$, we ensure that the predicted trajectory's transition points are at most $\epsilon$ from those dictated by the property map. If we cannot find such a trajectory, then we default to using $F_{\text{traj}}^0$. In our experiments, we choose $\epsilon=0.001$. In general, we consider $F_{\text{traj}}^0$ to be an approximation of $F_{\text{traj}}^2$ used for backpropagation. We also have a penalty term added to the training loss that encourages smoother trajectories so that finding a $\mathcal{C}^2$ trajectory is easier. We use $F_{\text{traj}}^2$ for both validation and test errors, so we do not expect any major issues associated with that.
>
> ### References
> [1] Chen, R. T., Rubanova, Y., Bettencourt, J., & Duvenaud, D. K. (2018). Neural ordinary differential equations.
>
> [2] Holt, S. I., Qian, Z., & van der Schaar, M. (2022). Neural laplace: Learning diverse classes of differential equations in the laplace domain.
>
> [3] Lu, L., Jin, P., & Karniadakis, G. E. (2019). Deeponet: Learning nonlinear operators for identifying differential equations based on the universal approximation theorem of operators.

---

> ### Author Response · Authors · 2024-11-26
>
> Dear Reviewer DbZG,
>
> Thank you once again for your thoughtful feedback, which has significantly improved our paper. We have carefully addressed your comments in our rebuttal and hope you find our responses satisfactory.
>
> We wanted to check if you have any remaining concerns or questions we can address. If our revisions meet your expectations, we hope you might consider increasing your score.
>
> We truly appreciate your time and invaluable insights!
>
> Kind regards,
>
> Authors

---

> > ### Comment · Reviewer_DbZG · 2024-11-27
> >
> > Thank the authors for the responses, the additional experiments, and the modifications to the manuscript. I think its innovation and the amount of work are sufficient for acceptance. However, as a new algorithm, there are practical details that remain unclear, which makes me skeptical about its performance. Therefore, I believe a score of 6 is appropriate.

---

> > > ### Author Response · Authors · 2024-11-28
> > >
> > > Dear Reviewer DbZG,
> > >
> > > Thank you so much for reading through our responses. We want to do our best to clarify the practical details of our algorithm. You can now see the whole codebase using this anonymous [link](https://anonymous.4open.science/r/semantic-odes/ ). We also describe the main components of our implementation below.
> > >
> > > The main interface is accessed through `SemanticODE` class in `api.py`. This class has methods such as `fit` and `predict` and can be used like many other implementations of ML methods.
> > >
> > > Analogously to our paper, the goal of running `fit` is to train an instance of `SemanticPredictor`, which is a wrapper around a `CompositionMap` and a list of `SinglePropertyMap`s. `SemanticPredictor` has a `predict` method that predicts an instance of `SemanticRepresentation`, which corresponds to a semantic representation of a trajectory (Definition 1). `SemanticODE`'s `predict` method then uses this semantic representation to predict the actual values of the trajectory using `PredictiveModel` or `ApproximatePredictiveModel` (from `reconstruct_cubic.py`) that correspond to trajectory predictors $F_{\text{traj}}^2$ and $F_{\text{traj}}^0$.
> > >
> > > `SemanticPredictor` is trained in two steps. First, we fit a composition map by computing scores for each sample and for each composition (`_compute_composition_scores_df`) and finding optimal branches using Algorithm 1 (`_solve_branching_problem`). Then for each composition in the fitted `CompositionMap`, we create a `pytorch` model (`CubicModel` in `model_torch.py`) and fit the functions for each of the properties.
> > >
> > > `CubicModel`'s `forward` method takes the initial measurement $x_0$ as well as the values of pre-specified basis functions evaluated on $x_0$. By doing so, we can quickly evaluate "raw properties" by matrix-vector multiplication. Then (as described in Appendix C.2) we transform these raw properties into $t$- and $x$-coordinates of the transition points (`extract_coordinates_finite_composition`), values of the derivatives and properties of the unbounded motif. The coordinates and derivatives are then used to extract the coefficients of the cubics in `get_coefficients_and_coordinates_finite_composition` and the properties of the unbounded motif are used by an appropriate class from `infinite_motifs.py` to predict the values of the trajectory beyond the last transition point. This allows us to calculate the loss in `loss` and minimize it using LBFGS. After training, `CubicModel` is used to define `SinglePropertyMap` through `_construct_single_property_map`.
> > >
> > > We hope the code and the synopsis above clarify the details of our implementation. Please let us know if there are any details of our method that remain unclear; we are more than happy to clarify them.
> > >
> > > Kind regards,
> > >
> > > Authors

---

> > > > ### Author Response · Authors · 2024-12-03
> > > >
> > > > Dear Reviewer DbZG,
> > > >
> > > > Thank you again for your valuable feedback. We hope the provided code and synopsis of the implementation address your earlier comment about unclear practical details and that the paper aligns more closely with the criteria for a higher rating. We would greatly appreciate clarification on any specific points that might still be unclear or ambiguous. Your insights would be invaluable in helping us refine our presentation to make our paper as clear as possible to the readers.
> > > >
> > > > Thank you for your time and efforts in reviewing our work and helping us improve our paper's clarity.
> > > >
> > > > Kind regards,
> > > >
> > > > Authors

---

### Official Review · Reviewer_DLBF · 2024-11-02

**Soundness:** 3
**Presentation:** 3
**Contribution:** 3
**Rating:** 6
**Confidence:** 5

**Summary:**

This work compares syntactic and semantic representations of ordinary differential equations. In the second representation, the authors introduce direct semantic modeling via predefined bounded, unbounded, and horizontal asymptote dynamic motifs. This data-driven method is demonstrated through a one-dimensional ordinary differential equation. A semantic predictor is trained via a classification-base composition predictor and a properties prediction via a set of univariate functions, and fed into a trajectory predictor, which outperforms the traditional two-step process of equation discovery and analysis in terms of prior knowledge injection, model editing, feedback mechanism, and noise resistance. The syntactic inductive bias due to the analytic and human-understandable nature of equations can be mitigated and the semantic inductive bias can be cured by editing the model.

**Strengths:**

Compared to the two-step modeling approach exemplified by recently developed equation discovery models, the current work reports a data-driven model where the one-dimensional ordinary differential equation is segmented via transition points, and fitted by semantic motifs following the idea originally proposed in Kacprzyk 2023. The so-called direct semantic modeling approach allows more complexities beyond the analytical models and can integrate empirical knowledge via model editing. The definition of the problem and solution properties including the features of dynamic motifs are well formulated. Although the logic, workflow, and performance of the semantic predictor is demonstrated by using a one-dimensional model, the idea extends to higher dimensions.

**Weaknesses:**

Although in the discussion and appendix G.2, the authors proposed a roadmap for high-dimensional applications of the current model. However, there needs some elaborated discussion on the computational complexity and cost to achieve that. Even for the comparison made with the two-step model in the main text, the computational costs especially in problems with rising complexities (e.g. singularities) are not discussed.
The caption of Figure 1 is too brief, which makes the reading below very difficult (actually all figure captions are brief). The writing could be improved, for example, in lines 257-258, t_{i} and x(t_{i}) are not defined in the previous definition of p_{x}. Some content such as Fig. 10 may need to be moved to the main text since this is the key innovation in the current work. Fig. 7 may need to be merged into Fig. 6 for a better illustration of the improvement.

**Questions:**

How to technically treat singular points in a solution via the discrete representation since the determination of the positions of these points depend on the numerical grids? Does this process limit the model performance for solutions with a high density of singular points or short-wavelength features?

---

> ### Author Response · Authors · 2024-11-21
> **Response to Reviewer DLBF [1/2]**
>
> Dear Reviewer DLBF,
>
> Thank you so much for your time spent reviewing our paper.  By addressing your questions and concerns, we truly believe we have strengthened our paper. We provide our responses below and point to relevant changes in the paper.
>
> ---
>
> ### (A) Higher dimensions
>
> **Extension**
>
> We have now elaborated on extending semantic modeling to multiple dimensions in Appendix G.2. The easiest way to model multiple trajectories (or multi-dimensional trajectories) is to model each trajectory dimension independently but conditioned on the multi-dimensional initial condition $x_0\in\mathbb{R}^M$, i.e., for $M$ trajectories, we want to fit $M$ forecasting models, $F_1, \ldots, F_M$ such that for each $m\in[M]$, $F_{m}:\mathbb{R}^M \to \mathcal{C}^2$. Each $F_m$ can still be represented as $F_{\text{traj}} \circ F_{\text{sem}}^{(m)}$, where $F_{\text{traj}}$ is the same as described in the paper and $F_{sem}^{m} : \mathbb{R}^M \to \mathcal{C}\times\mathcal{P}$. $F_{sem}^{m}$ can similarly be represented using a composition map and property maps for each predicted composition. However, these maps are no longer described by univariate functions. We currently envision the composition maps to resemble decision trees (or similar structures) and the property maps as generalized additive models to make the whole structure comprehensible. Moreover, the semantic predictors can also accommodate auxiliary variables beyond the initial conditions. For instance, the parameters of the ODEs. This approach should scale to multiple dimensions as it is very modular. Each $F_{\text{sem}}^m$ can be analyzed independently. Each property map of  $F_{\text{sem}}^m$ corresponding to a particular composition can be analyzed independently. And finally, each shape function for each of the predicted properties can be analyzed independently. We believe this modularity is one of the reasons why even systems with multiple dimensions can remain understandable and, more importantly, can be edited as each change has a localized impact. In contrast, changing a single parameter in a system of ODEs may change all the trajectories in ways that may be difficult to predict without a careful analysis.
>
> **Proof of concept**
>
> This extension deserves a separate paper, but we wanted to demonstrate a proof of concept. We implemented the property maps as we defined above (each property described as a generalized additive model) and fitted data following a SIR epidemiological model for different initial conditions. To simplify the problem, we prespecify the composition map (we only train property maps). We assume $S$ follows $(s_{--c},s_{-+h})$, $I$ follows $(s_{++c},s_{+-c},s_{--c},s_{-+h})$, and $R$ follows $(s_{++c},s_{+-h})$. We have included the results in Appendix B.5, which shows the predicted trajectories and some of the property maps. The average RMSE on the test dataset is 0.019 for $S$ trajectory, 0.011 for $I$ and 0.014 for $R$. Note that the irreducible error on this dataset (caused by added Gaussian noise) is 0.01. The shown shape functions let us draw the following insights about the model:
> - The time when $I$ is at its maximum $t_{\text{max}}(I)$ is on average just below $0.2$. $I_0$ has a relatively large impact on $t_{\text{max}}(I)$ by increasing it by 0.1 for very low $I_0$ or decreasing it by 0.05 for very high $I_0$. The larger the $I_0$, the faster the maximum is achieved.
> - $S_0$ also has a negative impact on $t_{\text{max}}(I)$ but it is much smaller ($\pm 0.02$).
> - The maximum of $I$ (denoted $I_{\text{max}}$) increases linearly with both $S_0$ and $I_0$. This time $S_0$ has slightly bigger impact ($\pm 0.1$) compared to $I_0$ ($\pm 0.04$) .
> - In both $t_{\text{max}}(I)$ and $I_{\text{max}}$, the impact of $R_0$ is insignificant.
> - The horizontal asymptote of $R$ increases linearly with all three initial conditions. In particular, the shape function associated with $R_0$ has a unit slope as expected.
>
> An interesting advantage of our approach is that even though three variables describe the system, we do not need to observe all of them to fit the trajectory (similarly to the pharmacokinetic example in the paper). ODE discovery methods assume that all variables are observed, which constrains their applicability in many settings.

---

> ### Author Response · Authors · 2024-11-21
> **Response to Reviewer DLBF [2/2]**
>
> **Computational cost**
>
> We have now included the following discussion in Appendix G.2.
>
> The computational cost of extending semantic modeling to $M$ dimensions depends on the exact approach taken. We will try to make our best guess, assuming the approach remains as similar as possible to the one we describe in the paper.
>
> With our current ideas, the added cost of having multiple dimensions comes mostly from the need to perform the same tasks $M$ times. For $M$ trajectories, we need to fit $M$ composition maps and $M$ property maps.
>
> We do not expect the time to fit a single composition map to increase significantly. The main computational burden of the current implementation comes from fitting a composition to each *sample* to see how well it fits. For the same number of samples, the computation cost is going to be the same. Although there will be an added cost to find optimal decision boundaries in multiple dimensions (as opposed to one), we do not think this will be a significant burden given current efficient classification algorithms.
>
> Fitting of each property map will require $M$ times the number of parameters. However, it is important to note that the current property maps have just tens of parameters in most cases, so this number is still likely to be a few hundred at most. The number of property maps needed to be fitted (for each of the compositions) may be larger. However, a user would still like to narrow it down to a reasonable number (likely less than 10) to make sure that the composition map is understandable.
>
> Overall, we suspect the training time to increase at least linearly with the number of dimensions, but we do not expect any combinatorial explosion as often happens in symbolic regression, so for the number of trajectories, we may be interested (probably less than 10), the computational costs should not be a major concern.
>
> ---
>
> ###  (B) Critical points
> As we mentioned in the limitations, our approach is limited to trajectories with a finite composition. We have now elaborated in Appendix G.1 how this can be extended to oscillatory/periodic trajectories by meta-motifs and their associated meta-properties. The idea is to take the infinite composition (infinite sequence of motifs) and represent it using some finite representation. For instance, a trajectory $\sin(t)$ would be described as $(s_{+-c},s_{--c},s_{-+c},s_{++c})$ repeating forever, where the meta-properties may include the "frequency" and "amplitude" that may also vary with time and be itself described using compositions. With this extension, we do not expect the performance to be limited for short-wavelength features as long as the measurements of the trajectory are not too coarse.
>
> However, we admit that the current implementation may struggle if the ground-truth composition is finite but long. In that case, it may fall outside of the chosen set of compositions. We have now explicitly stated this limitation in Appendix G.1.
>
> ---
>
> ### (C) Writing
> Thank you for your comments on how to improve our presentation. As suggested, we have now moved Fig. 10 to the main text and expanded the caption of Figure 1.

---

> ### Author Response · Authors · 2024-11-26
>
> Dear Reviewer DLBF,
>
> Thank you once again for your thoughtful feedback, which has significantly improved our paper. We have carefully addressed your comments in our rebuttal and hope you find our responses satisfactory.
>
> We wanted to check if you have any remaining concerns or questions we can address. If our revisions meet your expectations, we hope you might consider increasing your score.
>
> We truly appreciate your time and invaluable insights!
>
> Kind regards,
>
> Authors

---

> > ### Comment · Reviewer_DLBF · 2024-11-30
> > **Comments on the authors reply**
> >
> > I would like to thank the authors for addressing my comments. The work has been significantly improved.

---

> > > ### Author Response · Authors · 2024-12-03
> > >
> > > Dear Reviewer DLBF,
> > >
> > > Thank you for your response and for acknowledging the significant improvements to our paper. We deeply value your thoughtful feedback.
> > >
> > > If there are any remaining concerns or aspects you feel require further clarification, we would be happy to address them. Given the revisions and your positive feedback, we hope the paper might now align more closely with the criteria for a higher rating.
> > >
> > > Thank you again for your time and insights.
> > >
> > > Kind regards,
> > >
> > > Authors

---

### Official Review · Reviewer_rHqF · 2024-11-03

**Soundness:** 3
**Presentation:** 4
**Contribution:** 4
**Rating:** 8
**Confidence:** 4

**Summary:**

This paper presents an approach to modeling the solutions (trajectories) of dynamical systems by bypassing the need to derive (discover) the closed-form of ordinary differential equations (ODEs). Instead of the conventional two-step approach (discovering an ODE and then analyzing it), the authors propose *direct semantic modeling*.

The promised contributions include:

1. A shift from Syntactic to Semantic representations. The latter delivers a formal means to describing the behavior of trajectories and system dynamics.

2. A direct Semantic Modeling framework, which essentially comprises a *trajectory predictor* that generates trajectories fitting the semantic description. This model alleviates the need for attaching a compact symbolic equation, and employs so-called "motifs" to describe trajectories, while remaining interpretable.

3. Through a set of mostly 1D experiments, including a pharmacokinetic model, the authors demonstrate that the proposed semantic ODE offers improved performance, interpretability, and robustness to noise compared to traditional models like SINDy.

**Strengths:**

The paper puts forth a scheme that moves away from the two-step modeling process of equation discovery and analysis, proposing instead that the semantic (behavioral) representation of a system can be derived directly from data, which in essence renders it closer to a solution discovery (as opposed to) equation discovery scheme. The concept of focusing on semantic rather than syntactic representation is original, as it allows modelers to interact with and interpret the behavior of systems without delving into complex symbolic mathematics. The originality here lies in both the reframing of the problem and the technical solution that bypasses the need for closed-form equations.

The paper demonstrates a clear and well presented theoretical development and practical implementation. Key symbols are defined in a notation table and important terms such as semantic representation, syntactic representation, and direct semantic modeling are well-defined, with illustrative examples and figures that clarify the difference between traditional ODE modeling and the proposed approach. The evaluation experiments are cover a range of dynamical systems (albeit low dimensional), including pharmacokinetic models, systems with delay differential equations, and integro-differential equations. The authors further validate the practical usability of Semantic ODE, with case studies and error analyses that underscore its flexibility, robustness, and potential for adaptation.

The potential significance of this paper is substantial. By removing the reliance on closed-form equations and focusing on the semantic representation of system behaviors, the proposed method has applications across multiple domains, where dynamical systems modeling is essential. Additionally, the demonstrated robustness to noise and the flexibility of Semantic ODE to adapt to various types of differential equations indicate that this method could be robust, at least for the type of low dimensional problems that are here treated.

**Weaknesses:**

Despite the interesting idea, which draws some ties to a (flexible) curve fitting approach, certain weaknesses are identified as follows:

1. **Limitations Due to Motif-Based Semantic Representation**: While the paper argues that semantic modeling offers flexibility over traditional ODE discovery methods, the reliance on a predefined set of motifs to construct semantic representations introduces notable constraints. The specific motifs used limit the types of trajectories that can be effectively modeled, potentially narrowing the applicability of the method to systems with well-defined, predictable patterns. This approach risks excluding complex or novel trajectory shapes that fall outside the pre-selected motif set, thereby introducing a form of inductive bias that contradicts the method’s stated goal of flexibility.

2. **Scalability and Extension to Higher Dimensions**: The current implementation focuses exclusively on one-dimensional systems, and while the authors mention potential future work on extending to higher-dimensional systems, they do not provide concrete strategies for achieving this. The direct semantic modeling approach, as presented, lacks a clear path for scaling motifs and trajectory predictions in multiple dimensions, which is crucial for broadening applicability in fields like biology, physics, and engineering where systems often involve multiple interacting variables.

3. **Evaluation Against More Complex Baselines**: While the authors compare Semantic ODE to methods like SINDy, the experiments could be strengthened by benchmarking against more advanced and recent models in dynamical system solution discovery. Specifically, neural ODEs (e.g., Chen et al., 2018) and hybrid models that combine symbolic and neural approaches (like symbolic regression combined with neural networks) could serve as competitive baselines, as these methods have demonstrated flexibility and scalability, even in higher dimensions.  Do take note of the fact that such schemes as well, e.g. Neural ODEs, are linked to interpretability due to their ties to known dynamical system forms (e.g. state-state ODE formulation)

4. **Conceptual Contribution**: At the end of the day, it is not so clear why using predefined motifs is really different to assumptions of classes of solutions (as SINDy does) that sever as a bases to define convexity or concavity (which is really what the motifs do). One should also keep in mind that equations discovery is not always a necessary step. It is, however, a very useful step for understanding the type of physics that is underlying the problem. How does the proposed scheme offer the same granularity of information?

5. **Robustness to Unpredictable System Behaviors**: While the authors demonstrate that Semantic ODE is robust to noise, they do not address its performance in systems with unpredictable or highly chaotic behaviors. Since the semantic model relies on motif sequences, unpredictable behaviors could challenge the motif-based representation and disrupt accurate prediction or interpretation. Why not try this for example on nonlinear systems, such as the Duffing oscillator.

**Questions:**

It would be helpful if the authors reply to the following questions, which are linked to the previously identified weaknesses.

1. How flexible is the proposed motif-based representation in accommodating unforeseen trajectory shapes or behaviors? What kind of motifs would the authors suggest adding if new trajectory patterns arise that fall outside the current motif set?

2. Extending to higher-dimensional systems appears non-trivial within the current approach. Do the authors have any specific strategies or theoretical insights on how they might define and handle multi-dimensional semantic representations and motifs?

3. Why did the authors choose not to include comparisons with neural ODEs or hybrid methods that combine neural networks with symbolic regression, which have become prominent in modeling complex systems? Could these methods offer competitive advantages in interpretability or performance?

4. Do the authors believe that the proposed scheme offers the same granularity of information as i) equation discovery scheme and ii) solution discovery schemes making use of specific analytical functions? If so, in what sense?

4. Since the model relies on motifs for capturing trajectory shapes, how would it perform in the presence of chaotic or unpredictable system dynamics? Have the authors tested the model’s performance on chaotic systems, and if so, what were the results?
If possible, testing on a few chaotic systems or adding a discussion on expected limitations in handling chaotic behaviors could clarify the approach’s robustness and generalizability. If chaotic dynamics prove problematic, the authors could suggest potential modifications for better handling non-smooth or irregular trajectories.

5. What specific adjustments or extensions do the authors envision to broaden the adoption of Semantic ODE? Are there plans for enhancing the method to accommodate periodic or oscillatory behaviors that cannot be fully captured within finite compositions?

---

> ### Author Response · Authors · 2024-11-21
> **Response to Reviewer rHqF [1/3]**
>
> Dear Reviewer rHqF,
>
> Thank you very much for such a comprehensive and insightful review. We are glad you found our work original, paper clear, and potential significance substantial. By addressing your questions and concerns, we truly believe we have strengthened our paper. We provide our responses below and point to relevant changes in the paper.
>
> ---
> ### (A) Flexibility of motif-based semantic representation
>
> **Representation theorem**
>
> We have proved the following representation theorem that shows the generality of our used motifs.
>
> > Any function $x \in \mathcal{C}^2(t_0, +\infty)$ whose second derivative vanishes at a finite number of points (i.e., $|\{t \ : \  \ddot{x}(t) = 0\}| < \infty$) can be described as a finite composition constructed from proposed motifs.
>
> The proof is provided in Appendix G.5.
>
> **Extension to infinite compositions**
>
> As we mentioned in the limitations, our approach is limited to trajectories with a finite composition. We have now elaborated in Appendix G.1 how this can be extended to oscillatory/periodic trajectories by meta-motifs and their associated meta-properties. The idea is to take the infinite composition (infinite sequence of motifs) and represent it using some finite representation. For instance, a trajectory $\sin(t)$ would be described as $(s_{+-c},s_{--c},s_{-+c},s_{++c})$ repeating forever, where the meta-properties may include the "frequency" and "amplitude" that may also vary with time and be itself described using compositions.  We admit that this extension still assumes some underlying pattern behind the infinite sequence of motifs that can be "compressed" into a shorter representation. This is slightly related to the Kolmogorov complexity of this sequence. If the infinite sequence of motifs is algorithmically random, then motif-based representation may not be helpful. We have now made it clear in the limitations.
>
> **Dependence on a pre-defined set**
>
> Although our current implementation depends on a pre-specified set of compositions (e.g., all possible compositions of length less than $5$), ODE discovery methods also rely on the specification of the token set (e.g., a library of terms in SINDy or a list of operators and functions in symbolic regression). Both of them introduce a form of inductive bias. As we argue in Section 6.1, *semantic* inductive biases may be more meaningful and intuitive for users than syntactic ones. Especially when there is no prior knowledge about the functional form of the ODE that best describes the dynamical system.
>
> ---
> ### (B) Higher dimensions
>
> We have now elaborated on extending semantic modeling to multiple dimensions in Appendix G.2. The easiest way to model multiple trajectories (or multi-dimensional trajectories) is to model each trajectory dimension independently but conditioned on the multi-dimensional initial condition $x_0\in\mathbb{R}^M$, i.e., for $M$ trajectories, we want to fit $M$ forecasting models, $F_1, \ldots, F_M$ such that for each $m\in[M]$, $F_{m}:\mathbb{R}^M \to \mathcal{C}^2$. Each $F_m$ can still be represented as $F_{\text{traj}} \circ F_{\text{sem}}^{(m)}$, where $F_{\text{traj}}$ is the same as described in the paper and $F_{sem}^{m} : \mathbb{R}^M \to \mathcal{C}\times\mathcal{P}$. $F_{sem}^{m}$ can similarly be represented using a composition map and property maps for each predicted composition. However, these maps are no longer described by univariate functions. We currently envision the composition maps to resemble decision trees (or similar structures) and the property maps as generalized additive models (GAMs) so that the whole structure remains comprehensible. Moreover, the semantic predictors can also accommodate auxiliary variables beyond the initial conditions. For instance, the parameters of the ODEs. This approach should scale to multiple dimensions as it is very modular. Each $F_{\text{sem}}^m$ can be analyzed independently. Each property map of  $F_{\text{sem}}^m$ corresponding to a particular composition can be analyzed independently. And finally, each shape function for each of the predicted properties can be analyzed independently. We believe this modularity is one of the reasons why even systems with multiple dimensions can remain understandable and, more importantly, can be edited as each change has a localized impact. In contrast, changing a single parameter in a system of ODEs may result in a change of all the trajectories in ways that may be difficult to predict without a careful analysis.
>
> This extension deserves a separate paper, but we wanted to demonstrate a proof of concept. We implemented the property maps as we defined above (each property described as GAMs) and fitted data following an SIR epidemiological model for different initial conditions. To simplify the problem, we prespecify the composition map (we only train property maps). We assume $S$ follows $(s_{--c},s_{-+h})$, $I$ follows $(s_{++c},s_{+-c},s_{--c},s_{-+h})$, and $R$ follows $(s_{++c},s_{+-h})$.

---

> ### Author Response · Authors · 2024-11-21
> **Response to Reviewer rHqF [2/3]**
>
> We have included the results in Appendix B.5, which shows the predicted trajectories and some of the property maps. The average RMSE on the test dataset is 0.019 for $S$ trajectory, 0.011 for $I$ and 0.014 for $R$. Note that the irreducible error on this dataset (caused by added Gaussian noise) is 0.01. The shown shape functions let us draw the following insights about the model:
> - The time when $I$ is at its maximum $t_{\text{max}}(I)$ is on average just below $0.2$. $I_0$ has a relatively large impact on $t_{\text{max}}(I)$ by increasing it by 0.1 for very low $I_0$ or decreasing it by 0.05 for very high $I_0$. The larger the $I_0$, the faster the maximum is achieved.
> - $S_0$ also has a negative impact on $t_{\text{max}}(I)$ but it is much smaller ($\pm 0.02$).
> - The maximum of $I$ (denoted $I_{\text{max}}$) increases linearly with both $S_0$ and $I_0$. This time $S_0$ has slightly bigger impact ($\pm 0.1$) compared to $I_0$ ($\pm 0.04$) .
> - In both $t_{\text{max}}(I)$ and $I_{\text{max}}$, the impact of $R_0$ is insignificant.
> - The horizontal asymptote of $R$ increases linearly with all three initial conditions. In particular, the shape function associated with $R_0$ has a unit slope as expected.
>
> An interesting advantage of our approach is that even though three variables describe the system, we do not need to observe all of them to fit the trajectory (similar to the pharmacokinetic example in the paper). ODE discovery methods assume that all variables are observed, which constrains their applicability in many settings.
>
> ---
> ### (C) Comparison with black-box baselines
>
> We have now added a comparison with three black box approaches: NeuralODE [1], NeuralLaplace [2], and DeepONet [3]. All results are now included in Table 3 and can be seen below. In nearly all settings, Semantic ODE outperforms the black-box method. This is likely because we operate in a low-data regime, where we always have fewer than 200 samples measured at most 20 times throughout the trajectory.
>
> |                | Logistic Growth  | Logistic Growth  | General ODE      | General ODE      | Pharmacokinetic  | Pharmacokinetic  | Mackey-Glass     | Mackey-Glass     | Integro-DE       | Integro-DE       |
> | -------------- | ---------------- | ---------------- | ---------------- | ---------------- | ---------------- | ---------------- | ---------------- | ---------------- | ---------------- | ---------------- |
> | Noise Level    | low              | high             | low              | high             | low              | high             | low              | high             | low              | high             |
> | Neural ODE     | $0.023_{(.004)}$ | $0.197_{(.005)}$ | $0.029_{(.005)}$ | $0.075_{(.006)}$ | $0.036_{(.008)}$ | $0.203_{(.007)}$ | $0.177_{(.010)}$ | $0.194_{(.010)}$ | $0.073_{(.007)}$ | $0.215_{(.009)}$ |
> | Neural Laplace | $0.126_{(.036)}$ | $0.230_{(.017)}$ | $0.108_{(.030)}$ | $0.138_{(.023)}$ | $0.100_{(.022)}$ | $0.229_{(.013)}$ | $0.057_{(.006)}$ | $0.094_{(.009)}$ | $0.075_{(.044)}$ | $0.249_{(.014)}$ |
> | DeepONet       | $0.184_{(.040)}$ | $0.306_{(.023)}$ | $0.160_{(.033)}$ | $0.195_{(.027)}$ | $0.058_{(.010)}$ | $0.212_{(.005)}$ | $0.107_{(.014)}$ | $0.132_{(.012)}$ | $0.100_{(.015)}$ | $0.230_{(.014)}$ |
> | Semantic ODE | $0.015_{(.005)}$ | $0.200_{(.009)}$ | $0.015_{(.001)}$ | $0.068_{(.002)}$ | $0.023_{(.014)}$ | $0.211_{(.015)}$ | $0.037_{(.003)}$ | $0.075_{(.003)}$ | $0.025_{(.003)}$ | $0.203_{(.007)}$ |
>
> ---
> ### (D) Conceptual comparison with SINDy
>
> As described above, our approach and SINDy contain assumptions on the class of solutions. We argue that often the users may have some prior knowledge about the behavior of the dynamical system (e.g., the shape of the trajectory, its asymptotic behavior, or the fact that it is always increasing at the beginning) but not necessarily about its functional form. It may also be the case that no compact closed-form ODE exists. For instance, the dynamics may depend on history (e.g., delay and integral differential equations), involve unobserved variables (e.g., pharmacological model), or empirically determined functions (e.g., stress-strain curve). Even if we are able to translate our prior knowledge into assumptions on the functional form, enforcing them may be challenging.
>
> The library of terms in SINDy on its own is usually insufficient to determine the convexity and monotonicity of the predicted trajectories. Even a small library of terms ($x$, $t$, $1$) can give trajectories that are increasing, decreasing, convex, and concave, as well as trajectories with trend changes, depending on the exact coefficients as well as the initial condition and the time domain. So, there is a substantial difference between choosing the library of terms and the set of admissible compositions.

---

> > ### Comment · Reviewer_rHqF · 2024-11-24
> > **Feedback to Revised Manuscript**
> >
> > Thank you for your thoughtful responses. Your clarifications address some of my primary concerns, and I will revise my score accordingly.

---

> ### Author Response · Authors · 2024-11-21
> **Response to Reviewer rHqF [3/3]**
>
> ### (E) Granularity of information
> Thank you for drawing our attention to this very interesting point about the granularity of information. We have now included the following discussion in Appendix G.3
>
> We believe the granularity of information (in terms of the shape of the trajectory, its maxima, minima, and asymptotic behavior) should be comparable between direct semantic modeling and other approaches, as long as all approaches are framed as solutions to finding a forecasting model. The key difference is that direct semantic modeling gives immediate access to the semantic representation of the model without any further analysis and allows us to easily edit the model and incorporate semantic inductive biases during training.
>
> There is, however, one way in which ODEs *may* offer more granular information by assuming a particular causal structure. *Note*: it may not be correct, and it makes them less flexible. For a discretized ODE, we can write the underlying causal graph as $x(t_0) \to x(t_0 + \Delta t) \to \ldots \to x(t)$. At the same time, our model assumes a more general $x(t_0) \to x(t)$. This generality is useful (as we show in Section 6.4), and it does not negatively impact the performance. However, this model does not allow us to do interventions where we change the value of $x$ at a particular $t$ and predict the future. ODEs do allow for this kind of inference, but of course, there is no guarantee that it is correct. For instance, a system governed by a delayed differential equation has a different causal structure. We believe direct semantic modeling in the future can be extended to accommodate some kinds of intervention, where they can be described as additional features beyond the initial conditions.
>
> ---
>
> ### (F) Chaotic behavior
> We could not find chaotic systems that would not have some kind of oscillatory behavior (i.e., that could be described by a finite composition). As such, our current implementation of Semantic ODEs would not be able to successfully predict the behavior beyond the seen time domain. However, we could use it to predict on a bounded time domain. That is why, following your suggestion, we have included an experiment with the Duffing oscillator. Results are shown below and in Appendix B.7. Although our method achieves decent results, this performance would drop drastically if we tried to predict beyond the training time domain. We have clarified this limitation in Appendix G.1.
>
> | | Duffing | Duffing |
> | --- | --- | --- |
> | Noise Level | low | high |
> | SINDy-5 | $0.278_{(.032)}$ | $0.389_{(.059)}$ |
> | WSINDy-5 | $0.262_{(.033)}$ | $0.361_{(.072)}$ |
> | PySR-20 | $0.312_{(.049)}$ | $0.396_{(.020)}$ |
> | SINDy | $0.284_{(.026)}$ | $0.386_{(.022)}$ |
> | WSINDy | $0.263_{(.027)}$ | $0.339_{(.037)}$ |
> | Neural ODE | $0.212_{(.018)}$ | $0.291_{(.021)}$ |
> | Neural Laplace | $0.176_{(.032)}$ | $0.299_{(.017)}$ |
> | DeepONet | $0.429_{(.084)}$ | $0.528_{(.066)}$ |
> | Semantic ODE | $0.090_{(.012)}$ | $0.251_{(.027)}$ |
>
>
> In general, Semantic ODEs can model significant changes in the trajectory following a slight change in the initial condition (as is characteristic for chaotic systems). As the composition map is discontinuous at points where the composition changes, it can model a sudden change in the shape of the trajectory.
>
> ---
>
> ### Questions
>
> **Q1**: As long as the trajectory is twice continuously differentiable and somewhat well-behaved, we do not think it can fall outside the proposed motif set. However, we agree it can fall outside the chosen composition library if, e.g., the number of inflection points is very large or infinite. Please see our response in "(A) Flexibility of motif-based semantic representation".
>
> **Q2**: Response in "(B) Higher dimensions"
>
> **Q3**:  Response in "(C) Comparison with black-box baselines"
>
> **Q4**: Response in "(E) Granularity of information"
>
> **Q5**: Response in "(F) Chaotic behavior"
>
> **Q6**: This paper lays the vital groundwork for direct semantic modeling with many future opportunities to extend it to even more settings, broadening its applicability. Our current focus is indeed on extending the framework by meta-motifs to accommodate some infinite compositions and to extend to multi-dimensional inputs as described in our response in "(B) Higher dimensions".
>
> ---
>
> ### References
> [1] Chen, R. T., Rubanova, Y., Bettencourt, J., & Duvenaud, D. K. (2018). Neural ordinary differential equations.
>
> [2] Holt, S. I., Qian, Z., & van der Schaar, M. (2022). Neural laplace: Learning diverse classes of differential equations in the laplace domain.
>
> [3] Lu, L., Jin, P., & Karniadakis, G. E. (2019). Deeponet: Learning nonlinear operators for identifying differential equations based on the universal approximation theorem of operators.

---

> ### Author Response · Authors · 2024-11-25
>
> Dear Reviewer rHqF,
>
> We are deeply grateful for your thoughtful consideration of our responses and for increasing your score. Your feedback has been invaluable in refining our work. If there are any additional suggestions or concerns you have, we would be more than happy to address them to further improve our paper.
>
> Thank you once again for your support and encouragement.
>
> Kind regards,
>
> Authors

---

### Author Response · Authors · 2024-11-21
**Summary of main changes**

We are very grateful to all reviewers for their time reviewing our work. We are glad reviewers appreciated its novelty, potential significance, and thorough empirical validation. We thank the reviewers for raising many interesting questions and suggesting ways of further improving the paper. We have summarized the main changes below, and we truly believe they strengthened the paper. All changes in the text are marked blue.

### New experiments
- Comparison with black box models (Neural ODE, Neural Laplace, DeepONet) in Table 3 (**rHqF**, **DbZG**).
- Extension to multiple dimensions (proof of concept) in Appendix B.5 (**rHqF**, **DLBF**).
- Evaluation on a chaotic system (Duffing oscillator) in Appendix B.7 (**rHqF**).
- Evaluation on real datasets in Appendix B.8 (**BR8f**).
- Bifurcation point identification in Appendix B.9 (**BR8f**).
- Generalization outside of the training distribution in Appendix B.10 (**DbZG**, **BR8f**).

### Additional/Elaborated discussion
- Finite compositions and possible extensions in Appendix G.1 (**rHqF**, **DLBF**).
- Chaotic systems in Appendix G.1 (**rHqF**).
- A more detailed roadmap to multiple dimensions in Appendix G.2 (**rHqF**, **DLBF**).
- Granularity of information in Appendix G.3 (**rHqF**).
- Partial differential equations in Appendix G.4 (**BR8f**).
- Sharing samples between property maps in Appendix G.6 (**DbZG**)

### New theoretical results
- Flexibility of motif-based semantic representation in Appendix G.5 (**rHqF**).

### Changes to the main text
- Moved the figure with the block diagram showing the main elements of training the semantic predictor to the main text. (**DLBF**)

---

### Meta-Review · Area_Chair_tvrV · 2024-12-19

**Metareview:**

This paper introduces an approach to modeling dynamical systems by replacing the traditional two-step process of equation discovery and analysis with direct "semantic modeling". The authors propose a framework that bypasses the need for closed-form ordinary differential equations. Instead, they propose to focus instead on predicting semantic representations that describe the behavior of system directly from data. The approach enables the integration of prior knowledge and facilitates model editing. This makes the approach suited for applications in domains such as pharmacokinetics and low-dimensional dynamical systems.

Conceptually, the shift from syntactic (equation-based) to semantic (behavioral) representations is both interesting: this offers a different way to model and interact with dynamical systems. The paper provides a rigorous theoretical foundation, with a clear problem formulations and discussions of solution properties. In practice, the method performs well across various use cases, including delay differential equations and pharmacokinetic modelling. While the experiments focus on low-dimensional systems, the authors convincingly demonstrate the potential for extending the approach to higher-dimensional settings

The article offers an important contribution to the ML community by addressing an important limitation in traditional ODE modeling and opening up new avenues for data-driven analysis of dynamical systems.

**Additional Comments On Reviewer Discussion:**

* New experiments comparing the proposed method to black-box models such as Neural ODE, Neural Laplace, and DeepONet, as well as proof-of-concept extensions to multiple dimensions and evaluations on challenging scenarios, including chaotic systems (Duffing oscillator), real datasets, bifurcation point identification, and generalization outside the training distribution.
* Theoretical discussions were also expanded, including topics such as finite compositions, chaotic systems, and extensions to multiple dimensions, alongside a demonstration of the flexibility of the motif-based semantic representation. Additional clarifications and improvements to the main text, such as moving the block diagram of the semantic predictor to the main paper.

---

### Decision · Program_Chairs · 2025-01-22

Accept (Poster)